# Overcoming the Sim-to-Real Gap: Leveraging Simulation to Learn to Explore for Real-World RL

**Andrew Wagenmaker**[*]
University of California, Berkeley

**Kevin Huang**
University of Washington

**Liyiming Ke**
University of Washington

**Kevin Jamieson**
University of Washington

**Abhishek Gupta**
University of Washington

## Abstract

In order to mitigate the sample complexity of real-world reinforcement learning, common practice is to first train a policy in a simulator where samples are cheap, and then deploy this policy in the real world, with the hope that it generalizes effectively. Such *direct sim2real* transfer is not guaranteed to succeed, however, and in cases where it fails, it is unclear how to best utilize the simulator. In this work, we show that in many regimes, while direct sim2real transfer may fail, we can utilize the simulator to learn a set of *exploratory* policies which enable efficient exploration in the real world. In particular, in the setting of low-rank MDPs, we show that coupling these exploratory policies with simple, practical approaches—least-squares regression oracles and naive randomized exploration—yields a polynomial sample complexity in the real world, an exponential improvement over direct sim2real transfer, or learning without access to a simulator. To the best of our knowledge, this is the first evidence that simulation transfer yields a provable gain in reinforcement learning in settings where direct sim2real transfer fails. We validate our theoretical results on several realistic robotic simulators and a real-world robotic sim2real task, demonstrating that transferring exploratory policies can yield substantial gains in practice as well.

## 1 Introduction

Over the last decade, reinforcement learning (RL) techniques have been deployed to solve a variety of real-world problems, with applications in robotics, the natural sciences, and beyond [27, 54, 52, 26, 46, 23]. While promising, the broad application of RL methods has been severely limited by its large sample complexity—the number of interactions with the environment required for the algorithm to learn to solve the desired task. In applications of interest, it is often the case that collecting samples is very costly, and the number of samples required by RL algorithms is prohibitively expensive.

In many domains, while collecting samples in the desired deployment environment may be very costly, we have access to a *simulator* where the cost of samples is virtually nonexistent. As a concrete example, in robotic applications where the goal is real-world deployment, directly training in the real world typically requires an infeasibly large number of samples. However, it is often possible to obtain a simulator—derived from first principles or knowledge of the robot's actuation—which provides an approximate model of the real-world deployment environment. Given such a simulator, common practice is to first train a policy to accomplish the desired task in the simulator, and then deploy it in the real world, with the hope that the policy generalizes effectively from the simulator to the goal deployment environment. Indeed, such "sim2real" transfer has become a key piece in the

---

[*]Correspondance to: `ajwagen@berkeley.edu`

38th Conference on Neural Information Processing Systems (NeurIPS 2024).

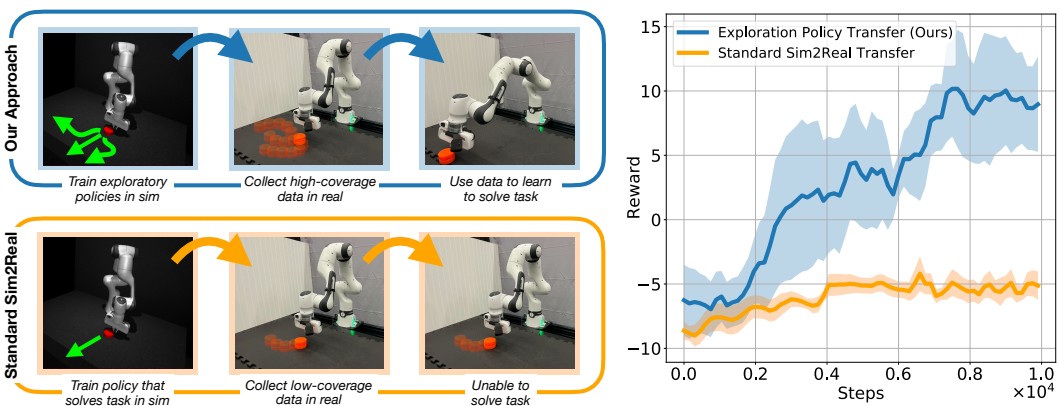

Figure 1: **Left:** Overview of our approach compared to standard sim2real transfer on puck pushing task. Standard sim2real transfer first trains a policy to solve the goal task in sim and then transfers this policy to real. This policy may fail to solve the task in real due to the sim2real gap, and furthermore may not provide sufficient data coverage to successfully learn a policy that does solve the goal task in real. In contrast, our approach trains a set of exploratory policies in sim which achieve high-coverage data when deployed in real, even if they are unable to solve the task 0-shot. This high-coverage data can then be used to successfully learn a policy that solves the goal task in real. **Right:** Quantitative results running our approach on the puck pushing task illustrated on left, compared to standard sim2real transfer. Over 6 real-world trials, our approach solves the task 6/6 times while standard sim2real transfer solves the task 0/6 times.

application of RL to robotic settings, as well as many other domains of interest such as the natural sciences [12, 15], and is a promising approach towards reducing the sample complexity of RL in real-world deployment [19, 4, 18].

Effective sim2real transfer can be challenging, however, as there is often a non-trivial mismatch between the simulated and real environments. The real world is difficult to model perfectly, and some discrepancy is inevitable. As such, directly transferring the policy trained in the simulator to the real world often fails, the mismatch between sim and real causing the policy—which may perfectly solve the task in sim—to never solve the task in real. While some attempts have been made to address this—for example, utilizing domain randomization to extend the space of environments covered by simulator [60, 49], or finetuning the policy learned in sim in the real world [50, 73]—these approaches are not guaranteed to succeed. In settings where such methods fail, can we still utilize a simulator to speed up real-world RL?

In this work we take steps towards developing principled approaches to sim2real transfer that addresses this question. Our key intuition is that it is often *easier to learn to explore than to learn to solve the goal task*. While solving the goal task may require very precise actions, collecting high-quality exploratory data can require significantly less precision. For example, successfully solving a complex robotic manipulation task requires a particular sequence of motions, but obtaining a policy that will interact with the object of interest in some way, providing useful exploratory data on its behavior, would require significantly less precision.

Formally, we show that, in the setting of low-rank MDPs where there is a mismatch in the dynamics between the "sim" and "real" environments, even when this mismatch is such that direct sim2real transfer fails, under certain conditions we can still effectively transfer a set of *exploratory* policies from sim to real. In particular, we demonstrate that access to such exploratory policies, coupled with random exploration and a least-squares regression oracle—which are insufficient for efficient learning on their own, but often still favored in practice due to their simplicity—enable provably efficient learning in real. Our results therefore demonstrate that simulators, when carefully applied, can yield a provable—exponential—gain over both naive sim2real transfer and learning without a simulator, and enable algorithms commonly used in practice to learn efficiently.

Furthermore, our results motivate a simple, easy-to-implement algorithmic principle: rather than training and transferring a policy that solves the task in the simulator, utilize the simulator to train a set of exploratory policies, and transfer these, coupled with random exploration, to generate high

quality exploratory data in real. We show experimentally—through a realistic robotic simulator and real-world sim2real transfer problem on the Franka robot platform—that this principle of transferring exploratory policies from sim to real yields a significant practical gain in sample efficiency, often enabling efficient learning in settings where naive transfer fails completely (see Figure 1).

## 2 Related Work

**Provable Transfer in RL.** Perhaps the first theoretical result on transfer in RL is the "simulation lemma", which transforms a bound on the total-variation distance between the dynamics to a bound on policy value [24, 25, 6, 22]—we argue that we can do significantly better with exploration transfer. More recent work has considered transfer in the setting of block MDPs [34], but requires relatively strong assumptions on the similarity between source and target MDPs, or the meta-RL setting [69], but only consider tabular MDPs, and assume the target MDP is covered by the training distribution. Perhaps most relevant to this work is the work of [36], which presents several lower bounds showing that efficient transfer in RL is not feasible in general. In relation to this work, our work can be seen as providing a set of sufficient conditions that do enable efficient transfer; the lower bounds presented in [36] do not hold in the low-rank MDP setting we consider. Several other works exist, but either consider different types of transfer than what we consider (e.g., observation space mismatch), or only learn a policy that has suboptimality bounded by the sim2real mismatch [37, 56, 58]. Another somewhat tangential line of work considers representation transfer in RL, where it is assumed the source and target tasks share a common representation [35, 10, 2]. We remark as well that the formal sim2real setting we consider is a special case of the MF-MDP setting of [53].

**Simulators and Low-Rank MDPs.** Several existing works show that there are provable benefits to training a policy in "simulation" due to the ability to reset on command [67, 33, 5, 68, 70, 42]. These works do not consider the transfer problem, however. The setting of linear and low-rank MDPs which we consider has seen a significant amount of attention over the last several years, and many provably efficient algorithms exist [21, 1, 62, 63, 43, 41]. These works typically assume access to powerful oracles which enable efficient learning; we only consider access to a simple regression oracle. Beyond the theory literature, recent work has also shown that low-rank MDPs can effectively model a variety of standard RL settings in practice [72].

**Sim2Real Transfer in Practice.** The sim2real literature is vast and we only highlight particularly relevant works here; see [74] for a full survey. To mitigate the inconsistency between the simulator and real world's physical parameters and modeling, domain randomization creates a variety of simulated environments with randomized properties to develop a robust policy [60, 49, 44, 8, 39]. Domain adaptation instead constructs encoding of deployment conditions (e.g., physical condition or past histories) and adapts to the deployment environment by matching the encoding [29, 9, 66, 55, 38, 40]. In contrast, our work assumes a fundamental sim2real mismatch where we do not expect the real system to match the simulator for any parameter settings. A related line of work shows that policies trained with robust exploration strategies generalize better to disturbed or unseen environments [13, 20]. Our work is complimentary to this work in that our goal is not to transfer a policy that solves the task in new environment, but rather explores the environment.

## 3 Preliminaries

We let $\triangle_{\mathcal{X}}$ denote the set of distributions over set $\mathcal{X}$, $[H] := \{1, 2, \ldots, H\}$, and $\|P - Q\|_{\mathrm{TV}}$ the total-variation distance between distributions $P$ and $Q$.

**Markov Decision Processes.** We consider the setting of episodic Markov Decision Processes (MDPs). An MDP is denoted by a tuple $\mathcal{M} = (\mathcal{S}, \mathcal{A}, \{P_h\}_{h=1}^H, \{r_h\}_{h=1}^H, s_1, H)$, where $\mathcal{S}$ denotes the set of states, $\mathcal{A}$ the set of actions, $P_h : \mathcal{S} \times \mathcal{A} \to \triangle_{\mathcal{S}}$ the transition function, $r_h : \mathcal{S} \times \mathcal{A} \to [0, 1]$ the reward (which we assume is deterministic and known), $s_1$ the initial state, and $H$ the horizon. We assume $\mathcal{A}$ is finite and denote $A := |\mathcal{A}|$. Interaction with an MDP starts from state $s_1$, the agent takes some action $a_1$, transitions to state $s_2 \sim P_1(\cdot \mid s_1, a_1)$, and receives reward $r_1(s_1, a_1)$. This process continues for $H$ steps at which points the episode terminates, and the process resets.

The goal of the learner is to find a policy $\pi = \{\pi_h\}_{h=1}^H$, $\pi_h : \mathcal{S} \to \triangle_{\mathcal{A}}$, that achieves maximum reward. We can quantify the reward received by some policy $\pi$ in terms of the value and $Q$-value

functions. The $Q$-value function is defined as $Q_h^\pi(s,a) := \mathbb{E}^\pi[\sum_{h'=h}^H r_{h'}(s_{h'}, a_{h'}) \mid s_h = s, a_h = a]$, and value function is defined in terms of the $Q$-value function as $V_h^\pi(s) := \mathbb{E}_{a \sim \pi_h(\cdot|s)}[Q_h^\pi(s,a)]$. The value of policy $\pi$, its expected reward, is denoted by $V_0^\pi := V_1^\pi(s_1)$, and the value of the optimal policy, the maximum achievable reward, by $V_0^\star := \sup_\pi V_0^\pi$.

In this work we are interested in the setting where we wish to solve some task in the "real" environment, represented as an MDP, and we have access to a simulator which approximates the real environment in some sense. We denote the real MDP as $\mathcal{M}^{\mathsf{real}}$, and the simulator as $\mathcal{M}^{\mathsf{sim}}$. We assume that $\mathcal{M}^{\mathsf{real}}$ and $\mathcal{M}^{\mathsf{sim}}$ have the same state and actions spaces, reward function, and initial state, but different transition functions, $P^{\mathsf{real}}$ and $P^{\mathsf{sim}}$. We denote value functions in $\mathcal{M}^{\mathsf{real}}$ and $\mathcal{M}^{\mathsf{sim}}$ as $V_h^{\mathsf{real},\pi}(s)$ and $V_h^{\mathsf{sim},\pi}(s)$, respectively. We make the following assumption.

**Assumption 1.** *For all $(s,a,h) \in \mathcal{S} \times \mathcal{A} \times [H]$ and some $\epsilon_{\mathrm{sim}} > 0$, we have:*
$$\|P_h^{\mathsf{real}}(\cdot \mid s,a) - P_h^{\mathsf{sim}}(\cdot \mid s,a)\|_{\mathrm{TV}} \le \epsilon_{\mathrm{sim}}.$$

We do not assume that the value of $\epsilon_{\mathrm{sim}}$ is known, simply that there exists some such $\epsilon_{\mathrm{sim}}$.

**Function Approximation.** In order to enable efficient learning, some structure on the MDPs of interest is required. We will assume that $\mathcal{M}^{\mathsf{real}}$ and $\mathcal{M}^{\mathsf{sim}}$ are low-rank MDPs, as defined below.

**Definition 3.1** (Low-Rank MDP). *We say an MDP is a low-rank MDP with dimension $d$ if there exists some featurization $\phi : \mathcal{S} \times \mathcal{A} \to \mathbb{R}^d$ and measure $\mu : [H] \times \mathcal{S} \to \mathbb{R}^d$ such that:*
$$P_h(\cdot \mid s,a) = \langle \phi(s,a), \mu_h(\cdot) \rangle, \quad \forall s,a,h.$$
We assume that $\|\phi(s,a)\|_2 \le 1$ for all $(s,a)$, and for all $h$, $\||\mu_h|(\mathcal{S})\|_2 = \|\int_{s \in \mathcal{S}} |\mathrm{d}\mu_h(s)|\|_2 \le \sqrt{d}$.

Formally, we make the following assumption on the structure of $\mathcal{M}^{\mathsf{sim}}$ and $\mathcal{M}^{\mathsf{real}}$.

**Assumption 2.** *Both $\mathcal{M}^{\mathsf{sim}}$ and $\mathcal{M}^{\mathsf{real}}$ satisfy Definition 3.1 with feature maps and measures $(\phi^{\mathsf{s}}, \mu^{\mathsf{s}})$ and $(\phi^{\mathsf{r}}, \mu^{\mathsf{r}})$, respectively. Furthermore, $\phi^{\mathsf{s}}$ is* known*, but all of $\mu^{\mathsf{s}}, \phi^{\mathsf{r}}$, and $\mu^{\mathsf{r}}$ are unknown.*

In the literature, MDPs satisfying Definition 3.1 but where $\phi$ is known are typically referred to as "linear" MDPs, while MDPs satisfying Definition 3.1 but with $\phi$ unknown are typically referred to as "low-rank" MDPs. Given this terminology, we have that $\mathcal{M}^{\mathsf{sim}}$ is a linear MDP[2], while $\mathcal{M}^{\mathsf{real}}$ is a low-rank MDP. We assume the following reachability condition on $\mathcal{M}^{\mathsf{sim}}$.

**Assumption 3.** *There $\exists \lambda_{\min}^\star > 0$ with $\min_h \sup_\pi \lambda_{\min}(\mathbb{E}^{\mathcal{M}^{\mathsf{sim}},\pi}[\phi^{\mathsf{s}}(s_h,a_h)\phi^{\mathsf{s}}(s_h,a_h)^\top]) \ge \lambda_{\min}^\star$.*

Assumption 3 posits that each direction in the feature space in our simulator can be activated by some policy, and can be thought of as a measure of how easily each direction can be reached. Similar assumptions have appeared before in the literature on linear and low-rank MDPs [71, 3, 2]. Note that we only require this reachability assumption in $\mathcal{M}^{\mathsf{sim}}$. We also assume we are given access to function classes $\mathcal{F}_h : \mathcal{S} \times \mathcal{A} \to [0, H]$ and let $\mathcal{F} := \mathcal{F}_1 \times \mathcal{F}_2 \times \ldots \times \mathcal{F}_H$. Since no reward is collected in the $(H+1)$th step we take $f_{H+1} = 0$. For any $f : \mathcal{S} \times \mathcal{A} \to \mathbb{R}$, we let $\pi_h^f(s) := \arg\max_{a \in \mathcal{A}} f_h(s,a)$. We define the *Bellman operator* on some function $f_{h+1} : \mathcal{S} \times \mathcal{A} \to \mathbb{R}$ as:
$$\mathcal{T}f_{h+1}(s,a) := r_h(s,a) + \mathbb{E}_{s' \sim P_h(\cdot|s,a)}[\max_{a'} f_{h+1}(s',a')].$$

We make the following standard assumption on $\mathcal{F}$.

**Assumption 4** (Bellman Completeness). *For all $f_{h+1} \in \mathcal{F}_{h+1}$, we have $\mathcal{T}^{\mathsf{real}}f_{h+1}, \mathcal{T}^{\mathsf{sim}}f_{h+1} \in \mathcal{F}_h$, where $\mathcal{T}^{\mathsf{real}}$ and $\mathcal{T}^{\mathsf{sim}}$ denote the Bellman operators on $\mathcal{M}^{\mathsf{real}}$ and $\mathcal{M}^{\mathsf{sim}}$, respectively.*

**PAC Reinforcement Learning.** Our goal is to find a policy $\hat{\pi}$ that achieves maximum reward in $\mathcal{M}^{\mathsf{real}}$. Formally, we consider the PAC (Probably-Approximately-Correct) RL setting.

**Definition 3.2** (PAC Reinforcement Learning). *Given some $\epsilon > 0$ and $\delta > 0$, with probability at least $1 - \delta$ identify some policy $\hat{\pi}$ such that: $V_0^{\mathsf{real},\hat{\pi}} \ge V_0^{\mathsf{real},\star} - \epsilon$.*

We will be particularly interested in solving the PAC RL problem with the aid of a simulator, using the minimum number of samples from $\mathcal{M}^{\mathsf{real}}$ possible, as we will formalize in the following. As we will see, while it is straightforward to achieve this objective using $\mathcal{M}^{\mathsf{sim}}$ if $\epsilon = \mathcal{O}(\epsilon_{\mathrm{sim}})$, naive transfer methods can fail to achieve this completely if $\epsilon \ll \epsilon_{\mathrm{sim}}$. As such, our primary focus will be on developing efficient sim2real methods in this regime.

---

[2]The assumption that $\phi^{\mathsf{s}}$ is known is for simplicity only—similar results could be obtained were $\phi^{\mathsf{s}}$ also unknown using more complex algorithmic tools in $\mathcal{M}^{\mathsf{sim}}$.

# 4 Theoretical Results

In this section we provide our main theoretical results. We first present two negative results: in Section 4.1 showing that "naive exploration"—utilizing only a least-squares regression oracle and random exploration approaches such as $\zeta$-greedy[3]—is provably inefficient, and in Section 4.2 showing that directly transferring the optimal policy from $\mathcal{M}^{\text{sim}}$ to $\mathcal{M}^{\text{real}}$ is unable to efficiently obtain a policy with suboptimality better than $\mathcal{O}(\epsilon_{\text{sim}})$ in real. Then in Section 4.3 we present our main positive result, showing that by utilizing the same oracles as in Sections 4.1 and 4.2—a least-squares regression oracle, simulator access, and the ability to take actions randomly—we *can* efficiently learn an $\epsilon$-optimal policy for $\epsilon \ll \epsilon_{\text{sim}}$ in $\mathcal{M}^{\text{real}}$ by carefully utilizing the simulator to learn exploration policies.

## 4.1 Naive Exploration is Provably Inefficient

While a variety of works have developed provably efficient methods for solving PAC RL in low-rank MDPs [1, 62, 43, 41], these works typically either rely on complex computation oracles or carefully directed exploration strategies which are rarely utilized in practice. In contrast, RL methods utilized in practice typically rely on "simple" computation oracles and exploration strategies. Before considering the sim2real setting, we first show that such "simple" strategies are insufficient for efficient PAC RL. To instantiate such strategies, we consider a least-squares regression oracle, often available in practice.

**Oracle 4.1** (Least-Squares Regression Oracle). We assume access to a least-squares regression oracle such that, for any $h$ and dataset $\mathfrak{D} = \{(s^t, a^t, y^t)\}_{t=1}^T$, we can compute $\arg\min_{f \in \mathcal{F}_h} \sum_{t=1}^T (f(s^t, a^t) - y^t)^2$.

We couple this oracle with "naive exploration", which here we use to refer to any method that explores by randomly perturbing the action recommended by the current estimate of the optimal policy. While a variety of instantiations of naive exploration exist (see e.g. [11]), we consider a particularly common formulation, $\zeta$-greedy exploration.

**Protocol 4.1** ($\zeta$-Greedy Exploration). Given access to a regression oracle, any $\zeta \in [0, 1]$, and time horizon $T$, consider the following protocol:

1. Interact with $\mathcal{M}^{\text{real}}$ for $T$ episodes. At every step of episode $t + 1$, play $\pi_h^{f^t}(s)$ with probability $1 - \zeta$, and $a \sim \text{unif}(\mathcal{A})$ otherwise, where:
$$f_h^t = \arg\min_{f \in \mathcal{F}_h} \sum_{t'=1}^t (f(s_h^{t'}, a_h^{t'}) - r_h^{t'} - \max_{a'} f_{h+1}^t(s_{h+1}^{t'}, a'))^2.$$

2. Using collected data in any way desired, propose a policy $\widehat{\pi}$.

Protocol 4.1 forms the backbone of many algorithms used in practice. Despite its common application, as existing work [11] and the following result show, it is provably inefficient.

**Proposition 1.** *For any $H > 1$, $\zeta \in [0, 1]$, and $c \leq 1/6$, there exist some $\mathcal{M}^{\text{real},1}$ and $\mathcal{M}^{\text{real},2}$ such that both $\mathcal{M}^{\text{real},1}$ and $\mathcal{M}^{\text{real},2}$ satisfy Assumptions 2 and 4, and unless $T \geq \Omega(2^{H/2})$, when running Protocol 4.1 we have:*
$$\sup_{\mathcal{M}^{\text{real}} \in \{\mathcal{M}^{\text{real},1}, \mathcal{M}^{\text{real},2}\}} \mathbb{E}^{\mathcal{M}^{\text{real}}}[V_0^{\mathcal{M}^{\text{real}},\star} - V_0^{\mathcal{M}^{\text{real}},\widehat{\pi}}] \geq c/32.$$

Proposition 1 shows that, in a minimax sense, $\zeta$-greedy exploration is insufficient for provably efficient reinforcement learning: on one of $\mathcal{M}^{\text{real},1}$ and $\mathcal{M}^{\text{real},2}$, $\zeta$-greedy exploration will only be able to find a policy that is suboptimal by a constant factor, unless we take an exponentially large number of samples. While we focus on $\zeta$-greedy exploration in Proposition 1, this result extends to other types of naive exploration, for example, those given in [11]. See Section 5.2 for further discussion of the construction for Proposition 1.

## 4.2 Understanding the Limits of Direct sim2real Transfer

Proposition 1 shows that in general utilizing a least-squares regression oracle with $\zeta$-greedy exploration is insufficient for provably efficient RL. Can this be made efficient with access to a simulator

---

[3]Throughout this paper, we use "$\zeta$-greedy" to refer to the method more commonly known as "$\epsilon$-greedy" in the literature, to avoid ambiguity between this $\epsilon$ and the $\epsilon$ in our definition of PAC RL, Definition 3.2.

$\mathcal{M}^{\mathsf{sim}}$? In practice, standard sim2real methodology typically trains a policy to accomplish the goal task in $\mathcal{M}^{\mathsf{sim}}$, and then transfers this policy to $\mathcal{M}^{\mathsf{real}}$. We refer to this methodology as *direct* sim2real *transfer*. The following canonical result, usually referred to as the "simulation lemma" [24, 25, 6, 22], provides a sufficient guarantee for direct sim2real transfer to succeed under Assumption 1.

**Proposition 2** (Simulation Lemma). *Let $\pi^{\mathsf{sim},\star}$ denote an optimal policy in $\mathcal{M}^{\mathsf{sim}}$. Then under Assumption 1 we have $V_0^{\mathsf{real},\pi^{\mathsf{sim},\star}} \geq V_0^{\mathsf{real},\star} - 2H^2\epsilon_{\mathsf{sim}}$.*

Proposition 2 shows that, as long as $\epsilon \geq 2H^2\epsilon_{\mathsf{sim}}$, direct sim2real transfer succeeds in obtaining an $\epsilon$-optimal policy in $\mathcal{M}^{\mathsf{real}}$. While this justifies direct sim2real transfer in settings where $\mathcal{M}^{\mathsf{sim}}$ and $\mathcal{M}^{\mathsf{real}}$ are sufficiently close, we next show that given access only to $\pi^{\mathsf{sim},\star}$ and a least-squares regression oracle—even when coupled with random exploration—we cannot hope to efficiently obtain a policy with suboptimality less than $\mathcal{O}(\epsilon_{\mathsf{sim}})$ on $\mathcal{M}^{\mathsf{real}}$ using naive exploration. To formalize this, we consider the following interaction protocol.

**Protocol 4.2** (Direct sim2real Transfer with Naive Exploration). Given access to $\pi^{\mathsf{sim},\star}$, an optimal policy in $\mathcal{M}^{\mathsf{sim}}$, any $\zeta \in [0,1]$, and time horizon $T$, consider the following protocol:

1. Interact with $\mathcal{M}^{\mathsf{real}}$ for $T$ episodes, and at each step $h$ and state $s$ play $\pi_h^{\mathsf{sim},\star}(\cdot \mid s)$ with probability $1 - \zeta$, and $a \sim \mathrm{unif}(\mathcal{A})$ with probability $\zeta$.

2. Using collected data in any way desired, propose a policy $\widehat{\pi}$.

Protocol 4.2 is a standard instantiation of direct sim2real transfer commonly found in the literature, and couples playing the optimal policy from $\mathcal{M}^{\mathsf{sim}}$ with naive exploration. We have the following.

**Proposition 3.** *With the same choice of $\mathcal{M}^{\mathsf{real},1}$ and $\mathcal{M}^{\mathsf{real},2}$ as in Proposition 1, there exists some $\mathcal{M}^{\mathsf{sim}}$ such that both $\mathcal{M}^{\mathsf{real},1}$ and $\mathcal{M}^{\mathsf{real},2}$ satisfy Assumption 1 with $\mathcal{M}^{\mathsf{sim}}$ for $\epsilon_{\mathsf{sim}} \leftarrow c$, Assumptions 2 to 4 hold, and unless $T \geq \Omega(2^H)$ when running Protocol 4.2, we have:*

$$\sup_{\mathcal{M}^{\mathsf{real}} \in \{\mathcal{M}^{\mathsf{real},1}, \mathcal{M}^{\mathsf{real},2}\}} \mathbb{E}^{\mathcal{M}^{\mathsf{real}}}[V_0^{\mathcal{M}^{\mathsf{real}},\star} - V_0^{\mathcal{M}^{\mathsf{real}},\widehat{\pi}}] \geq \epsilon_{\mathsf{sim}}/32.$$

Proposition 3 shows that there exists a setting where there are two possible $\mathcal{M}^{\mathsf{real}}$ satisfying Assumption 1 with $\mathcal{M}^{\mathsf{sim}}$, and where, using direct policy transfer, unless we interact with $\mathcal{M}^{\mathsf{real}}$ for exponentially many episodes (in $H$), we cannot determine a better than $\Omega(\epsilon_{\mathsf{sim}})$-optimal policy for the worst-case $\mathcal{M}^{\mathsf{real}}$. Together, Propositions 2 and 3 show that, while we can utilize direct sim2real transfer to learn a policy that is $\mathcal{O}(\epsilon_{\mathsf{sim}})$-optimal in $\mathcal{M}^{\mathsf{real}}$, if our goal is to learn an $\epsilon$-optimal policy for $\epsilon \ll \epsilon_{\mathsf{sim}}$, direct sim2real transfer is unable to efficiently achieve this.

### 4.3  Efficient sim2real Transfer via Exploration Policy Transfer

Does there exist *some* way to utilize $\mathcal{M}^{\mathsf{sim}}$ and a least-squares regression oracle to enable efficient learning in $\mathcal{M}^{\mathsf{real}}$, even when $\epsilon \ll \epsilon_{\mathsf{sim}}$? Our key insight is that, rather than transferring the policy that optimally solves the task in $\mathcal{M}^{\mathsf{sim}}$, we should instead transfer policies that *explore* effectively in $\mathcal{M}^{\mathsf{sim}}$. While learning to solve a task may require very precise actions, we can often obtain sufficiently rich data with relatively imprecise actions—it is easier to learn to explore than learn to solve a task. In such settings, directly transferring a policy to solve the task will likely fail due to imprecision in the simulator, but it may be possible to still transfer a policy that generates exploratory data. To formalize this, we consider the following access model to $\mathcal{M}^{\mathsf{sim}}$.

**Oracle 4.2** ($\mathcal{M}^{\mathsf{sim}}$ Access). We may interact with $\mathcal{M}^{\mathsf{sim}}$ by either:

1. **(Trajectory Sampling)** For any policy $\pi$, sampling a trajectory $\{(s_h, a_h, r_h, s_{h+1})\}_{h=1}^H$ generated by playing $\pi$ on $\mathcal{M}^{\mathsf{sim}}$.

2. **(Policy Optimization)** For any reward $\widetilde{r}$, computing a policy $\pi^{\mathsf{sim}}(\widetilde{r})$ maximizing $\widetilde{r}$ on $\mathcal{M}^{\mathsf{sim}}$.

While access to such a policy optimization oracle is unrealistic in $\mathcal{M}^{\mathsf{real}}$, where we want to minimize the number of samples collected, given cheap access to samples in $\mathcal{M}^{\mathsf{sim}}$, such an oracle can often be (approximately) implemented in practice[4]. Note that under Oracle 4.2 we only assume *black-box*

---

[4] While for simplicity we assume that the truly optimal policy can be computed, our results easily extend to settings where we only have access to an oracle which can compute an approximately optimal policy.

access to our simulator—rather than allowing the behavior of the simulator to be queried at arbitrary states, we are simply allowed to roll out policies on $\mathcal{M}^{\mathsf{sim}}$, and compute optimal policies. Given Oracle 4.2, as well as our least-squares regression oracle, Oracle 4.1, we propose the following algorithm.

---

**Algorithm 1** sim2real Exploration Policy Transfer

---

1: **input:** budget $T$, confidence $\delta$, simulator $\mathcal{M}^{\mathsf{sim}}$
   `// Learn policies` $\Pi_{\mathsf{exp}}^h$ `which cover feature space in` $\mathcal{M}^{\mathsf{sim}}$
2: $\Pi_{\mathsf{exp}}^h \leftarrow \text{LEARNEXPPOLICIES}(\mathcal{M}^{\mathsf{sim}}, \delta, \frac{4A^3\epsilon}{H}, h)$ (Algorithm 5) for all $h \in [H]$
3: $\widetilde{\Pi}_{\mathsf{exp}}^h \leftarrow \{\widetilde{\pi}_{\mathsf{exp}} : \widetilde{\pi}_{\mathsf{exp}} \text{ plays } \pi_{\mathsf{exp}} \text{ up to step } h, \text{ then plays actions randomly}, \forall \pi_{\mathsf{exp}} \in \Pi_{\mathsf{exp}}^h\}$
   `// Explore in` $\mathcal{M}^{\mathsf{real}}$ `via` $\widetilde{\Pi}_{\mathsf{exp}}$
4: Play $\pi_{\mathsf{exp}} \sim \text{unif}(\{\text{unif}(\widetilde{\Pi}_{\mathsf{exp}}^h)\}_{h=1}^H)$ for $T/2$ episodes in $\mathcal{M}^{\mathsf{real}}$, add data to $\mathfrak{D}$
   `// Estimate optimal policy on collected data`
5: **for** $h = H, H-1, \ldots, 1$ **do**
6:     $\widehat{f}_h \leftarrow \arg\min_{f\in\mathcal{F}} \sum_{(s,a,r,s')\in\mathfrak{D}}(f_h(s,a) - r - \max_{a'}\widehat{f}_{h+1}(s',a'))^2$
7: Compute $\pi^{\mathsf{sim},\star}$ via Oracle 4.2
8: Play $\pi^{\mathsf{sim},\star}$ for $T/4$ episodes in real, compute average return $\widehat{V}_0^{\mathsf{real},\pi^{\mathsf{sim},\star}}$
9: Play $\pi^{\widehat{f}}$ for $T/4$ episodes in real, compute average return $\widehat{V}_0^{\mathsf{real},\pi^{\widehat{f}}}$
10: **return** $\widehat{\pi} \leftarrow \arg\max_{\pi\in\{\pi^{\widehat{f}},\pi^{\mathsf{sim},\star}\}} \widehat{V}_0^{\mathsf{real},\pi}$

---

Algorithm 1 first calls a subroutine LEARNEXPPOLICIES, which learns a set of policies that provide rich data coverage on $\mathcal{M}^{\mathsf{sim}}$—precisely, LEARNEXPPOLICIES returns policies $\{\Pi_{\mathsf{exp}}^h\}_{h\in[H]}$ which induce covariates with lower-bounded minimum eigenvalue on $\mathcal{M}^{\mathsf{sim}}$ and relies only on Oracle 4.2 (as well as knowledge of the featurization of $\mathcal{M}^{\mathsf{sim}}$, $\phi^{\mathsf{s}}$) to find such policies. Algorithm 1 then plays these exploration policies in $\mathcal{M}^{\mathsf{real}}$, coupled with random exploration, and applies the regression oracle to the data they collect. Finally, it estimates the value of the policy learned by the regression oracle and $\pi^{\mathsf{sim},\star}$, and returns whichever is best. We have the following.

**Theorem 1.** *If Assumptions 1 to 4 hold and*

$$\epsilon_{\mathsf{sim}} \leq \frac{\lambda_{\min}^\star}{64dHA^3}, \tag{4.1}$$

*then as long as*

$$T \geq c \cdot \frac{d^2 H^{16}}{\epsilon^8} \cdot \log\frac{H|\mathcal{F}|}{\delta},$$

*with probability at least $1 - \delta$, Algorithm 1 returns a policy $\widehat{\pi}$ such that $V_0^{\mathsf{real},\star} - V_0^{\mathsf{real},\widehat{\pi}} \leq \epsilon$, and Oracles 4.1 and 4.2 are invoked at most $\text{poly}(d, H, \epsilon^{-1}, \log\frac{1}{\delta})$ times.*

Theorem 1 shows that, as long as $\epsilon_{\mathsf{sim}}$ satisfies (4.1), utilizing a simulator and least-squares regression oracle, Oracles 4.1 and 4.2, allows for efficient learning in $\mathcal{M}^{\mathsf{real}}$, achieving a complexity scaling polynomially in problem parameters. This yields an *exponential improvement* over learning without a simulator using naive exploration or direct sim2real transfer—which Propositions 1 and 3 show have complexity scaling exponentially in the horizon—despite utilizing the same practical computation oracles. To the best of our knowledge, this result provides the first theoretical evidence that sim2real transfer can yield provable gains in RL beyond trivial settings where direct transfer succeeds.

Note that the condition in (4.1) is independent of $\epsilon$—unlike direct sim2real transfer, which requires $\epsilon = \mathcal{O}(\epsilon_{\mathsf{sim}})$, we simply must assume $\epsilon_{\mathsf{sim}}$ is small enough that (4.1) holds, and Theorem 1 shows that we can efficiently learn an $\epsilon$-optimal policy in $\mathcal{M}^{\mathsf{real}}$ for any $\epsilon > 0$. In Appendix B.4, we also present an extended version of Theorem 1, Theorem 3, which utilizes data from $\mathcal{M}^{\mathsf{sim}}$ to reduce the dependence on $\log|\mathcal{F}|$. In particular, instead of scaling with $\log|\mathcal{F}|$, it only scales with the log-cardinality of functions that are (approximately) Bellman-consistent on $\mathcal{M}^{\mathsf{sim}}$. To illustrate the effectiveness of Theorem 1, we return to the instance of Propositions 1 and 3, where naive exploration and direct sim2real transfer fails. We have the following.

**Proposition 4.** *In the setting of Propositions 1 and 3 and assuming that $\epsilon_{\mathsf{sim}} \leq \frac{1}{8192} \cdot \frac{1}{H}$, running Algorithm 1 will require $\text{poly}(H, \epsilon^{-1}) \cdot \log\frac{1}{\delta}$ samples from $\mathcal{M}^{\mathsf{real}}$ in order to identify an $\epsilon$-optimal policy in $\mathcal{M}^{\mathsf{real}}$ with probability at least $1 - \delta$, for any $\epsilon > 0$.*

Note that the condition required by Proposition 4 is simply that $\epsilon_{\text{sim}} \lesssim 1/H$—as long as our simulator satisfies this condition, we can efficiently transfer exploration policies to learn an $\epsilon$-optimal policy, for any $\epsilon > 0$, while naive methods would be limited to only obtaining an $\Omega(1/H)$-optimal policy.

**Remark 4.1** (Necessity of Random Exploration). Algorithm 1 achieves efficient exploration in $\mathcal{M}^{\text{real}}$ by learning policies $\Pi^h_{\text{exp}}$ in $\mathcal{M}^{\text{sim}}$ that span the feature space of $\mathcal{M}^{\text{sim}}$ (Line 2), and then playing these policies in $\mathcal{M}^{\text{real}}$, coupled with random exploration (Line 4). This use of random exploration is critical to obtaining Theorem 1. As we show in Proposition 5, if we omit the random exploration, Assumption 1 is not sufficient to guarantee $\Pi^h_{\text{exp}}$ explores effectively in $\mathcal{M}^{\text{real}}$, even when (4.1) holds.

**Remark 4.2** (Computational Efficiency). Algorithm 1, as well as its main subroutine LEARNEXP-POLICIES, relies only on calls to Oracle 4.1 and Oracle 4.2. Thus, assuming we can efficiently implement these oracles, which is often the case in problem settings of interest, Algorithm 1 can be run in a computationally efficient manner.

## 5 Practical Algorithm and Experiments

We next validate the effectiveness of our proposal in practice: can a set of diverse exploration policies obtained from simulation improve the efficiency of real-world reinforcement learning? We start by showing that this holds for a simple, didactic, tabular environment in Section 5.2. From here, we consider several more realistic task domains: simulators inspired by real-world robotic manipulation tasks (sim2sim transfer, Section 5.3); and an actual real-world sim2real experiment on a Franka robotic platform (sim2real transfer, Section 5.4). Further details on all experiments, including additional baselines, can be found in Appendix E. Before stating our experimental results, we first provide a practical instantiation of Algorithm 1 that we can apply with real robotic systems and neural network function approximators.

### 5.1 Practical Instantiation of Exploration Policy Transfer

The key idea behind Algorithm 1 is quite simple: learn a set of exploratory policies in $\mathcal{M}^{\text{sim}}$—policies which provide rich data coverage in $\mathcal{M}^{\text{sim}}$—and transfer these policies to $\mathcal{M}^{\text{real}}$, coupled with random exploration, using the collected data to determine a near-optimal policy for $\mathcal{M}^{\text{real}}$. Algorithm 1 provides a particular instantiation of this principle, learning exploratory policies in $\mathcal{M}^{\text{sim}}$ via the LEARNEXPPOLICIES subroutine, which aims to cover the feature space of $\mathcal{M}^{\text{sim}}$, and utilizing a least-squares regression oracle to compute an optimal policy given the data collected in $\mathcal{M}^{\text{real}}$. In practice, however, other instantiations of this principle are possible by replacing LEARNEXPPOLICIES with any procedure which generates exploratory policies in $\mathcal{M}^{\text{sim}}$, and replacing the regression oracle with any RL algorithm able to learn from off-policy data. We consider a general meta-algorithm instantiating this in Algorithm 2.

---

**Algorithm 2** Practical sim2real Exploration Policy Transfer Meta-Algorithm

1: **Input:** Simulator $\mathcal{M}^{\text{sim}}$, real environment $\mathcal{M}^{\text{real}}$, simulator budget $T_{\text{sim}}$, real budget $T$, algorithm to generate exploratory policies in sim $\mathfrak{A}_{\text{exp}}$, algorithm to solve policy optimization in real $\mathfrak{A}_{\text{po}}$
    // Learn exploratory policies in $\mathcal{M}^{\text{sim}}$
2: Run $\mathfrak{A}_{\text{exp}}$ for $T_{\text{sim}}$ steps in $\mathcal{M}^{\text{sim}}$ to generate set of exploratory policies $\Pi_{\text{exp}}$
    // Deploy exploratory policies in $\mathcal{M}^{\text{real}}$
3: **for** $t = 1, 2, \ldots, T/2$ **do**
4:     Draw $\pi_{\text{exp}} \sim \text{unif}(\Pi_{\text{exp}})$, play in $\mathcal{M}^{\text{real}}$ for one episode, add data to replay buffer of $\mathfrak{A}_{\text{po}}$
5:     Run $\mathfrak{A}_{\text{po}}$ for one episode         // optional if $\mathfrak{A}_{\text{po}}$ learns fully offline

---

In practice, $\mathfrak{A}_{\text{exp}}$ and $\mathfrak{A}_{\text{po}}$ can be instantiated with a variety of algorithms. For example, we might take $\mathfrak{A}_{\text{exp}}$ to be an RND [7] or bootstrapped Q-learning-style [45, 31] algorithm, or any unsupervised RL procedure [48, 14, 32, 47], and $\mathfrak{A}_{\text{po}}$ to be an off-policy policy optimization algorithm such as soft actor-critic (SAC) [16] or implicit $Q$-learning (IQL) [28].

For the following experiments, we instantiate Algorithm 2 by setting $\mathfrak{A}_{\text{exp}}$ to an algorithm inspired by recent work on inducing diverse behaviors in RL [14, 30], and $\mathfrak{A}_{\text{po}}$ to SAC. In particular, $\mathfrak{A}_{\text{exp}}$ simultaneously trains an ensemble of policies $\Pi_{\text{exp}} = \{\pi^i_{\text{exp}}\}^n_{i=1}$ and a discriminator $d_\theta : \mathcal{S} \times [n] \to \mathbb{R}$, where $d_\theta$ is trained to discriminate between the behaviors of each policy $\pi^i_{\text{exp}}$, and $\pi^i_{\text{exp}}$

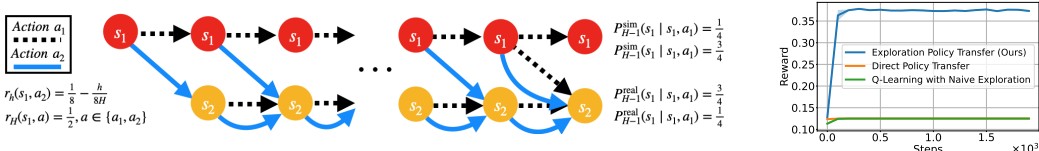

Figure 2: **Left:** Illustration ofCombination Lock Example. **Right:** Results on Combination Lock.

is optimized on a weighting of the true task reward and the exploration reward induced by the discriminator, $r_e(s, i) := \log \frac{\exp(d_\theta(s,i))}{\sum_{j \in [n]} \exp(d_\theta(s,j))}$. As shown in existing work [14, 30], this simple training objective effectively induces diverse behavior with temporally correlated exploration while remaining within the vicinity of the optimal policy, using standard optimization techniques. Note that the particular choice of algorithm is less critical here than abiding by the recipes laid out in the meta-algorithm (Algorithm 2). The particular instantiation that we run for our experiments is detailed in Algorithm 6, along with further details in Appendix E.2.

## 5.2 Didactic Combination Lock Experiment

We first consider a variant of the construction used to prove Propositions 1 and 3, itself a variant of the classic combination lock instance. We illustrate this instance in Figure 2. Unless noted, all transitions occur with probability 1, and rewards are 0. Here, in $\mathcal{M}^{\text{sim}}$ the optimal policy, $\pi^{\text{sim},\star}$, plays action $a_2$ for all steps $h < H - 1$, while in $\mathcal{M}^{\text{real}}$, the optimal policy plays action $a_1$ at every step. Which policy is optimal is determined by the outgoing transition from $s_1$ at the $(H - 1)$th step and, as such, to identify the optimal policy, any algorithm must reach $s_1$ at the $(H - 1)$th step. As $s_1$ will only be reached at step $H - 1$ by playing $a_1$ for $H - 1$ consecutive times, any algorithm relying on naive exploration will take exponentially long to identify the optimal policy. Furthermore, playing $\pi^{\text{sim},\star}$ coupled with random exploration will similarly take an exponential number of episodes, since $\pi^{\text{sim},\star}$ always plays $a_2$. As such, both direct sim2real policy transfer as well as $Q$-learning with naive exploration (Protocol 4.1) will fail to find the optimal policy in $\mathcal{M}^{\text{real}}$. However, if we transfer exploratory policies from $\mathcal{M}^{\text{sim}}$, since $\mathcal{M}^{\text{sim}}$ and $\mathcal{M}^{\text{real}}$ behave identically up to step $H - 1$, these policies can efficiently traverse $\mathcal{M}^{\text{real}}$, reach $s_1$ at step $H - 1$, and identify the optimal policy. We compare our approach of exploration policy transfer to these baselines methods and illustrate the performance of each in Figure 2. As this is a simple tabular instance, we implement Algorithm 1 directly here. As Figure 2 shows, the intuition described above leads to real gains in practice—exploration policy transfer quickly identifies the optimal policy, while more naive approach fail completely over the time horizon we considered.

## 5.3 Realistic Robotics sim2sim Experiment

To test the ability of our proposed method to scale to more complex problems, we next experiment on a sim2sim transfer setting with a realistic robotic simulator. We consider TychoEnv, a simulator of the 7DOF Tycho robotics platform introduced by [73], and shown in Figure 3. We test sim2sim transfer on a reaching task where the goal is to touch a small ball hanging in the air with the tip of the chopstick end effector. The agent perceives the ball and its own end effector pose and outputs a delta in its desired end effector pose as a command. We set $\mathcal{M}^{\text{sim}}$ and $\mathcal{M}^{\text{real}}$ to be two instances of TychoEnv with slightly different parameters to model real-world sim2real transfer. Precisely, we change the action bounds and control frequency from $\mathcal{M}^{\text{sim}}$ to $\mathcal{M}^{\text{real}}$.

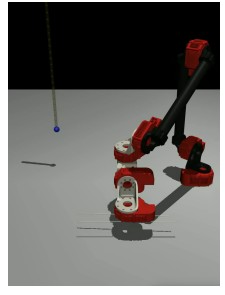

Figure 3: TychoEnv Reach Task Setup

We aim to compare our approach of exploration policy transfer with direct sim2real policy transfer. To this end, we first train a policy in $\mathcal{M}^{\text{sim}}$ that solves the task in $\mathcal{M}^{\text{sim}}$, $\pi^{\text{sim},\star}$, and then utilize this policy in place of $\Pi_{\exp}$ in Algorithm 2. We instantiate our approach of exploration policy transfer as outlined above. Our aim in this experiment is to illustrate how the quality of the data provided by direct policy transfer vs. exploration policy transfer affects learning. As such, for both approaches we simply initialize our SAC agent in $\mathcal{M}^{\text{real}}$, $\mathfrak{A}_{\text{po}}$, from scratch, and set the reward in $\mathcal{M}^{\text{real}}$ to be sparse: the agent only receives a non-zero reward if it successfully touches the ball. For each approach, we repeat the process of training in $\mathcal{M}^{\text{sim}}$ four times, and for each of these run them for two trials in $\mathcal{M}^{\text{real}}$.

We illustrate our results in Figure 4. As this figure illustrates, direct policy transfer fails to learn completely, while exploration policy transfer successfully solves the task. Investigating the behavior of each method, we find that the policies transferred via exploration policy transfer, while failing to solve the task with perfect accuracy, when coupled with naive exploration are able to successfully make contact with the ball on occasion. This provides sufficiently rich data for SAC to ultimately learn to solve the task. In contrast, direct policy transfer fails to collect any reward when run in $\mathcal{M}^{\text{real}}$, and, given the sparse reward nature of the task, SAC is unable to locate any reward and learn. We include an additional sim2sim experiment on the Franka Emika Panda Robot Arm in Appendix E.4.

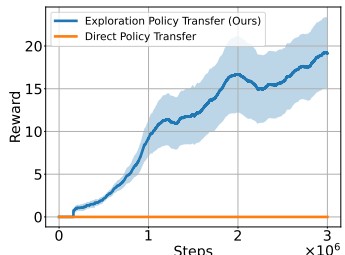

Figure 4: Results on sim2sim Transfer in TychoEnv Simulator

## 5.4 Real-World Robotic sim2real Experiment

Finally, we demonstrate our algorithm for actual sim2real policy transfer for a manipulation task on a real-world Franka Emika Panda robot arm [17] with a parallel gripper. Our task is to push a 75mm diameter cylindrical "puck" from the center to the edge of the surface, as shown in Figure 1, with the arm initialized at random locations. The observed state $s = [\mathbf{p}_{\text{ee}}, \mathbf{p}_{\text{obj}}] \in \mathbb{R}^4$ consists of the planar Cartesian coordinate of the end effector $\mathbf{p}_{\text{ee}}$ along with the center of mass of the puck $\mathbf{p}_{\text{obj}}$. Our policy outputs planar end effector position deltas $a = \Delta \mathbf{p}_{\text{ee}} \in \mathbb{R}^2$, evaluated at 8 Hz, which are passed into a lower-level joint position PID controller. We use an Intel Realsense D435 depth camera to track the location of the puck. Our reward function is a sum of a success indicator (indicating when the puck has been pushed to the edge of the surface) and terms which give negative reward if the distance from the end effector to the puck, or puck to the goal, are too large (see (E.1)); in particular, a reward greater than 0 indicates success.

We run the instantiation of Algorithm 2 outlined above. In particular, we train an ensemble of $n = 15$ exploration policies, training for 20 million steps in $\mathcal{M}^{\text{sim}}$. In addition, we train a policy that solves the task in $\mathcal{M}^{\text{sim}}$, $\pi^{\text{sim},\star}$. We use a custom simulator of the arm, where during training the friction of the table is randomized and noise is added to the observations.

We observe a substantial sim2real gap between our simulator and the real robot, with policies trained in simulation failing to complete the pushing task zero shot in real, even when trained with domain randomization. We compare direct sim2real policy transfer against our method of transferring exploration policies. For direct policy transfer, we simply run SAC to finetune $\pi^{\text{sim},\star}$ in the real world, using the current policy to collect data. For exploration policy transfer, we instead utilize $\Pi_{\text{exp}}$, our ensemble of exploration policies, to collect data in the real world. We run this in tandem with an SAC agent, feeding the data from the exploration policies into the SAC agent's replay buffer. Unlike in Section 5.3, rather than initializing the SAC policy from scratch, we set the initial policy as $\pi^{\text{sim},\star}$, and fine-tune from this on the data collected from playing $\Pi_{\text{exp}}$. See Appendix E.5 for additional details.

Our results are shown on the right side of Figure 1. Statistics are computed over 6 runs for each method. Direct policy transfer with finetuning is unable to solve the task in real in each of the 6 runs, and converges to a suboptimal solution. However, our method is able to solve the task successfully each time and achieve a substantially higher reward.

## 6 Discussion

In this work, we have demonstrated that simulators can make naive exploration efficient even in settings where direct sim2real transfer fails, if they are used to train a set of exploration policies. We highlight several limitations of this work, which we believe are interesting future research questions:

- Our focus is purely on dynamics shift—where the dynamics of sim and real differ, but the environments are otherwise the same. While dynamics shift is common in many scenarios, other types of shift can exist as well, for example perceptual shift. How can we best handle these types of shift?

- How can we utilize a simulator in sim2real transfer if we can reset it arbitrarily, rather than just allowing for black-box access? Does the ability to reset allow us to improve sample efficiency further?

- Is the reachability condition, Assumption 3, necessary for successful exploration transfer?

**Acknowledgements**

The work of AW and KJ was partially supported by the NSF through the University of Washington Materials Research Science and Engineering Center, DMR-2308979, and awards CCF 2007036 and CAREER 2141511. The work of LK was partially supported by Toyota Research Institute URP.

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

# A  Technical Results

We denote the state-visitations for some policy $\pi$ as $w_h^\pi(s,a) := \mathbb{P}^\pi[(s_h, a_h) = (s,a)]$, $w_h^\pi(\mathcal{Z}) := \mathbb{P}^\pi[(s_h, a_h) \in \mathcal{Z}]$, for $\mathcal{Z} \subseteq \mathcal{S} \times \mathcal{A}$. For $\mathcal{X} \subseteq \mathbb{R}^d$, we denote $w_h^\pi(\mathcal{X}) := \mathbb{P}^\pi[\phi(s_h, a_h) \in \mathcal{X}]$, for $\phi$ the featurization of the environment.

**Lemma A.1.** *Consider MDPs $M$ and $\widetilde{M}$ with transition kernels $P$ and $\widetilde{P}$. Assume that both $M$ and $\widetilde{M}$ start in the same state $s_0$ and that, for each $(s,a,h)$:*

$$\|P_h(\cdot \mid s,a) - \widetilde{P}_h(\cdot \mid s,a)\|_{\mathrm{TV}} \le \epsilon_{\mathrm{sim}}. \tag{A.1}$$

*Consider some reward function $r$ such that $\sum_{h=1}^H r_h(s_h, a_h) \le R$ for all possible sequences $\{(s_h, a_h)\}_{h=1}^H$. Then it follows that, for any $\pi$ and $(s,a,h)$,*

$$|Q_h^{M,\pi}(s,a) - Q_h^{\widetilde{M},\pi}(s,a)| \le HR \cdot \epsilon_{\mathrm{sim}}.$$

*Proof.* We prove this by induction. First, assume that for some $h$ and all $s, a$, we have $|Q_{h+1}^{M,\pi}(s,a) - Q_{h+1}^{\widetilde{M},\pi}(s,a)| \le \epsilon_{h+1}$. By definition we have

$$Q_h^{M,\pi}(s,a) = r_h(s,a) + \mathbb{E}^{M,\pi}[Q_{h+1}^{M,\pi}(s_{h+1}, a_{h+1}) \mid s_h = s, a_h = a]$$

and similarly for $Q_{h+1}^{\widetilde{M},\pi}(s,a)$. Thus:

$$|Q_h^{M,\pi}(s,a) - Q_h^{\widetilde{M},\pi}(s,a)|$$
$$\overset{(a)}{\le} |\mathbb{E}^{M,\pi}[Q_{h+1}^{M,\pi}(s_{h+1}, a_{h+1}) \mid s_h = s, a_h = a] - \mathbb{E}^{\widetilde{M},\pi}[Q_{h+1}^{M,\pi}(s_{h+1}, a_{h+1}) \mid s_h = s, a_h = a]|$$
$$\quad + \mathbb{E}^{\widetilde{M},\pi}[|Q_{h+1}^{M,\pi}(s_{h+1}, a_{h+1}) - Q_{h+1}^{\widetilde{M},\pi}(s_{h+1}, a_{h+1})| \mid s_h = s, a_h = a]$$
$$\overset{(b)}{\le} |\mathbb{E}^{M,\pi}[Q_{h+1}^{M,\pi}(s_{h+1}, a_{h+1}) \mid s_h = s, a_h = a] - \mathbb{E}^{\widetilde{M},\pi}[Q_{h+1}^{M,\pi}(s_{h+1}, a_{h+1}) \mid s_h = s, a_h = a]| + \epsilon_{h+1}$$

where $(a)$ follows from the triangle inequality and $(b)$ follows from the inductive hypothesis. Under (A.1), we can bound

$$|\mathbb{E}^{M,\pi}[Q_{h+1}^{M,\pi}(s_{h+1}, a_{h+1}) \mid s_h = s, a_h = a] - \mathbb{E}^{\widetilde{M},\pi}[Q_{h+1}^{M,\pi}(s_{h+1}, a_{h+1}) \mid s_h = s, a_h = a]| \le \epsilon_{\mathrm{sim}} \cdot R.$$

It follows that for any $(s,a)$, $|Q_h^{M,\pi}(s,a) - Q_h^{\widetilde{M},\pi}(s,a)| \le \epsilon_h =: \epsilon_{\mathrm{sim}} R + \epsilon_{h+1}$.

The base case follows trivially with $\epsilon_H = 0$ since for any MDP we have that $Q_H^{M,\pi}(s,a) = r_H(s,a) = Q_H^{\widetilde{M},\pi}(s,a)$. $\qquad\square$

**Lemma A.2.** *Under the same setting as Lemma A.1 and for any $h$, $\pi$, and $\mathcal{Z} \subseteq \mathcal{S} \times \mathcal{A}$, we have*

$$|w_h^{M,\pi}(\mathcal{Z}) - w_h^{\widetilde{M},\pi}(\mathcal{Z})| \le H\epsilon_{\mathrm{sim}}.$$

*Proof.* This is an immediate consequence of Lemma A.1 since, setting the reward $r_{h'}(s,a) = \mathbb{I}\{(s,a) \in \mathcal{Z}, h' = h\}$, we can set $R = 1$ and have $V_0^{M,\pi} = w_h^{M,\pi}(\mathcal{Z})$. $\qquad\square$

**Lemma A.3** (Proposition 2). *Under Assumption 1, we have that*

$$V_0^{\mathrm{real},\star} - V_0^{\mathrm{real},\pi^{\mathrm{sim},\star}} \le 2H^2 \epsilon_{\mathrm{sim}} \quad \text{and} \quad V_0^{\mathrm{sim},\star} - V_0^{\mathrm{sim},\pi^{\mathrm{real},\star}} \le 2H^2 \epsilon_{\mathrm{sim}}.$$

*Proof.* We prove the result for real—the result for sim follows analogously. We have

$$V_0^{\mathrm{real},\star} - V_0^{\mathrm{real},\pi^{\mathrm{sim},\star}} = V_0^{\mathrm{real},\pi^{\mathrm{real},\star}} - V_0^{\mathrm{sim},\pi^{\mathrm{real},\star}} + \underbrace{V_0^{\mathrm{sim},\pi^{\mathrm{real},\star}} - V_0^{\mathrm{sim},\pi^{\mathrm{sim},\star}}}_{\le 0} + V_0^{\mathrm{sim},\pi^{\mathrm{sim},\star}} - V_0^{\mathrm{real},\pi^{\mathrm{sim},\star}}$$
$$\le |V_0^{\mathrm{real},\pi^{\mathrm{real},\star}} - V_0^{\mathrm{sim},\pi^{\mathrm{real},\star}}| + |V_0^{\mathrm{sim},\pi^{\mathrm{sim},\star}} - V_0^{\mathrm{real},\pi^{\mathrm{sim},\star}}|.$$

The result then follows by applying Lemma A.1 to bound each of these terms by $H^2 \epsilon_{\mathrm{sim}}$. $\qquad\square$

**Lemma A.4.** *For any $f \in \mathcal{F}$,*

$$V_0^\star - V_0^{\pi^f} \le \max_{\pi \in \{\pi^f, \pi^\star\}} \sum_{h=0}^{H-1} 2 \left| \mathbb{E}^\pi [f_h(s_h, a_h) - \mathcal{T} f_{h+1}(s_h, a_h)] \right|.$$

*Proof.* We write

$$V_0^\star - V_0^{\pi^f} = \underbrace{V_0^\star - \max_a f_0(s_0, a)}_{(a)} + \underbrace{\max_a f_0(s_0, a) - V_0^{\pi^f}}_{(b)}$$

and then bound each of these terms separately. By Lemma 5 of [57] we have

$$(a) \le \sum_{h=0}^{H} \left| \mathbb{E}^{\pi^\star} [f_h(s_h, a_h) - r_h - \max_{a'} f_{h+1}(s_{h+1}, a')] \right|$$

$$= \sum_{h=0}^{H-1} \left| \mathbb{E}^{\pi^\star} [f_h(s_h, a_h) - \mathbb{E}[r_h + \max_{a'} f_{h+1}(s_{h+1}, a') \mid s_h, a_h]] \right|.$$

Similarly, by Lemma 4 of [57] we have

$$(b) \le \sum_{h=0}^{H-1} \left| \mathbb{E}^{\pi^f} [f_h(s_h, a_h) - r_h - \max_{a'} f_{h+1}(s_{h+1}, a')] \right|$$

$$= \sum_{h=0}^{H-1} \left| \mathbb{E}^{\pi^f} [f_h(s_h, a_h) - \mathbb{E}[r_h + \max_{a'} f_{h+1}(s_{h+1}, a') \mid s_h, a_h]] \right|.$$

$\square$

# B    Proof of Main Results

In Appendix B.1 we first provide a general result on learning in real when collecting data via a fixed set of exploration policies, given a particular coverage assumption. Then in Appendix B.2, we show that by playing a set of policies which induce full-rank covariates in sim, these policies provide sufficient coverage for learning in real. Finally in Appendices B.3 and B.4, we use these results to prove Theorems 1 and 3. Throughout the appendix we develop the supporting lemmas for our more general result, Theorem 3, which utilizes the simulator to restrict the version space (i.e. the dependence on $|\mathcal{F}|$) in addition to utilizing the simulator to aid in exploration.

Throughout this and the following section we assume that Assumption 4 holds. We also assume that $f_h \in [0, V_{\max}]$ instead of $f_h \in [0, H]$, for some $V_{\max} > 0$. For any $f \in \mathcal{F}$, we denote the Bellman residual as

$$\mathcal{E}_h(f)(s, a) := \mathcal{T} f_{h+1}(s, a) - f_h(s, a).$$

Note that by assumption on $\mathcal{F}$, we have $\mathcal{E}_h(f)(s, a) \in [-V_{\max}, V_{\max}]$.

For any policy $\pi$, we denote $\mathbf{\Lambda}_{\pi,h}^{\mathsf{s}} := \mathbb{E}^{\mathsf{sim}, \pi}[\phi^{\mathsf{s}}(s_h, a_h)\phi^{\mathsf{s}}(s_h, a_h)^\top]$ and $\mathbf{\Lambda}_{\pi,h}^{\mathsf{r}} := \mathbb{E}^{\mathsf{real}, \pi}[\phi^{\mathsf{r}}(s_h, a_h)\phi^{\mathsf{r}}(s_h, a_h)^\top]$.

**Necessity of Random Exploration.**    Algorithm 1 achieves efficient exploration in $\mathcal{M}^{\mathsf{real}}$ by first learning a set of policies $\Pi_{\mathsf{exp}}^h$ in $\mathcal{M}^{\mathsf{sim}}$ that span the feature space of $\mathcal{M}^{\mathsf{sim}}$ (Line 2), achieving

$$\lambda_{\min}\left( \tfrac{1}{|\Pi_{\mathsf{exp}}^h|} \sum_{\pi \in \Pi_{\mathsf{exp}}^h} \mathbb{E}^{\mathcal{M}^{\mathsf{sim}}, \pi}[\phi^{\mathsf{s}}(s_h, a_h)\phi^{\mathsf{s}}(s_h, a_h)^\top] \right) \gtrsim \lambda_{\min}^\star, \tag{B.1}$$

and then playing these policies in $\mathcal{M}^{\mathsf{real}}$, coupled with random exploration (Line 4). In particular, Algorithm 1 plays policies from $\widetilde{\Pi}_{\mathsf{exp}}^h$, where each $\widetilde{\pi}_{\mathsf{exp}} \in \widetilde{\Pi}_{\mathsf{exp}}^h$ is defined as the policy which plays some $\pi_{\mathsf{exp}} \in \Pi_{\mathsf{exp}}^h$ up to step $h$, and then for steps $h' = h + 1, \dots, H$ chooses actions uniformly at random. This use of random exploration is critical to obtaining Theorem 1. Indeed, under our

transfer model, condition (4.1) of Theorem 1 is not strong enough to ensure that policies satisfying (B.1) collect rich enough data in $\mathcal{M}^{\text{real}}$ to allow for learning a near-optimal policy. While (4.1) is sufficient to guarantee that playing $\Pi_{\exp}^h$ on $\mathcal{M}^{\text{real}}$ collects data which spans the feature space of $\mathcal{M}^{\text{sim}}$—that is, satisfying (B.1) but with the expectation over $\mathcal{M}^{\text{sim}}$ replaced by an expectation of $\mathcal{M}^{\text{real}}$— this is insufficient for learning, as the following result shows.

**Proposition 5.** *For any $\epsilon_{\text{sim}} \leq 1/2$, there exist some $\mathcal{M}^{\text{sim}}$, $\mathcal{M}^{\text{real},1}$, and $\mathcal{M}^{\text{real},2}$ such that:*

1. *Both $\mathcal{M}^{\text{real},1}$ and $\mathcal{M}^{\text{real},2}$ satisfy Assumption 1 with $\mathcal{M}^{\text{sim}}$ and Assumptions 2 to 4 hold.*

2. *There exists some policy $\pi_{\exp}$ such that $\lambda_{\min}(\mathbb{E}^{\mathcal{M}^{\text{sim}},\pi_{\exp}}[\phi^{\mathsf{s}}(s_h, a_h)\phi^{\mathsf{s}}(s_h, a_h)^\top]) = 1/2$, $\forall h \in [H]$, and for any $T \geq 0$, if we play $\pi_{\exp}$ on $\mathcal{M}^{\text{real}}$ for $T$ steps, we have:*

$$\inf_{\widehat{\pi}} \sup_{\mathcal{M}^{\text{real}} \in \{\mathcal{M}^{\text{real},1}, \mathcal{M}^{\text{real},2}\}} \mathbb{E}^{\mathcal{M}^{\text{real}},\pi_{\exp}}[V_0^{\mathcal{M}^{\text{real}},\star} - V_0^{\mathcal{M}^{\text{real}},\widehat{\pi}}] \geq \epsilon_{\text{sim}}.$$

Proposition 5 holds because two MDPs may be "close" in the sense of Assumption 1 but admit very different feature representations. As a result, transferring a policy that covers the feature space of $\mathcal{M}^{\text{sim}}$ is not necessarily sufficient for covering the feature space of $\mathcal{M}^{\text{real}}$, which ultimately means that data collected from $\pi_{\exp}$ is unable to identify the optimal policy in $\mathcal{M}^{\text{real}}$. Our key technical result, Lemma B.4, shows, however, that under Assumption 1 and (4.1), policies which achieve high coverage in $\mathcal{M}^{\text{sim}}$ (i.e. satisfy (B.1)) are able to reach within a *logarithmic* number of steps of relevant states in $\mathcal{M}^{\text{real}}$. While the sample complexity of random exploration typically scales exponentially in the horizon, if the horizon over which we must explore is only logarithmic, the total complexity is then only polynomial. Theorem 1 critically relies on these facts—by playing policies in $\Pi_{\exp}^h$ up to step $h$ and then exploring randomly, and repeating this for each $h \in [H]$, we show that sufficiently rich data is collected in $\mathcal{M}^{\text{real}}$ for learning an $\epsilon$-optimal policy.

## B.1 Learning in real with Fixed Exploration Policies

---
**Algorithm 3** sim2real transfer with fixed exploration policies (EXPLOREREAL)

---
1: **input:** exploration policies $\{\pi_{\exp}^h\}_{h=1}^H$, budget $T$, sim date $\mathfrak{D}_{\text{sim}}$, sim regularization $\gamma$
2: Play $\pi_{\exp} = \text{unif}(\{\pi_{\exp}^h\}_{h=1}^H)$ for $T$ episodes in real, add data to $\mathfrak{D}$
3: **for** $h = H, H-1, \ldots, 1$ **do**
4:

$$\widetilde{f}_h \leftarrow \arg\min_{f \in \mathcal{F}} \sum_{(s,a,r,s') \in \mathfrak{D}_{\text{sim}}^h} (f_h(s,a) - r - \max_{a'} \widehat{f}_{h+1}(s',a'))^2$$

$$\widehat{f}_h \leftarrow \arg\min_{f \in \mathcal{F}} \sum_{(s,a,r,s') \in \mathfrak{D}_h} (f_h(s,a) - r - \max_{a'} \widehat{f}_{h+1}(s',a'))^2$$

$$\text{s.t.} \quad \frac{1}{|\mathfrak{D}_{\text{sim}}|} \sum_{(s,a) \in \mathfrak{D}_{\text{sim}}^h} (f_h(s,a) - \widetilde{f}_h(s,a))^2 \leq \gamma \qquad \text{(B.2)}$$

5: **return** $\pi^{\widehat{f}}$

---

**Lemma B.1.** *Consider running Algorithm 3. Assume that $\mathfrak{D}_{\text{sim}}$ was generated as in Assumption 5, via the procedure of Lemma C.3 run with some parameter $\beta$, and $\gamma$ satisfies*

$$2V_{\max}^2 \epsilon_{\text{sim}}^2 + \frac{43 V_{\max}^2 \beta^2}{dH} \cdot \log \frac{8H|\mathcal{F}_h|}{\delta} + 6V_{\max}^2 \beta \sqrt{\frac{\log \frac{8H|\mathcal{F}_h|}{\delta}}{dH}} \leq \gamma.$$

*Furthermore, assume that there exists some $\mathfrak{C}, \epsilon > 0$ such that, for any $\pi$, $h \in [H]$, and $\mathcal{Z}' \subseteq \mathcal{S} \times \mathcal{A}$, we have:*

$$w_h^{\text{real},\pi}(\mathcal{Z}') \leq \mathfrak{C} \cdot w_h^{\text{real},\pi_{\exp}}(\mathcal{Z}') + \epsilon. \qquad \text{(B.3)}$$

*Then with probability at least $1 - 2\delta$, the policy $\pi^{\widehat{f}}$ generated by Algorithm 3 satisfies*

$$V_0^{\star} - V_0^{\pi^{\widehat{f}}} \leq 4\mathfrak{C}H\sqrt{\frac{256V_{\max}^2 \log(4H|\widetilde{\mathcal{F}}(\pi_{\exp}^{\mathsf{sim}})|/\delta)}{T}} + 4HV_{\max}\epsilon$$

*for*

$$\widetilde{\mathcal{F}}(\pi_{\exp}^{\mathsf{sim}}) := \{f \in \mathcal{F} \; : \; \mathbb{E}^{\mathsf{sim},\pi_{\exp}^{\mathsf{sim}}}[(f_h(s_h, a_h) - \mathcal{T}^{\mathsf{sim}}f_{h+1}(s_h, a_h))^2] \leq 2\gamma, \forall h \in [H]\}.$$

*Proof.* Let $\mathcal{E}$ denote the good event of Lemma B.2, which holds with probability at least $1 - 2\delta$. By Lemma A.4 we have

$$V_0^{\mathsf{real},\star} - V_0^{\mathsf{real},\pi^{\widehat{f}}} \leq \max_{\pi \in \{\pi^{\widehat{f}}, \pi^{\mathsf{real},\star}\}} \sum_{h=0}^{H-1} 2\left|\mathbb{E}^{\mathsf{real},\pi}[\widehat{f}_h(s_h, a_h) - \mathcal{T}^{\mathsf{real}}\widehat{f}_{h+1}(s_h, a_h)]\right|$$

$$\leq \max_{\pi} \sum_{h=0}^{H-1} 2\mathbb{E}^{\mathsf{real},\pi}[|\mathcal{E}_h^{\mathsf{real}}(\widehat{f})(s_h, a_h)|].$$

Let

$$\mathcal{Z}_{h,i} := \{(s,a) \; : \; |\mathcal{E}_h^{\mathsf{real}}(\widehat{f})(s,a)| \in [V_{\max} \cdot 2^{-i}, V_{\max} \cdot 2^{-i+1})\}.$$

Then we have, for any $\pi$,

$$\mathbb{E}^{\mathsf{real},\pi}[|\mathcal{E}_h^{\mathsf{real}}(\widehat{f})(s_h, a_h)|] \leq \sum_{i=1}^{\infty} w_h^{\mathsf{real},\pi}(\mathcal{Z}_{h,i}) \cdot V_{\max}2^{-i+1}$$

$$\leq \mathfrak{C} \cdot \sum_{i=1}^{\infty} w_h^{\mathsf{real},\pi_{\exp}}(\mathcal{Z}_{h,i}) \cdot V_{\max}2^{-i+1} + 2V_{\max}\epsilon$$

$$\leq 2\mathfrak{C} \cdot \mathbb{E}^{\mathsf{real},\pi_{\exp}}[|\mathcal{E}_h^{\mathsf{real}}(\widehat{f})(s_h, a_h)|] + 2V_{\max}\epsilon$$

where the second inequality follows from (B.3). On $\mathcal{E}$, by Lemma B.2 and Jensen's inequality, we have

$$\mathbb{E}^{\mathsf{real},\pi_{\exp}}[|\mathcal{E}_h^{\mathsf{real}}(\widehat{f})(s_h, a_h)|] \leq \sqrt{\mathbb{E}^{\mathsf{real},\pi_{\exp}}[\mathcal{E}_h^{\mathsf{real}}(\widehat{f})(s_h, a_h)^2]} \leq \sqrt{\frac{1}{T} \cdot 256V_{\max}^2 \log \frac{2H|\widetilde{\mathcal{F}}_h(\pi_{\exp}^{\mathsf{sim}})|}{\delta}}.$$

As this holds for each $h$ and $\pi$, we have therefore shown that

$$V_0^{\mathsf{real},\star} - V_0^{\mathsf{real},\pi^{\widehat{f}}} \leq 4\mathfrak{C} \cdot \sum_{h=0}^{H-1} \sqrt{\frac{1}{T} \cdot 256V_{\max}^2 \log \frac{2H|\widetilde{\mathcal{F}}_h(\pi_{\exp}^{\mathsf{sim}})|}{\delta}} + 4HV_{\max}\epsilon$$

$$\leq 4\mathfrak{C}H\sqrt{\frac{1}{T} \cdot 256V_{\max}^2 \log \frac{2H|\widetilde{\mathcal{F}}(\pi_{\exp}^{\mathsf{sim}})|}{\delta}} + 4HV_{\max}\epsilon.$$

This proves the result. $\qquad\square$

**Lemma B.2.** *With probability at least $1 - 2\delta$, for each $h \in [H]$ simultaneously, as long as the conditions on $\gamma$ given in Lemma B.3 hold, we have*

$$\mathbb{E}^{\mathsf{real},\pi_{\exp}}[(\widehat{f}_h(s_h, a_h) - \mathcal{T}^{\mathsf{real}}\widehat{f}_{h+1}(s_h, a_h))^2] \leq \frac{1}{T} \cdot 256V_{\max}^2 \log(2H|\widetilde{\mathcal{F}}_h(\pi_{\exp}^{\mathsf{sim}})|/\delta),$$

*and $\widehat{f}_h \in \widetilde{\mathcal{F}}_h(\pi_{\exp}^{\mathsf{sim}})$ for all $h \in [H]$, where*

$$\widetilde{\mathcal{F}}_h(\pi_{\exp}^{\mathsf{sim}}) := \{f_h \in \mathcal{F}_h \; : \; \exists f_{h+1} \in \mathcal{F}_{h+1} \text{ s.t. } \mathbb{E}^{\mathsf{sim},\pi_{\exp}^{\mathsf{sim}}}[(f_h(s_h, a_h) - \mathcal{T}^{\mathsf{sim}}f_{h+1}(s_h, a_h))^2] \leq 2\gamma\}.$$

*Proof.* Let $\widehat{\mathcal{F}}_h$ denote the feasible set of (B.2) at step $h$. By Lemma B.3, with probability at least $1 - \delta$, $\widehat{\mathcal{F}}_h^t \subseteq \widetilde{\mathcal{F}}_h$, and, furthermore, that $\mathcal{T}^{\mathsf{real}}\widehat{f}_{h+1}$ is feasible. The result then follows from Lemma 3 of [57], since the constraint on the regression problem restricts the version space. $\qquad\square$

**Lemma B.3.** *Assume that data in $\mathfrak{D}_{\mathsf{sim}}$ is generated as in Assumption 5 via the procedure of Lemma C.3 run with some parameter $\beta$, and $\gamma$ satisfies*

$$2V_{\max}^2 \epsilon_{\mathsf{sim}}^2 + \frac{43V_{\max}^2\beta^2}{dH} \cdot \log \frac{8H|\mathcal{F}_h|}{\delta} + 6V_{\max}^2\beta\sqrt{\frac{\log \frac{8H|\mathcal{F}_h|}{\delta}}{dH}} \leq \gamma.$$

*Then with probability at least $1 - \delta$ we have, for each $h \in [H]$:*

1. *$\mathcal{T}^{\mathsf{real}}\widehat{f}_{h+1}$ is feasible for (B.2).*

2. *The set of feasible $f$ for (B.2) is a subset of*

$$\{f \in \mathcal{F} \ : \ \mathbb{E}^{\mathsf{sim},\pi_{\exp}^{\mathsf{sim}}}[(f_h(s_h,a_h) - \mathcal{T}^{\mathsf{sim}}\widehat{f}_{h+1}(s_h,a_h))^2] \leq 2\gamma\}.$$

*Proof.* By Lemma C.1, we have that with probability at least $1 - \delta/2H$,

$$\frac{1}{T_{\mathsf{sim}}} \sum_{t=1}^{T_{\mathsf{sim}}} (\mathcal{T}^{\mathsf{real}}\widehat{f}_{h+1}(\widetilde{s}_h^t, \widetilde{a}_h^t) - \widetilde{f}_h(\widetilde{s}_h^t, \widetilde{a}_h^t))^2 \leq 2V_{\max}^2\epsilon_{\mathsf{sim}}^2 + \frac{512V_{\max}^2}{T_{\mathsf{sim}}} \cdot \log \frac{8H|\mathcal{F}_h|}{\delta} + V_{\max}^2\sqrt{\frac{2\log \frac{4H|\mathcal{F}_h|}{\delta}}{T_{\mathsf{sim}}}}.$$

By Lemma C.3, we have $\frac{12dH}{\beta^2} \leq T_{\mathsf{sim}}$, which implies

$$\frac{1}{T_{\mathsf{sim}}} \sum_{t=1}^{T_{\mathsf{sim}}} (\mathcal{T}^{\mathsf{real}}\widehat{f}_{h+1}(\widetilde{s}_h^t, \widetilde{a}_h^t) - \widetilde{f}_h(\widetilde{s}_h^t, \widetilde{a}_h^t))^2 \leq 2V_{\max}^2\epsilon_{\mathsf{sim}}^2 + \frac{43V_{\max}^2\beta^2}{dH} \cdot \log \frac{8H|\mathcal{F}_h|}{\delta} + V_{\max}^2\beta\sqrt{\frac{\log \frac{4H|\mathcal{F}_h|}{\delta}}{6dH}}.$$

Part 1 then follows given our assumption on $\gamma$.

To bound the feasible set for (B.2) we appeal to Lemma C.2 which states that with probability at least $1 - \delta/2H$ we have that the feasible set of (B.2) is a subset of

$$\left\{ f_h \in \mathcal{F}_h \ : \ \mathbb{E}^{\mathsf{sim},\pi_{\exp}^{\mathsf{sim}}}[(f_h(s_h,a_h) - \mathcal{T}^{\mathsf{sim}}\widehat{f}_{h+1}(s_h,a_h))^2] \leq \gamma + 18V_{\max}^2\sqrt{\frac{\log \frac{8H|\mathcal{F}_h|}{\delta}}{T_{\mathsf{sim}}}} \right\}.$$

Again using that $\frac{12dH}{\beta^2} \leq T_{\mathsf{sim}}$, we have have that this is a subset of

$$\left\{ f_h \in \mathcal{F}_h \ : \ \mathbb{E}^{\mathsf{sim},\pi_{\exp}^{\mathsf{sim}}}[(f_h(s_h,a_h) - \mathcal{T}^{\mathsf{sim}}\widehat{f}_{h+1}(s_h,a_h))^2] \leq \gamma + 18V_{\max}^2\beta\sqrt{\frac{\log \frac{8H|\mathcal{F}_h|}{\delta}}{12dH}} \right\}$$

$$\subseteq \left\{ f_h \in \mathcal{F}_h \ : \ \mathbb{E}^{\mathsf{sim},\pi_{\exp}^{\mathsf{sim}}}[(f_h(s_h,a_h) - \mathcal{T}^{\mathsf{sim}}\widehat{f}_{h+1}(s_h,a_h))^2] \leq 2\gamma \right\}$$

where the inclusion follows from our assumption on $\gamma$. The result then follows from a union bound. $\qquad\square$

## B.2 Performance of Full-Rank sim Policies in real

**Lemma B.4.** *Consider policies $\{\pi_{\exp}^h\}_{h=1}^H$, and assume that*

$$\lambda_{\min}\left(\mathbf{\Lambda}_{\pi_{\exp}^h,h}^{\mathsf{s}}\right) \geq \bar{\lambda}_{\min}, \quad \forall h \in [H] \tag{B.4}$$

*and that $\pi_{\exp}^h$ plays actions uniformly at random for $h' > h$. Let $\pi_{\exp} = \mathrm{unif}(\{\pi_{\exp}^h\}_{h=1}^H)$. Then, for any $\pi$, $\kappa > 0$, $\gamma > 0$, $h \in [H]$, and $\mathcal{Z}' \subseteq \mathcal{S} \times \mathcal{A}$, we have*

$$w_h^{\mathsf{real},\pi}(\mathcal{Z}') \leq \frac{4H\gamma A^{k^\star - 2}}{\kappa} \cdot w_h^{\mathsf{real},\pi_{\exp}}(\mathcal{Z}') + 4\kappa,$$

*where*

$$\xi := 2\sqrt{\frac{A}{\bar{\lambda}_{\min}}\left(\frac{d}{\gamma} + H\epsilon_{\mathsf{sim}}\right)} \quad and \quad k^\star := \lceil \frac{\log 1/\kappa}{\log 1/\xi} \rceil.$$

*Proof.* Denote

$$\widetilde{\mathcal{Z}}_{h+1} := \{(s,a) \ : \ \phi^{\mathsf{r}}(s,a)^\top (\mathbf{\Lambda}^{\mathsf{r}}_{\pi^h_{\exp},h+1})^{-1} \phi^{\mathsf{r}}(s,a) > \gamma\}$$

for some $\gamma > 0$. We have

$$
\begin{aligned}
w^{\mathsf{real},\pi^h_{\exp}}_{h+1}(\widetilde{\mathcal{Z}}_{h+1}) &= \mathbb{E}^{\mathsf{real},\pi^h_{\exp}}[\mathbb{I}\{(s_{h+1},a_{h+1}) \in \widetilde{\mathcal{Z}}_{h+1}\}] \\
&\overset{(a)}{\le} \mathbb{E}^{\mathsf{real},\pi^h_{\exp}}\left[\frac{\phi^{\mathsf{r}}(s_{h+1},a_{h+1})^\top (\mathbf{\Lambda}^{\mathsf{r}}_{\pi^h_{\exp},h+1})^{-1} \phi^{\mathsf{r}}(s_{h+1},a_{h+1})}{\gamma} \cdot \mathbb{I}\{(s_{h+1},a_{h+1}) \in \widetilde{\mathcal{Z}}_{h+1}\}\right] \\
&\le \mathbb{E}^{\mathsf{real},\pi^h_{\exp}}\left[\frac{\phi^{\mathsf{r}}(s_{h+1},a_{h+1})^\top (\mathbf{\Lambda}^{\mathsf{r}}_{\pi^h_{\exp},h+1})^{-1} \phi^{\mathsf{r}}(s_{h+1},a_{h+1})}{\gamma}\right] \\
&= \frac{1}{\gamma} \cdot \mathrm{tr}\left(\mathbb{E}^{\mathsf{real},\pi^h_{\exp}}[\phi^{\mathsf{r}}(s_{h+1},a_{h+1})\phi^{\mathsf{r}}(s_{h+1},a_{h+1})^\top](\mathbf{\Lambda}^{\mathsf{r}}_{\pi^h_{\exp},h+1})^{-1}\right) \\
&= \frac{d}{\gamma}
\end{aligned}
$$

where $(a)$ follows since for all $(s,a) \in \widetilde{\mathcal{Z}}_{h+1}$, we have $1 < \phi^{\mathsf{r}}(s,a)^\top (\mathbf{\Lambda}^{\mathsf{r}}_{\pi^h_{\exp},h+1})^{-1}\phi^{\mathsf{r}}(s,a)/\gamma$. By Lemma A.2, we then have that

$$w^{\mathsf{sim},\pi^h_{\exp}}_{h+1}(\widetilde{\mathcal{Z}}_{h+1}) \le \frac{d}{\gamma} + H\epsilon_{\mathsf{sim}}. \tag{B.5}$$

Let $\widetilde{\mathcal{S}}_{h+1} := \{s \ : \ \exists a \text{ s.t. } (s,a) \in \widetilde{\mathcal{Z}}_{h+1}\}$ and note that

$$
\begin{aligned}
w^{\mathsf{sim},\pi^h_{\exp}}_{h+1}(\widetilde{\mathcal{Z}}_{h+1}) &= \mathbb{E}^{\mathsf{sim},\pi^h_{\exp}}\left[\int_{\widetilde{\mathcal{S}}_{h+1}} \sum_{a:(s,a)\in\widetilde{\mathcal{Z}}_{h+1}} \pi^h_{\exp}(a \mid s, h+1)\mathrm{d}P^{\mathsf{sim}}_h(s \mid s_h, a_h)\right] \\
&\ge \frac{1}{A}\mathbb{E}^{\mathsf{sim},\pi^h_{\exp}}\left[\int_{\widetilde{\mathcal{S}}_{h+1}} \mathrm{d}\boldsymbol{\mu}^{\mathsf{s}}_h(s)^\top \phi^{\mathsf{s}}(s_h, a_h)\right] \\
&= \frac{1}{A}\mathbb{E}^{\mathsf{sim},\pi^h_{\exp}}[P^{\mathsf{sim}}_h(\widetilde{\mathcal{S}}_{h+1} \mid s_h, a_h)] \\
&\ge \frac{1}{A}\mathbb{E}^{\mathsf{sim},\pi^h_{\exp}}[P^{\mathsf{sim}}_h(\widetilde{\mathcal{S}}_{h+1} \mid s_h, a_h)^2]
\end{aligned}
$$

where we have used the fact that $\pi^h_{\exp}(a \mid s, h+1) = 1/A$ for all $(s,a)$ by assumption, and define $P^{\mathsf{sim}}_h(\widetilde{\mathcal{S}}_{h+1} \mid s, a) := \mathbb{P}^{\mathsf{sim}}[s_{h+1} \in \widetilde{\mathcal{S}}_{h+1} \mid s_h = s, a_h = a] = \int_{\widetilde{\mathcal{S}}_{h+1}} \mathrm{d}\boldsymbol{\mu}^{\mathsf{s}}_h(s)^\top \phi^{\mathsf{s}}(s,a)$, where the last equality follows from the definition of a linear MDP. Letting $\boldsymbol{\mu}^{\mathsf{s}}_h(\widetilde{\mathcal{S}}_{h+1}) := \int_{\widetilde{\mathcal{S}}_{h+1}} \mathrm{d}\boldsymbol{\mu}^{\mathsf{s}}_h(s)$, note that:

$$
\begin{aligned}
\frac{1}{A}\mathbb{E}^{\mathsf{sim},\pi^h_{\exp}}[P^{\mathsf{sim}}_h(\widetilde{\mathcal{S}}_{h+1} \mid s_h, a_h)^2] &= \frac{1}{A}\boldsymbol{\mu}^{\mathsf{s}}_h(\widetilde{\mathcal{S}}_{h+1})^\top \mathbb{E}^{\mathsf{sim},\pi^h_{\exp}}[\phi^{\mathsf{s}}(s_h,a_h)\phi^{\mathsf{s}}(s_h,a_h)^\top]\boldsymbol{\mu}^{\mathsf{s}}_h(\widetilde{\mathcal{S}}_{h+1}) \\
&= \frac{1}{A}\boldsymbol{\mu}^{\mathsf{s}}_h(\widetilde{\mathcal{S}}_{h+1})^\top \mathbf{\Lambda}^{\mathsf{s}}_{\pi^h_{\exp},h}\boldsymbol{\mu}^{\mathsf{s}}_h(\widetilde{\mathcal{S}}_{h+1}) \\
&\ge \frac{\overline{\lambda}_{\min}}{A}\|\boldsymbol{\mu}^{\mathsf{s}}_h(\widetilde{\mathcal{S}}_{h+1})\|^2_2,
\end{aligned}
$$

where the last inequality follows from (B.4). Combining this with (B.5), we have

$$\frac{d}{\gamma} + H\epsilon_{\mathsf{sim}} \ge \frac{\overline{\lambda}_{\min}}{A}\|\boldsymbol{\mu}^{\mathsf{s}}_h(\widetilde{\mathcal{S}}_{h+1})\|^2_2.$$

Now note that, for any $z \in \mathcal{S} \times \mathcal{A}$:

$$P^{\mathsf{sim}}_h(\widetilde{\mathcal{S}}_{h+1} \mid z) = \int_{\widetilde{\mathcal{S}}_{h+1}} \mathrm{d}P^{\mathsf{sim}}_h(s \mid z) = \left(\int_{\widetilde{\mathcal{S}}_{h+1}} \mathrm{d}\boldsymbol{\mu}^{\mathsf{s}}_h(s)\right)^\top \phi^{\mathsf{s}}(z) \le \|\boldsymbol{\mu}^{\mathsf{s}}_h(\widetilde{\mathcal{S}}_{h+1})\|_2$$

and we also have that $P_h^{\mathsf{sim}}(\widetilde{\mathcal{S}}_{h+1} \mid z) \geq P_h^{\mathsf{real}}(\widetilde{\mathcal{S}}_{h+1} \mid z) - \epsilon_{\mathrm{sim}}$ under Assumption 1. Putting this together we have that for all $z \in \mathcal{S} \times \mathcal{A}$:

$$P_h^{\mathsf{real}}(\widetilde{\mathcal{S}}_{h+1} \mid z) \leq \sqrt{\frac{A}{\bar{\lambda}_{\min}}\left(\frac{d}{\gamma} + H\epsilon_{\mathrm{sim}}\right)} + \epsilon_{\mathrm{sim}}.$$

Note that we can always take $\epsilon_{\mathrm{sim}} \leq 1$, and will always have $\bar{\lambda}_{\min} \leq 1$. This implies that $\epsilon_{\mathrm{sim}} \leq \sqrt{\frac{A}{\bar{\lambda}_{\min}}\left(\frac{d}{\gamma} + H\epsilon_{\mathrm{sim}}\right)}$. Thus,

$$P_h^{\mathsf{real}}(\widetilde{\mathcal{S}}_{h+1} \mid z) \leq 2\sqrt{\frac{A}{\bar{\lambda}_{\min}}\left(\frac{d}{\gamma} + H\epsilon_{\mathrm{sim}}\right)} =: \xi.$$

**Coverage of $\pi_{\exp}$ in real.** Let $k^\star := \lceil \frac{\log 1/\kappa}{\log 1/\xi} \rceil$, so that $\xi^{k^\star} \leq \kappa$. Let $\bar{\mathcal{Z}}_h := (\mathcal{S} \times \mathcal{A}) \backslash \widetilde{\mathcal{Z}}_h$. Fix some $\mathcal{Z}' \subseteq (\mathcal{S} \times \mathcal{A})$, $h \in [H]$, and policy $\pi$.

Consider some $z \in \bar{\mathcal{Z}}_h$, and some $\mathcal{S}' \subseteq \mathcal{S}$. Then note that[5]

$$\begin{aligned} P_h^{\mathsf{real}}(\mathcal{S}' \mid z) = \boldsymbol{\mu}_h^{\mathsf{r}}(\mathcal{S}')^\top \boldsymbol{\phi}^{\mathsf{r}}(z) &= \boldsymbol{\mu}_h^{\mathsf{r}}(\mathcal{S}')^\top (\boldsymbol{\Lambda}_{\pi_{\exp}^{h-1},h}^{\mathsf{r}})^{1/2} (\boldsymbol{\Lambda}_{\pi_{\exp}^{h-1},h}^{\mathsf{r}})^{-1/2} \boldsymbol{\phi}^{\mathsf{r}}(z) \\ &\leq \|\boldsymbol{\mu}_h^{\mathsf{r}}(\mathcal{S}')\|_{\boldsymbol{\Lambda}_{\pi_{\exp}^{h-1},h}^{\mathsf{r}}} \|\boldsymbol{\phi}^{\mathsf{r}}(z)\|_{(\boldsymbol{\Lambda}_{\pi_{\exp}^{h-1},h}^{\mathsf{r}})^{-1}} \\ &\leq \sqrt{\gamma} \|\boldsymbol{\mu}_h^{\mathsf{r}}(\mathcal{S}')\|_{\boldsymbol{\Lambda}_{\pi_{\exp}^{h-1},h}^{\mathsf{r}}} \end{aligned}$$

where the last inequality follows from the definition of $\bar{\mathcal{Z}}_h$. Note, though, that

$$\|\boldsymbol{\mu}_h^{\mathsf{r}}(\mathcal{S}')\|_{\boldsymbol{\Lambda}_{\pi_{\exp}^{h-1},h}^{\mathsf{r}}}^2 = \mathbb{E}^{\mathsf{real},\pi_{\exp}^{h-1}}[(\boldsymbol{\mu}_h^{\mathsf{r}}(\mathcal{S}')^\top \boldsymbol{\phi}^{\mathsf{r}}(z_h))^2] = \mathbb{E}^{\mathsf{real},\pi_{\exp}^{h-1}}[P_h^{\mathsf{real}}(\mathcal{S}' \mid z_h)^2].$$

This implies that for all $z \in \bar{\mathcal{Z}}_h$,

$$\mathbb{E}^{\mathsf{real},\pi_{\exp}^{h-1}}[P_h^{\mathsf{real}}(\mathcal{S}' \mid z_h)^2] \geq \frac{1}{\gamma} \cdot P_h^{\mathsf{real}}(\mathcal{S}' \mid z)^2.$$

For $h' < h$, define

$$\mathcal{S}_{h',i} := \{s \ : \ w_h^{\mathsf{real},\pi}(\mathcal{Z}' \mid s_{h'} = s) \in [2^{-i+1}, 2^{-i})\}$$

for $w_h^{\mathsf{real},\pi}(\mathcal{Z} \mid s_{h'} = s) := \mathbb{P}^{\mathsf{real},\pi}[z_h \in \mathcal{Z} \mid s_{h'} = s]$. Note that we then have $w_h^{\mathsf{real},\pi}(\mathcal{Z}' \mid \mathcal{S}_{h',i}) \in [2^{-i+1}, 2^{-i})$. By what we have just shown, we have that for $z \in \bar{\mathcal{Z}}_{h'}$

$$\mathbb{E}^{\mathsf{real},\pi_{\exp}^{h'-1}}[P_{h'}^{\mathsf{real}}(\mathcal{S}_{h'+1,i} \mid z_{h'})^2] \geq \frac{1}{\gamma} \cdot P_{h'}^{\mathsf{real}}(\mathcal{S}_{h'+1,i} \mid z)^2$$

which implies that

$$\mathbb{E}^{\mathsf{real},\pi_{\exp}^{h'-1}}[P_{h'}^{\mathsf{real}}(\mathcal{S}_{h'+1,i} \mid z_{h'})] \geq \frac{1}{\gamma} \cdot P_{h'}^{\mathsf{real}}(\mathcal{S}_{h'+1,i} \mid z)^2. \tag{B.6}$$

---

[5]If $\boldsymbol{\Lambda}_{\pi_{\exp}^{h-1},h}^{\mathsf{r}}$ is not invertible, we can repeat this argument with $\boldsymbol{\Lambda}_{\pi_{\exp}^{h-1},h}^{\mathsf{r}} + \lambda I$ and take $\lambda \to 0$.

Fix $z \in \bar{\mathcal{Z}}_{h'}$. Note that

$$w_h^{\text{real},\pi}(\mathcal{Z}' \mid z_{h'} = z) = \mathbb{E}_{s \sim P_{h'}^{\text{real}}(\cdot \mid z)}[w_h^{\text{real},\pi}(\mathcal{Z}' \mid s_{h'+1} = s)]$$

$$= \sum_{i=1}^{\infty} \mathbb{E}_{s \sim P_{h'}^{\text{real}}(\cdot \mid z)}[w_h^{\text{real},\pi}(\mathcal{Z}' \mid s_{h'+1} = s) \cdot \mathbb{I}\{s \in \mathcal{S}_{h'+1,i}\}]$$

$$\leq \sum_{i=1}^{\infty} 2^{-i+1} P_{h'}^{\text{real}}(\mathcal{S}_{h'+1,i} \mid z)$$

$$= \sum_{i=1}^{\lfloor \log 4/\kappa \rfloor} 2^{-i+1} P_{h'}^{\text{real}}(\mathcal{S}_{h'+1,i} \mid z) + \kappa$$

$$\leq \sum_{i=1}^{\lfloor \log 4/\kappa \rfloor} 2^{-i+1} P_{h'}^{\text{real}}(\mathcal{S}_{h'+1,i} \mid z) \cdot \mathbb{I}\{P_{h'}^{\text{real}}(\mathcal{S}_{h'+1,i} \mid z) \geq \kappa\} + 3\kappa$$

$$\leq 2 \sum_{i=1}^{\lfloor \log 4/\kappa \rfloor} \mathbb{E}_{s \sim \lambda_i}[w_h^{\text{real},\pi}(\mathcal{Z}' \mid s_{h'+1} = s)] P_{h'}^{\text{real}}(\mathcal{S}_{h'+1,i} \mid z) \cdot \mathbb{I}\{P_{h'}^{\text{real}}(\mathcal{S}_{h'+1,i} \mid z) \geq \kappa\} + 3\kappa$$

for any $\lambda_i \in \triangle_{\mathcal{S}_{h'+1,i}}$. Note also that, since $\pi_{\exp}^{h'-1}$ plays randomly for all $h'' \geq h'$, we have:

$$w_h^{\text{real},\pi_{\exp}^{h'-1}}(\mathcal{Z}' \mid s_{h'+1} = s) \geq \frac{1}{A^{h-h'}} \cdot w_h^{\text{real},\pi}(\mathcal{Z}' \mid s_{h'+1} = s),$$

since with probability $1/A^{h-h'}$ on any given episode, $\pi_{\exp}^{h'-1}$ will play the same sequence of actions as $\pi$ from steps $h'$ to $h$. It follows that we can bound the above as:

$$\leq 2A^{h-h'} \cdot \sum_{i=1}^{\lfloor \log 4/\kappa \rfloor} \mathbb{E}_{s \sim \lambda_i}[w_h^{\text{real},\pi_{\exp}^{h'-1}}(\mathcal{Z}' \mid s_{h'+1} = s)] P_{h'}^{\text{real}}(\mathcal{S}_{h'+1,i} \mid z) \cdot \mathbb{I}\{P_{h'}^{\text{real}}(\mathcal{S}_{h'+1,i} \mid z) \geq \kappa\} + 3\kappa$$

$$\overset{(a)}{\leq} \frac{2A^{h-h'}\gamma}{\kappa} \cdot \sum_{i=1}^{\lfloor \log 4/\kappa \rfloor} \mathbb{E}_{s \sim \lambda_i}[w_h^{\text{real},\pi_{\exp}^{h'-1}}(\mathcal{Z}' \mid s_{h'+1} = s)] \mathbb{E}^{\text{real},\pi_{\exp}^{h'-1}}[P_{h'}^{\text{real}}(\mathcal{S}_{h'+1,i} \mid z_{h'})] \mathbb{I}\{P_{h'}^{\text{real}}(\mathcal{S}_{h'+1,i} \mid z) \geq \kappa\} + 3\kappa$$

$$\leq \frac{2\gamma A^{h-h'}}{\kappa} \cdot \sum_{i=1}^{\lfloor \log 4/\kappa \rfloor} \mathbb{E}_{s \sim \lambda_i}[w_h^{\text{real},\pi_{\exp}^{h'-1}}(\mathcal{Z}' \mid s_{h'+1} = s)] \cdot w_{h'+1}^{\text{real},\pi_{\exp}^{h'-1}}(\mathcal{S}_{h'+1,i}) + 3\kappa$$

$$\overset{(b)}{=} \frac{2\gamma A^{h-h'}}{\kappa} \cdot \sum_{i=1}^{\lfloor \log 4/\kappa \rfloor} \sum_{s \in \mathcal{S}_{h'+1,i}} w_h^{\text{real},\pi_{\exp}^{h'-1}}(\mathcal{Z}' \mid s_{h'+1} = s) w_{h'+1}^{\text{real},\pi_{\exp}^{h'-1}}(s) + 3\kappa$$

$$\leq \frac{2\gamma A^{h-h'}}{\kappa} \cdot w_h^{\text{real},\pi_{\exp}^{h'-1}}(\mathcal{Z}') + 3\kappa$$

where $(a)$ follows from (B.6) and since $P_{h'}^{\text{real}}(\mathcal{S}_{h',i} \mid z) \geq \kappa$, and $(b)$ follows choosing $\lambda_i(s) = w_{h'+1}^{\text{real},\pi_{\exp}^{h'-1}}(s)/w_{h'+1}^{\text{real},\pi_{\exp}^{h'-1}}(\mathcal{S}_{h'+1,i}) \cdot \mathbb{I}\{s \in \mathcal{S}_{h'+1,i}\}$. We therefore have that, for all $z \in \bar{\mathcal{Z}}_{h'}$:

$$w_h^{\text{real},\pi}(\mathcal{Z}' \mid z_{h'} = z) \leq \frac{2\gamma A^{h-h'}}{\kappa} \cdot w_h^{\text{real},\pi_{\exp}^{h'-1}}(\mathcal{Z}') + 3\kappa. \tag{B.7}$$

**Controlling events.** Consider events $\mathcal{E} := \{z_h \in \mathcal{Z}'\}$ and $\mathcal{E}_{h'} := \{z_{h'} \in \bar{\mathcal{Z}}_{h'}\}$. We then have

$$w_h^{\text{real},\pi}(\mathcal{Z}') = \mathbb{P}^{\text{real},\pi}[\mathcal{E}]$$

$$= \mathbb{P}^{\text{real},\pi}[\mathcal{E} \cap \mathcal{E}_{h-1}] + \mathbb{P}^{\text{real},\pi}[\mathcal{E} \cap \mathcal{E}_{h-1}^c]$$

$$= \sum_{h'=h-k^\star+1}^{h} \mathbb{P}^{\text{real},\pi}[\mathcal{E} \cap \mathcal{E}_{h'-1} \cap \bigcap_{i=h'}^{h-1} \mathcal{E}_i^c] + \mathbb{P}^{\text{real},\pi}[\mathcal{E} \cap \mathcal{E}_{h-k^\star-1} \cap \bigcap_{i=h-k^\star}^{h-1} \mathcal{E}_i^c]$$

$$\leq \sum_{h'=h-k^\star+1}^{h} \mathbb{P}^{\text{real},\pi}[\mathcal{E} \cap \mathcal{E}_{h'-1}] + \mathbb{P}^{\text{real},\pi}[\mathcal{E} \cap \mathcal{E}_{h-k^\star-1} \cap \bigcap_{i=h-k^\star}^{h-1} \mathcal{E}_i^c].$$

We now analyze each of these terms. First, note that

$$\mathbb{P}^{\mathsf{real},\pi}[\mathcal{E} \cap \mathcal{E}_{h'-1}] = \mathbb{P}^{\mathsf{real},\pi}[\mathcal{E} \mid \mathcal{E}_{h'-1}]\mathbb{P}^{\mathsf{real},\pi}[\mathcal{E}_{h'-1}] \le \mathbb{P}^{\mathsf{real},\pi}[\mathcal{E} \mid \mathcal{E}_{h'-1}] = w_h^{\mathsf{real},\pi}(\mathcal{Z}' \mid z_{h'-1} \in \bar{\mathcal{Z}}_{h'-1}).$$

We can then bound

$$w_h^{\mathsf{real},\pi}(\mathcal{Z}' \mid z_{h'-1} \in \bar{\mathcal{Z}}_{h'-1}) \le \frac{2\gamma A^{h-h'-1}}{\kappa} \cdot w_h^{\mathsf{real},\pi_{\exp}^{h'-2}}(\mathcal{Z}') + 3\kappa$$

where the inequality follows from (B.7). For the second term, we have

$$\mathbb{P}^{\mathsf{real},\pi}[\mathcal{E} \cap \mathcal{E}_{h-k^\star-1} \cap \bigcap_{i=h-k^\star}^{h-1} \mathcal{E}_i^c] \le \mathbb{P}^{\mathsf{real},\pi}[\mathcal{E} \cap \bigcap_{i=h-k^\star}^{h-1} \mathcal{E}_i^c]$$

$$= \mathbb{P}^{\mathsf{real},\pi}[\mathcal{E} \mid \bigcap_{i=h-k^\star}^{h-1} \mathcal{E}_i^c] \cdot \prod_{j=1}^{k^\star} \mathbb{P}^{\mathsf{real},\pi}[\mathcal{E}_{h-j}^c \mid \bigcap_{i=h-k^\star}^{h-j-1} \mathcal{E}_i^c].$$

Note, however, that $\mathbb{P}^{\mathsf{real},\pi}[\mathcal{E} \mid \bigcap_{i=h-k^\star}^{h-1} \mathcal{E}_i^c] \le \xi$ and $\mathbb{P}^{\mathsf{real},\pi}[\mathcal{E}_{h-j}^c \mid \bigcap_{i=h-k^\star}^{h-j-1} \mathcal{E}_i^c] \le \xi$ for all $j$. We therefore can bound the above as

$$\xi^{k^\star+1} \le \kappa.$$

Altogether, then, we have that

$$w_h^{\mathsf{real},\pi}(\mathcal{Z}') \le \sum_{h'=h-k^\star+1}^{h} \frac{2\gamma A^{h-h'-1}}{\kappa} \cdot w_h^{\mathsf{real},\pi_{\exp}^{h'-2}}(\mathcal{Z}') + 4\kappa.$$

Furthermore, since $\pi_{\exp} = \mathrm{unif}(\{\pi_{\exp}^h\}_{h=1}^H)$, we have $w_h^{\mathsf{real},\pi_{\exp}^{h'-2}}(\mathcal{Z}') \le H w_h^{\mathsf{real},\pi_{\exp}}(\mathcal{Z}')$, so we conclude that

$$w_h^{\mathsf{real},\pi}(\mathcal{Z}') \le \frac{4H\gamma A^{k^\star-2}}{\kappa} \cdot w_h^{\mathsf{real},\pi_{\exp}}(\mathcal{Z}') + 4\kappa.$$

$\square$

## B.3 Proof of Unconstrained Upper Bound

**Theorem 2.** *Assume that one of the two conditions is met:*

1. *For each $h$, $\pi_{\exp}^h$ plays actions uniformly at random for $h' > h$,*

$$\lambda_{\min}\left(\mathbf{\Lambda}_{\pi_{\exp}^h,h}^{\mathsf{s}}\right) \ge \bar{\lambda}_{\min}, \tag{B.8}$$

*and*

$$T \ge c \cdot \frac{V_{\max}^4 H^4 d^2 A^{2(k^\star-2)} \log(2H|\mathcal{F}|/\delta)}{\epsilon^4 \epsilon_{\mathsf{sim}}^2},$$

*for*

$$k^\star = \lceil \frac{\log_A \frac{64HV_{\max}}{\epsilon}}{\log_A 1/\xi} \rceil, \quad \xi = 2\sqrt{\frac{2HA}{\bar{\lambda}_{\min}} \cdot \epsilon_{\mathsf{sim}}}.$$

2. *$\epsilon_{\mathsf{sim}} \le \epsilon/4H^2$ and*

$$T \ge \frac{16H^2 \log \frac{4}{\delta}}{\epsilon^2}.$$

*Then with probability at least $1 - \delta$, Algorithm 1 returns a $\widehat{\pi}$ such that $V_0^{\mathsf{real},\pi^{\mathsf{real},\star}} - V_0^{\mathsf{real},\widehat{\pi}} \le \epsilon$.*

*Proof.* We consider each of the conditions above.

**Condition 1.** First, note that by our assumption on $\pi_{\mathrm{exp}}$ and applying Lemma B.4 with $\kappa = \frac{\epsilon}{64HV_{\max}}$ and $\gamma = \frac{d}{H\epsilon_{\mathrm{sim}}}$, for any $\pi$ and $\mathcal{Z}' \subseteq \mathcal{S} \times \mathcal{A}$, we have

$$w_h^{\mathrm{real},\pi}(\mathcal{Z}') \le \frac{256dHV_{\max}A^{k^\star-2}}{\epsilon\epsilon_{\mathrm{sim}}} \cdot w_h^{\mathrm{real},\pi_{\mathrm{exp}}}(\mathcal{Z}') + \frac{\epsilon}{16HV_{\max}}$$

for

$$k^\star = \lceil \frac{\log_A \frac{64HV_{\max}}{\epsilon}}{\log_A 1/\xi} \rceil, \quad \xi = 2\sqrt{\frac{2HA}{\overline{\lambda}_{\min}} \cdot \epsilon_{\mathrm{sim}}}.$$

By Lemma B.1 we then have that, with probability at least $1 - \delta$[6],

$$V_0^{\mathrm{real},\pi^{\mathrm{real},\star}} - V_0^{\mathrm{real},\widehat{\pi}} \le \frac{256dHV_{\max}A^{k^\star-2}}{\epsilon\epsilon_{\mathrm{sim}}} \cdot 4H\sqrt{\frac{256V_{\max}^2 \log(2H|\mathcal{F}|/\delta)}{T}} + \epsilon/4$$

$$\le \epsilon/2$$

where the last inequality follows under our condition on $T$.

**Condition 2.** By Lemma A.3, we have that $V_0^{\mathrm{real},\star} - V_0^{\mathrm{real},\pi^{\mathrm{sim},\star}} \le 2H^2\epsilon_{\mathrm{sim}}$. Thus, if $\epsilon_{\mathrm{sim}} \le \epsilon/4H^2$, we have $V_0^{\mathrm{real},\star} - V_0^{\mathrm{real},\pi^{\mathrm{sim},\star}} \le \epsilon/2$.

**Concluding the Proof.** By what we have shown, as long as one of our conditions is met, we will have that with probability at least $1 - \delta/2$, there exists $\pi \in \{\pi^{\widehat{f}}, \pi^{\mathrm{sim},\star}\}$ such that $V_0^{\mathrm{real},\star} - V_0^{\mathrm{real},\pi} \le \epsilon/2$. Denote this policy as $\widetilde{\pi}$.

Note that $V_0^{\mathrm{real},\pi} = \mathbb{E}^{\mathrm{real},\pi}[\sum_{h=0}^{H-1} r_h]$ and that $\sum_{h=0}^{H-1} r_h \in [0, H]$ almost surely. Consider playing $\pi$ for $T/4$ episodes in real and let $R^i$ denote the total return of the $i$th episode. Let

$$\widehat{V}_0^\pi := \frac{4}{T} \sum_{i=1}^{T/4} R^i.$$

By Hoeffding's inequality we have that, with probability at least $1 - \delta/4$:

$$|\widehat{V}_0^\pi - V_0^{\mathrm{real},\pi}| \le H\sqrt{\frac{4\log\frac{4}{\delta}}{T}}.$$

Thus, if

$$T \ge \frac{16H^2 \log\frac{4}{\delta}}{\epsilon^2}, \tag{B.9}$$

we have that $|\widehat{V}_0^\pi - V_0^{\mathrm{real},\pi}| \le \epsilon/2$. Union bounding over this for both $\pi \in \{\pi^{\widehat{f}}, \pi^{\mathrm{sim},\star}\}$, we have that with probability at least $1 - \delta/2$:

$$V_0^{\mathrm{real},\widehat{\pi}} \ge \widehat{V}_0^{\widehat{\pi}} - \epsilon/4 \ge \widehat{V}_0^{\widetilde{\pi}} - \epsilon/4 \ge V_0^{\mathrm{real},\widetilde{\pi}} - \epsilon/2.$$

It follows that

$$V^{\mathrm{real},\star} - V_0^{\mathrm{real},\widehat{\pi}} \le V^{\mathrm{real},\star} - V_0^{\mathrm{real},\widetilde{\pi}} + \epsilon/2 \le \epsilon.$$

The proof follows from a union bound and our condition on $T$ (note that (B.9) is satisfied in both cases).

$\square$

---

[6]Note that, while Lemma B.1 applies to the constrained regression setting, this is equivalent to the unconstrained regression setting considered here if we choose $\gamma$ large enough so that the constraint is vacuous.

*Proof of Theorem 1.* We first assume that $\zeta \leq \frac{\lambda^\star_{\min}}{4d}$, for $\zeta$ the input regularization value given to Algorithm 5 by Algorithm 1, and Condition 1 of Theorem 2, and show that in this case $A^{k^\star-2}$ is at most polynomial in problem parameters.

First, by Lemma C.7 we have that, under the assumption that $\zeta \leq \frac{\lambda^\star_{\min}}{4d}$, the policy $\pi^h_{\exp}$ given by the uniform mixture of policies returned by Algorithm 5 will, with probability at least $1 - \delta$, satisfy $\lambda_{\min}(\mathbf{\Lambda}^s_{\pi^h_{\exp}, h}) \geq \frac{\lambda^\star_{\min}}{8d}$ under Assumption 3. Plugging $\bar\lambda_{\min} \leftarrow \frac{\lambda^\star_{\min}}{8d}$ into Theorem 2, we have that $\xi = 2\sqrt{\frac{16dHA}{\lambda^\star_{\min}} \cdot \epsilon_{\text{sim}}}$. Now note that

$$A^{k^\star-2} \leq A^{\frac{\log_A 64HV_{\max}/\epsilon}{\log_A 1/\xi}} = \left(\frac{64HV_{\max}}{\epsilon}\right)^{1/\log_A 1/\xi}.$$

It then suffices that we show $1/\log_A 1/\xi \leq 1 \iff 1/A \geq \xi$. However, this is clearly met by our condition on $\epsilon_{\text{sim}}$. Thus, as long as

$$T \geq c \cdot \frac{V^6_{\max} H^6 d^2 \log(2HT|\mathcal{F}|/\delta)}{\epsilon^6 \epsilon^2_{\text{sim}}},$$

by Theorem 2 we have that $\widehat\pi$ is $\epsilon$-optimal.

Now, if $\epsilon_{\text{sim}} \leq \epsilon/4H^2$ and $T \geq \frac{16H^2 \log 4/\delta}{\epsilon}$, we also have that $\widehat\pi$ is $\epsilon$-optimal, by Theorem 2. Thus, in the first case, we at most will require

$$T \geq c \cdot \frac{V^6_{\max} H^{10} d^2 \log(2HT|\mathcal{F}|/\delta)}{\epsilon^8}$$

to produce a policy that is $\epsilon$-optimal, since otherwise we will be in the second case.

It remains to justify the assumption that $\zeta \leq \frac{\lambda^\star_{\min}}{4d}$. Note that the condition of (4.1) is only required in the first case. Furthermore, if $\epsilon_{\text{sim}} \leq \epsilon/4H^2$ we will be in the second case. Thus, in the first case, we will have

$$\frac{\epsilon}{4H^2} \leq \epsilon_{\text{sim}} \leq \frac{\lambda^\star_{\min}}{64dHA^3}.$$

Rearranging this we obtain that, to be in the first case, we have

$$\frac{16dA^3\epsilon}{H} \leq \lambda^\star_{\min}$$

By our choice of $\zeta = \frac{4A^3\epsilon}{H}$, we then have that $\zeta \leq \frac{\lambda^\star_{\min}}{4d}$. By Lemma C.7 and our choice of $\zeta$, we have that Oracle 4.2 is called at most $\text{poly}(d, H, \epsilon^{-1}, \log\frac{1}{\delta})$ times, and we call the oracle of Oracle 4.1 only $H$ times. The result the follows from a union bound and rescaling $\delta$. $\qquad\square$

## B.4 Reducing the Version Space

As we noted, in general, given that we do not assume that $\phi^r$ is unknown, $\log|\mathcal{F}|$ could be significantly greater than the dimension. One might hope that, given access to $\mathcal{M}^{\text{sim}}$, we can reduce this dependence somewhat. We next show that this is possible given access to the following *constrained* regression oracle.

**Oracle B.1** (Constrained Regression Oracle). We assume access to a regression oracle such that, for any $h$ and datasets $\{(s^t, a^t, y^t)\}^T_{t=1}$ and $\{(\widetilde{s}^t, \widetilde{a}^t, \widetilde{y}^t)\}^{\widetilde{T}}_{t=1}$, we can compute:

$$\widehat{f}_h = \arg\min_{f\in\mathcal{F}_h} \sum_{t=1}^{T}(f(s^t, a^t) - y^t)^2 \quad \text{s.t.} \quad \sum_{t=1}^{\widetilde{T}}(f(\widetilde{s}^t, \widetilde{a}^t) - \widetilde{y}^t)^2 \leq \gamma.$$

While in general the oracle of Oracle B.1 cannot be reduced to the oracle of Oracle 4.1, under certain conditions on $\mathcal{F}$ this is possible. Given this oracle, we have the following result.

**Theorem 3.** *Assume that $\epsilon_{\mathsf{sim}} \leq \frac{\lambda^\star_{\min}}{64 dH A^3}$. Then if*

$$T \geq \widetilde{\mathcal{O}}\left(\frac{d^2 H^{16}}{\epsilon^8} \cdot \log \frac{H|\widetilde{\mathcal{F}}|}{\delta}\right),$$

*with probability at least $1 - \delta$, Algorithm 4 returns policy $\widehat{\pi}$ such that $V_0^{\mathsf{real}, \pi^{\mathsf{real}, \star}} - V_0^{\mathsf{real}, \widehat{\pi}} \leq \epsilon$, where*

$$\widetilde{\mathcal{F}} := \left\{ f \in \mathcal{F} \ : \ \sup_\pi \left(\mathbb{E}^{\mathsf{sim}, \pi}[f_h(s_h, a_h) - \mathcal{T}^{\mathsf{sim}} f_{h+1}(s_h, a_h)]\right)^2 \leq \alpha \cdot \epsilon_{\mathsf{sim}}^2 \right\}$$

*for $\alpha = \widetilde{\mathcal{O}}(AdH^3 \cdot \log^2 \frac{\log |\mathcal{F}|/\delta}{\epsilon_{\mathsf{sim}}})$. Furthermore, the computation oracles of Oracle 4.2 and Oracle B.1 are called at most $\mathrm{poly}(d, A, H, \epsilon^{-1}, \log \frac{|\mathcal{F}|}{\delta})$ times.*

Theorem 3 shows that, rather than paying for the full complexity of $\mathcal{F}$, we can pay only for the subset of $\mathcal{F}$ that is Bellman-consistent on $\mathcal{M}^{\mathsf{sim}}$.

### B.4.1 Algorithm and Proof

---
**Algorithm 4** sim-to-real transfer via simulated exploration (SIM2EXPLORE)
---
1: **input:** tolerance $\epsilon$, confidence $\delta$, budget $T$, $Q$-value function class $\mathcal{F}$
2: $\Pi_{\mathsf{exp}}^h \leftarrow$ LEARNEXPPOLICIES$(\mathcal{M}^{\mathsf{sim}}, \delta, \frac{4A^3\epsilon}{H}, h)$ for all $h \in [H]$
3: $\iota \leftarrow \mathcal{O}(\log_2 \frac{V_{\max} AdH}{\epsilon})$
4: **for** $\ell = 1, 2, \ldots, \iota$ **do**
5: $\quad \bar{\epsilon}^\ell \leftarrow 2^{-\ell}, T^\ell \leftarrow T/2\iota, \gamma^\ell \leftarrow 10V_{\max}^2(\bar{\epsilon}^\ell)^2$
6: $\quad$ Run exploration procedure of Lemma C.3 with $\beta_\ell \leftarrow \frac{\gamma^\ell}{20V_{\max}^2 \log \frac{8H|\mathcal{F}|}{\delta}}$ to obtain $\mathfrak{D}_{\mathsf{sim}}^\ell$
7: $\quad \widehat{\pi}^\ell \leftarrow$ EXPLOREREAL $(\{\mathrm{unif}(\Pi_{\mathsf{exp}}^h)\}_{h \in [H]}, T^\ell, \mathfrak{D}_{\mathsf{sim}}^\ell, \gamma^\ell)$ (Algorithm 3)
8: $\quad \widehat{V}_0^{\widehat{\pi}^\ell} \leftarrow$ average return running $\widehat{\pi}^\ell$ in real $T^\ell/2$ times
9: **return** $\widehat{\pi} \leftarrow \arg\max_{\ell \in [\iota]} \widehat{V}_0^{\widehat{\pi}^\ell}$
---

**Theorem 4.** *Assume that one of the two conditions is met:*

1. *For each $h$, $\pi_{\mathsf{exp}}^h$ plays actions uniformly at random for $h' > h$,*

$$\lambda_{\min}\left(\mathbf{\Lambda}_{\pi_{\mathsf{exp}}^h, h}^{\mathsf{s}}\right) \geq \bar{\lambda}_{\min}, \tag{B.10}$$

*and*

$$T \geq c \cdot \frac{V_{\max}^4 H^4 d^2 A^{2(k^\star - 2)} \iota \log(16H|\widetilde{\mathcal{F}}|/\delta)}{\epsilon^4 \epsilon_{\mathsf{sim}}^2},$$

*for*

$$k^\star = \left\lceil \frac{\log_A \frac{64HV_{\max}}{\epsilon}}{\log_A 1/\xi} \right\rceil, \quad \xi = 2\sqrt{\frac{2HA}{\bar{\lambda}_{\min}} \cdot \epsilon_{\mathsf{sim}}}$$

*and*

$$\widetilde{\mathcal{F}} := \left\{ f \in \mathcal{F} \ : \ \sup_\pi \left(\mathbb{E}^{\mathsf{sim}, \pi}[f_h(s_h, a_h) - \mathcal{T}^{\mathsf{sim}} f_{h+1}(s_h, a_h)]\right)^2 \right.$$
$$\left. \leq c \left(\log \frac{\log \frac{32H|\mathcal{F}|}{\delta}}{V_{\max}\epsilon_{\mathsf{sim}}^2} + 1\right) AdHV_{\max}^2 \log \frac{48d \log \frac{32H|\mathcal{F}|}{\delta}}{V_{\max}\epsilon_{\mathsf{sim}}^2} \cdot \epsilon_{\mathsf{sim}}^2 \right\}.$$

2. *$\epsilon_{\mathsf{sim}} \leq \epsilon/16H^2$ and*

$$T \geq c \cdot \frac{H^2 \iota \log \frac{16\iota}{\delta}}{\epsilon^2}.$$

*Then with probability at least $1 - \delta$, Algorithm 4 returns a policy $\widehat{\pi}$ such that $V_0^{\mathsf{real}, \pi^{\mathsf{real}, \star}} - V_0^{\mathsf{real}, \widehat{\pi}} \leq \epsilon$.*

*Proof.* We break the proof into two cases.

**Case 1:** $\epsilon_{\text{sim}} \geq \epsilon/16H^2$. Let $\bar{\ell} = \lfloor \log_2 \epsilon_{\text{sim}}^{-1} \rfloor$ and note that $\bar{\ell} \leq \iota$ in this case and that this is a deterministic quantity. Further, note that $\gamma^{\bar{\ell}} \in [10V_{\text{max}}^2 \epsilon_{\text{sim}}^2, 40V_{\text{max}}^2 \epsilon_{\text{sim}}^2]$ and $\bar{\epsilon}^{\bar{\ell}} \in [\epsilon_{\text{sim}}, 2\epsilon_{\text{sim}}]$. Note that by our assumption on $\pi_{\text{exp}}$ and applying Lemma B.4 with $\kappa = \frac{\epsilon}{64HV_{\text{max}}}$ and $\gamma = \frac{d}{H\epsilon_{\text{sim}}}$, for any $\pi$ and $\mathcal{Z}' \subseteq \mathcal{S} \times \mathcal{A}$, we have

$$w_h^{\text{real},\pi}(\mathcal{Z}') \leq \frac{256 d H V_{\text{max}} A^{k^\star - 2}}{\epsilon \epsilon_{\text{sim}}} \cdot w_h^{\text{real},\pi_{\text{exp}}}(\mathcal{Z}') + \frac{\epsilon}{16HV_{\text{max}}}$$

for

$$k^\star = \lceil \frac{\log_A \frac{64HV_{\text{max}}}{\epsilon}}{\log_A 1/\xi} \rceil, \quad \xi = 2\sqrt{\frac{2HA}{\overline{\lambda}_{\min}} \cdot \epsilon_{\text{sim}}}.$$

By Lemma B.1, as long as $\beta^{\bar{\ell}}$ and $\gamma^{\bar{\ell}}$ satisfy

$$2V_{\text{max}}^2 \epsilon_{\text{sim}}^2 + \frac{43V_{\text{max}}^2 \beta_{\bar{\ell}}^2}{dH} \cdot \log \frac{8H|\mathcal{F}_h|}{\delta} + 6V_{\text{max}}^2 \beta_{\bar{\ell}} \sqrt{\frac{\log \frac{8H|\mathcal{F}_h|}{\delta}}{dH}} \leq \gamma^{\bar{\ell}}, \qquad \text{(B.11)}$$

we have that with probability at least $1 - 2\delta$,

$$V_0^{\text{real},\pi^{\text{real},\star}} - V_0^{\text{real},\widehat{\pi}^{\bar{\ell}}} \leq \frac{256 d H V_{\text{max}} A^{k^\star - 2}}{\epsilon \epsilon_{\text{sim}}} \cdot 4H\sqrt{\frac{256 V_{\text{max}}^2 \log(4H|\widetilde{\mathcal{F}}^{\bar{\ell}}|/\delta)}{T^\ell}} + \epsilon/4$$

where

$$\widetilde{\mathcal{F}}^{\bar{\ell}} := \{ f \in \mathcal{F} \ : \ \mathbb{E}^{\text{sim},\pi_{\text{exp}}^{\text{sim}}}[(f_h(s_h, a_h) - \mathcal{T}^{\text{sim}} f_{h+1}(s_h, a_h))^2] \leq 2\gamma^{\bar{\ell}}, \forall h \in [H] \}.$$

However, since $V_{\text{max}}^2 \epsilon_{\text{sim}} \leq \frac{1}{10}\gamma^{\bar{\ell}}$, and by our choice of $\beta^{\bar{\ell}} = \frac{\gamma^{\bar{\ell}}}{20V_{\text{max}}^2 \log \frac{8H|\mathcal{F}|}{\delta}}$, we see that (B.11) is met, so the conclusion holds. Note that, by Lemma C.5, we have that with probability at least $1 - \delta$:

$$\widetilde{\mathcal{F}}^{\bar{\ell}} \subseteq \left\{ f \in \mathcal{F} \ : \ \sup_\pi \left( \mathbb{E}^{\text{sim},\pi}[f_h(s_h,a_h) - \mathcal{T}^{\text{sim}} f_{h+1}(s_h,a_h)] \right)^2 \right.$$
$$\left. \leq \left( 4\log\frac{1}{\beta_{\bar{\ell}}} + 6 \right) A \cdot \left[ 48dH \log \frac{48d}{\beta_{\bar{\ell}}^2} \cdot 2\gamma^{\bar{\ell}} + V_{\text{max}}^2 \sqrt{96dH \log \frac{48d}{\beta_{\bar{\ell}}^2} \log \frac{1}{\delta}} \cdot \beta_{\bar{\ell}} \right] \right\}$$

$$\subseteq \left\{ f \in \mathcal{F} \ : \ \sup_\pi \left( \mathbb{E}^{\text{sim},\pi}[f_h(s_h,a_h) - \mathcal{T}^{\text{sim}} f_{h+1}(s_h,a_h)] \right)^2 \right.$$
$$\left. \leq c \left( \log \frac{\log \frac{8H|\mathcal{F}|}{\delta}}{V_{\text{max}} \epsilon_{\text{sim}}^2} + 1 \right) AdHV_{\text{max}}^2 \log \frac{48d \log \frac{8H|\mathcal{F}|}{\delta}}{V_{\text{max}} \epsilon_{\text{sim}}^2} \cdot \epsilon_{\text{sim}}^2 \right\}$$

$$=: \widetilde{\mathcal{F}}$$

where the second inclusion follows from our setting of $\beta_{\bar{\ell}}$, and bounds on $\gamma^{\bar{\ell}}$.

Since $T^\ell \leftarrow T/2\iota$, it follows that if

$$T \geq c \cdot \frac{d^2 H^4 V_{\text{max}}^4 A^{2(k^\star - 2)} \iota \log(4H|\widetilde{\mathcal{F}}|/\delta)}{\epsilon^4 \epsilon_{\text{sim}}^2},$$

then we have that $V_0^{\text{real},\pi^{\text{real},\star}} - V_0^{\text{real},\widehat{\pi}^{\bar{\ell}}} \leq \epsilon/2$.

**Case 2:** $\epsilon_{\text{sim}} \leq \epsilon/16H^2$. By Lemma B.5 and our choice of $T_{\text{sim}}^\ell$, we have that with probability at least $1 - \delta$,

$$V_0^{\text{real},\star} - V_0^{\text{real},\widehat{\pi}^\iota} \leq 6H\left( 2\log \frac{20V_{\text{max}}^2 \log \frac{8H|\mathcal{F}|}{\delta}}{\gamma^\iota} + 3 \right) \cdot \sqrt{192AdH \log \frac{960dV_{\text{max}}^2 \log \frac{8H|\mathcal{F}|}{\delta}}{\gamma^\iota} \cdot \gamma^\iota} + 4H^2 \epsilon_{\text{sim}}.$$

By our choice of $\iota = \mathcal{O}(\log_2 \frac{V_{\text{max}} AdH}{\epsilon})$ and since $\gamma^\iota = 10V_{\text{max}}^2(\bar{\epsilon}^\iota)^2 = 10V_{\text{max}}^2 \cdot 2^{-2\iota}$, we can bound $V_0^{\text{real},\star} - V_0^{\text{real},\widehat{\pi}^\iota} \leq \epsilon/2$.

**Completing the Proof.** In either case, we have that with probability at least $1 - \delta$, there exists some $\widehat{i} \in [\iota]$ such that $V_0^{\mathsf{real},\star} - V_0^{\mathsf{real},\widehat{\pi}^{\widehat{i}}} \le \epsilon/2$.

Note that $V_0^{\mathsf{real},\pi} = \mathbb{E}^{\mathsf{real},\pi}[\sum_{h=0}^{H-1} r_h]$ and that $\sum_{h=0}^{H-1} r_h \in [0, H]$ almost surely. Consider playing $\pi$ for $n$ episodes in real and let $R^i$ denote the total return of the $i$th episode. Let

$$\widehat{V}_0^\pi := \frac{1}{n} \sum_{i=1}^n R^i.$$

By Hoeffding's inequality we have that, with probability at least $1 - \delta/\iota$:

$$|\widehat{V}_0^\pi - V_0^{\mathsf{real},\pi}| \le H \sqrt{\frac{\log \frac{2\iota}{\delta}}{n}}.$$

Thus, if

$$n \ge \frac{16 H^2 \log \frac{2\iota}{\delta}}{\epsilon^2},$$

we have that $|\widehat{V}_0^\pi - V_0^{\mathsf{real},\pi}| \le \epsilon/2$. However, as we run each $\pi \in \widehat{\Pi}^\ell$ $T_\ell/2 = T/2\iota$ times, and in either case we assume $T \ge \frac{c\iota H^2}{\epsilon^2} \cdot \log \frac{4\iota}{\delta}$, this will be met. Union bounding over this for all $\widehat{\pi}^\ell$, we have that with probability at least $1 - \delta$:

$$V_0^{\mathsf{real},\widehat{\pi}} \ge \widehat{V}_0^{\widehat{\pi}} - \epsilon/4 \ge \widehat{V}_0^{\widehat{\pi}^{\widehat{i}}} - \epsilon/4 \ge V_0^{\mathsf{real},\widehat{\pi}^{\widehat{i}}} - \epsilon/2.$$

It follows that

$$V^{\mathsf{real},\star} - V_0^{\mathsf{real},\widehat{\pi}} \le V^{\mathsf{real},\star} - V_0^{\mathsf{real},\widehat{\pi}^{\widehat{i}}} + \epsilon/2 \le \epsilon.$$

The result then follows from a union bound and rescaling $\delta$. $\square$

*Proof of Theorem 3.* The argument follows analogously to the proof of Theorem 1, but using Theorem 4 in place of Theorem 2. The bound on the number of oracle calls follows from Lemma C.3 and our choice of $\beta_\ell$. $\square$

**Lemma B.5.** *With probability at least $1 - \delta$, for some $\ell$, we have*

$$V_0^{\mathsf{sim},\star} - V_0^{\mathsf{sim},\widehat{\pi}^\ell} \le 6H \left( 2 \log \frac{20 V_{\max}^2 \log \frac{8H|\mathcal{F}|}{\delta}}{\gamma^\ell} + 3 \right) \cdot \sqrt{192 A d H \log \frac{960 d V_{\max}^2 \log \frac{8H|\mathcal{F}|}{\delta}}{\gamma^\ell}} \cdot \gamma^\ell,$$

$$V_0^{\mathsf{real},\star} - V_0^{\mathsf{real},\widehat{\pi}^\ell} \le 6H \left( 2 \log \frac{20 V_{\max}^2 \log \frac{8H|\mathcal{F}|}{\delta}}{\gamma^\ell} + 3 \right) \cdot \sqrt{192 A d H \log \frac{960 d V_{\max}^2 \log \frac{8H|\mathcal{F}|}{\delta}}{\gamma^\ell}} \cdot \gamma^\ell + 4 H^2 \epsilon_{\mathsf{sim}}.$$

*Proof.* By Lemma C.4 we have, with probability at least $1 - \delta$,

$$V_0^{\mathsf{sim},\star} - V_0^{\mathsf{sim},\widehat{\pi}^\ell} \le 2H \left( 2 \log \frac{1}{\beta_\ell} + 3 \right) \cdot \left[ \beta_\ell \sqrt{512 V_{\max}^2 A \log \frac{8H|\mathcal{F}|}{\delta}} + \sqrt{96 A d H \log \frac{48d}{\beta_\ell^2} \cdot \gamma^\ell} \right.$$

$$\left. + \sqrt{2 A V_{\max}^2 \sqrt{96 d H \log \frac{48d}{\beta_\ell^2} \log \frac{2}{\delta}} \cdot \beta_\ell} \right]$$

$$\le 6H \left( 2 \log \frac{20 V_{\max}^2 \log \frac{8H|\mathcal{F}|}{\delta}}{\gamma^\ell} + 3 \right) \cdot \sqrt{192 A d H \log \frac{960 d V_{\max}^2 \log \frac{8H|\mathcal{F}|}{\delta}}{\gamma^\ell}} \cdot \gamma^\ell$$

where the second inequality holds by our setting of $\beta_\ell$.

We have

$$V_0^{\mathsf{real},\star} - V_0^{\mathsf{real},\widehat{\pi}^t} = V_0^{\mathsf{real},\star} - V_0^{\mathsf{real},\pi^{\mathsf{sim},\star}} + V_0^{\mathsf{real},\pi^{\mathsf{sim},\star}} - V_0^{\mathsf{sim},\pi^{\mathsf{sim},\star}} + V_0^{\mathsf{sim},\pi^{\mathsf{sim},\star}} - V_0^{\mathsf{sim},\widehat{\pi}^t} + V_0^{\mathsf{sim},\widehat{\pi}^t} - V_0^{\mathsf{real},\widehat{\pi}^t}.$$

By Lemma A.3, we can bound

$$V_0^{\text{real},\star} - V_0^{\text{real},\pi^{\text{sim},\star}} \leq 2H^2 \epsilon_{\text{sim}}$$

and by Lemma A.1 we can bound

$$V_0^{\text{real},\pi^{\text{sim},\star}} - V_0^{\text{sim},\pi^{\text{sim},\star}} \leq H^2 \epsilon_{\text{sim}}, \quad V_0^{\text{sim},\widehat{\pi}^\ell} - V_0^{\text{real},\widehat{\pi}^\ell} \leq H^2 \epsilon_{\text{sim}}.$$

Combining this with our bound on $V_0^{\text{sim},\star} - V_0^{\text{sim},\widehat{\pi}^\ell}$ gives the result. $\qquad\square$

# C  Learning in sim

In this section we provide additional supporting lemmas for our main results and in particular, we focus on linear in sim. In Appendix C.1 we provide several technical results critical to showing that sim can be utilized to restrict the version space, as is done in Theorem 4. In order to restrict the version space using sim, sufficiently rich data must be collected from sim, and in Appendix C.2 we provide results on this data collection. Finally, in Appendix C.3 we provide a procedure to compute the exploration policies in sim which we ultimately transfer to real.

In Appendices C.1 and C.2, we let hypothesis $\widetilde{f}$ and $\widehat{f}$ be defined recursively as:

$$\widetilde{f}_h := \arg\min_{f_h \in \mathcal{F}_h} \frac{1}{T_{\text{sim}}} \sum_{t=1}^{T_{\text{sim}}} (f_h(\widetilde{s}_h^t, \widetilde{a}_h^t) - \widetilde{r}_h^t - \max_{a'} \widehat{f}_{h+1}(\widetilde{s}_{h+1}^t, a'))^2.$$

and $\widehat{f}_h \in \mathcal{F}_h$ some hypothesis satisfying

$$\frac{1}{T_{\text{sim}}} \sum_{t=1}^{T_{\text{sim}}} (\widehat{f}_h(\widetilde{s}_h^t, \widetilde{a}_h^t) - \widetilde{f}_h(\widetilde{s}_h^t, \widetilde{a}_h^t))^2 \leq \gamma$$

for parameter $\gamma > 0$.

In Appendix C.1 we make the following assumption on the data generating process.

**Assumption 5.** *Consider the dataset* $\mathfrak{D}_{\text{sim}} = \{(\widetilde{s}_0^t, \widetilde{a}_0^t, \widetilde{r}_0^t, \ldots, \widetilde{s}_{H-1}^t, \widetilde{a}_{H-1}^t, \widetilde{r}_{H-1}^t)\}_{t=1}^{T_{\text{sim}}}$. *We assume that episode* $t$ *in* $\mathfrak{D}_{\text{sim}}$ *was generated by playing an* $\mathcal{F}_{t-1}$*-measurable policy* $\widetilde{\pi}_{\text{exp}}^t$*, and denote* $\pi_{\text{exp}}^{\text{sim}} = \text{unif}(\{\widetilde{\pi}_{\text{exp}}^t\}_{t=1}^{T_{\text{sim}}})$.

We provide a specific instantiation of $\pi_{\text{exp}}^{\text{sim}}$ in Appendix C.2. In Appendix C.3, we provide a procedure for learning a set of policies which induce full-rank covariates in sim, a crucial piece in obtaining good exploration performance in real.

## C.1  Regularizing with Data from sim

**Lemma C.1.** *With probability at least* $1 - \delta$:

$$\frac{1}{T_{\text{sim}}} \sum_{t=1}^{T_{\text{sim}}} (\mathcal{T}^{\text{real}} \widehat{f}_{h+1}(\widetilde{s}_h^t, \widetilde{a}_h^t) - \widetilde{f}_h(\widetilde{s}_h^t, \widetilde{a}_h^t))^2 \leq 2V_{\max}^2 \epsilon_{\text{sim}}^2 + \frac{512 V_{\max}^2}{T_{\text{sim}}} \cdot \log \frac{4|\mathcal{F}_h|}{\delta} + V_{\max}^2 \sqrt{\frac{2 \log \frac{2|\mathcal{F}_h|}{\delta}}{T_{\text{sim}}}}.$$

*Proof.* First, note that $\mathcal{T}^{\text{real}} \widehat{f}_{h+1} \in \mathcal{F}_h$ by Assumption 4.

By Azuma-Hoeffding and a union bound, we have that, with probability at least $1 - \delta$, for each $f, f' \in \mathcal{F}_h$,

$$\frac{1}{T_{\text{sim}}} \sum_{t=1}^{T_{\text{sim}}} (f_h(\widetilde{s}_h^t, \widetilde{a}_h^t) - f_h'(\widetilde{s}_h^t, \widetilde{a}_h^t))^2 \leq \frac{1}{T_{\text{sim}}} \sum_{t=1}^{T_{\text{sim}}} \mathbb{E}^{\text{sim},\widetilde{\pi}_{\text{exp}}^t}[(f_h(\widetilde{s}_h, \widetilde{a}_h) - f_h'(\widetilde{s}_h, \widetilde{a}_h))^2] + V_{\max}^2 \sqrt{\frac{2 \log |\mathcal{F}_h|/\delta}{T_{\text{sim}}}}$$

$$= \mathbb{E}^{\text{sim},\pi_{\text{exp}}^{\text{sim}}}[(f_h(s_h, a_h) - f_h'(s_h, a_h))^2] + V_{\max}^2 \sqrt{\frac{2 \log |\mathcal{F}_h|/\delta}{T_{\text{sim}}}}.$$

In particular, this implies that

$$\frac{1}{T_{\mathsf{sim}}} \sum_{t=1}^{T_{\mathsf{sim}}} (\mathcal{T}^{\mathsf{real}} \widehat{f}_{h+1}(\widetilde{s}_h^t, \widetilde{a}_h^t) - \widetilde{f}_h(\widetilde{s}_h^t, \widetilde{a}_h^t))^2 \le \mathbb{E}^{\mathsf{sim}, \pi_{\mathsf{exp}}^{\mathsf{sim}}} [(\mathcal{T}^{\mathsf{real}} \widehat{f}_{h+1}(s_h, a_h) - \widetilde{f}_h(s_h, a_h))^2] + V_{\max}^2 \sqrt{\frac{2 \log |\mathcal{F}_h|/\delta}{T_{\mathsf{sim}}}}.$$

We can bound

$$\mathbb{E}^{\mathsf{sim}, \pi_{\mathsf{exp}}^{\mathsf{sim}}} [(\mathcal{T}^{\mathsf{real}} \widehat{f}_{h+1}(s_h, a_h) - \widetilde{f}_h(s_h, a_h))^2] \le 2 \underbrace{\mathbb{E}^{\mathsf{sim}, \pi_{\mathsf{exp}}^{\mathsf{sim}}} [(\mathcal{T}^{\mathsf{real}} \widehat{f}_{h+1}(s_h, a_h) - \mathcal{T}^{\mathsf{sim}} \widehat{f}_{h+1}(s_h, a_h))^2]}_{(a)}$$

$$+ 2 \underbrace{\mathbb{E}^{\mathsf{sim}, \pi_{\mathsf{exp}}^{\mathsf{sim}}} [(\mathcal{T}^{\mathsf{sim}} \widehat{f}_{h+1}(s_h, a_h) - \widetilde{f}_h(s_h, a_h))^2]}_{(b)}.$$

To bound $(a)$, we note that

$$\mathcal{T}^{\mathsf{real}} \widehat{f}_{h+1}(s_h, a_h) - \widetilde{f}_h(s_h, a_h) = \mathbb{E}^{\mathsf{real}} [\max_{a'} \widehat{f}_{h+1}(s', a') \mid s, a] - \mathbb{E}^{\mathsf{sim}} [\max_{a'} \widehat{f}_{h+1}(s', a') \mid s, a]$$

$$= \sum_{s'} (P_h^{\mathsf{real}}(s' \mid s, a) - P_h^{\mathsf{sim}}(s' \mid s, a)) \cdot \max_{a'} \widehat{f}_{h+1}(s', a')$$

$$\le V_{\max} \cdot \sum_{s'} |P_h^{\mathsf{real}}(s' \mid s, a) - P_h^{\mathsf{sim}}(s' \mid s, a)|$$

$$\le V_{\max} \epsilon_{\mathsf{sim}}$$

where the last inequality follows under Assumption 1. This gives that $(a) \le 2 V_{\max}^2 \epsilon_{\mathsf{sim}}^2$. To bound $(b)$, we apply Lemma 3 of [57], which gives that with probability at least $1 - \delta$,

$$(b) \le \frac{512 V_{\max}^2}{T_{\mathsf{sim}}} \cdot \log \frac{4|\mathcal{F}_h|}{\delta}.$$

Combining these with a union bound gives the result. $\qquad \square$

**Lemma C.2.** *Consider the set*

$$\widehat{\mathcal{F}}_h := \left\{ f_h \in \mathcal{F}_h \ : \ \frac{1}{T_{\mathsf{sim}}} \sum_{t=1}^{T_{\mathsf{sim}}} (f_h(\widetilde{s}_h^t, \widetilde{a}_h^t) - \widetilde{f}_h(\widetilde{s}_h^t, \widetilde{a}_h^t))^2 \le \gamma \right\}.$$

*Then with probability $1 - 2\delta$ we have*

$$\widehat{\mathcal{F}}_h \subseteq \left\{ f_h \in \mathcal{F}_h \ : \ \mathbb{E}^{\mathsf{sim}, \pi_{\mathsf{exp}}^{\mathsf{sim}}} [(f_h(s_h, a_h) - \mathcal{T}^{\mathsf{sim}} \widehat{f}_{h+1}(s_h, a_h))^2] \le \gamma + 18 V_{\max}^2 \sqrt{\frac{\log \frac{4|\mathcal{F}_h|}{\delta}}{T_{\mathsf{sim}}}} \right\}.$$

*Proof.* By Azuma-Hoeffding, we have that with probability at least $1 - \delta$, for each $f_h, f_h' \in \mathcal{F}_h$,

$$\mathbb{E}^{\mathsf{sim}, \pi_{\mathsf{exp}}^{\mathsf{sim}}} [(f_h(s_h, a_h) - f_h'(s_h, a_h))^2] - V_{\max}^2 \sqrt{\frac{2 \log |\mathcal{F}_h|/\delta}{T_{\mathsf{sim}}}} \le \frac{1}{T_{\mathsf{sim}}} \sum_{t=1}^{T_{\mathsf{sim}}} (f_h(\widetilde{s}_h^t, \widetilde{a}_h^t) - f_h'(\widetilde{s}_h^t, \widetilde{a}_h^t))^2$$

which implies in particular that, for any $f_h \in \mathcal{F}_h$,

$$\mathbb{E}^{\mathsf{sim}, \pi_{\mathsf{exp}}^{\mathsf{sim}}} [(f_h(s_h, a_h) - \widetilde{f}_h(s_h, a_h))^2] - V_{\max}^2 \sqrt{\frac{2 \log |\mathcal{F}_h|/\delta}{T_{\mathsf{sim}}}} \le \frac{1}{T_{\mathsf{sim}}} \sum_{t=1}^{T_{\mathsf{sim}}} (f_h(\widetilde{s}_h^t, \widetilde{a}_h^t) - \widetilde{f}_h(\widetilde{s}_h^t, \widetilde{a}_h^t))^2.$$

We can write

$$\mathbb{E}^{\mathsf{sim}, \pi_{\mathsf{exp}}^{\mathsf{sim}}} [(f_h(s_h, a_h) - \widetilde{f}_h(s_h, a_h))^2]$$

$$= \mathbb{E}^{\mathsf{sim}, \pi_{\mathsf{exp}}^{\mathsf{sim}}} [(f_h(s_h, a_h) - \mathcal{T}^{\mathsf{sim}} \widehat{f}_{h+1}(s_h, a_h))^2] + \mathbb{E}^{\mathsf{sim}, \pi_{\mathsf{exp}}^{\mathsf{sim}}} [(\widetilde{f}_h(s_h, a_h) - \mathcal{T}^{\mathsf{sim}} \widehat{f}_{h+1}(s_h, a_h))^2]$$

$$- 2 \mathbb{E}^{\mathsf{sim}, \pi_{\mathsf{exp}}^{\mathsf{sim}}} [(\widetilde{f}_h(s_h, a_h) - \mathcal{T}^{\mathsf{sim}} \widehat{f}_{h+1}(s_h, a_h))(f_h(s_h, a_h) - \mathcal{T}^{\mathsf{sim}} \widehat{f}_{h+1}(s_h, a_h))]$$

$$\ge \mathbb{E}^{\mathsf{sim}, \pi_{\mathsf{exp}}^{\mathsf{sim}}} [(f_h(s_h, a_h) - \mathcal{T}^{\mathsf{sim}} \widehat{f}_{h+1}(s_h, a_h))^2]$$

$$- 2 \mathbb{E}^{\mathsf{sim}, \pi_{\mathsf{exp}}^{\mathsf{sim}}} [(\widetilde{f}_h(s_h, a_h) - \mathcal{T}^{\mathsf{sim}} \widehat{f}_{h+1}(s_h, a_h))(f_h(s_h, a_h) - \mathcal{T}^{\mathsf{sim}} \widehat{f}_{h+1}(s_h, a_h))].$$

By Lemma 3 of [57], with probability at least $1 - \delta$,

$$\mathbb{E}^{\mathsf{sim}, \pi_{\exp}^{\mathsf{sim}}}[(\widetilde{f}_h(s_h, a_h) - \mathcal{T}^{\mathsf{sim}}\widehat{f}_{h+1}(s_h, a_h))^2] \leq \frac{256V_{\max}^2}{T_{\mathsf{sim}}} \cdot \log \frac{2|\mathcal{F}_h|}{\delta}.$$

We can therefore bound the final term as

$$\mathbb{E}^{\mathsf{sim}, \pi_{\exp}^{\mathsf{sim}}}[(\widetilde{f}_h(s_h, a_h) - \mathcal{T}^{\mathsf{sim}}\widehat{f}_{h+1}(s_h, a_h))(f_h(s_h, a_h) - \mathcal{T}^{\mathsf{sim}}\widehat{f}_{h+1}(s_h, a_h))]$$

$$\leq V_{\max} \cdot \mathbb{E}^{\mathsf{sim}, \pi_{\exp}^{\mathsf{sim}}}[|\widetilde{f}_h(s_h, a_h) - \mathcal{T}^{\mathsf{sim}}\widehat{f}_{h+1}(s_h, a_h)|]$$

$$\leq V_{\max} \cdot \sqrt{\mathbb{E}^{\mathsf{sim}, \pi_{\exp}^{\mathsf{sim}}}[(\widetilde{f}_h(s_h, a_h) - \mathcal{T}^{\mathsf{sim}}\widehat{f}_{h+1}(s_h, a_h))^2]}$$

$$\leq V_{\max} \cdot \sqrt{\frac{256V_{\max}^2}{T_{\mathsf{sim}}} \cdot \log \frac{2|\mathcal{F}_h|}{\delta}}.$$

Altogether then we have shown that, for any $f_h \in \mathcal{F}_h$, with probability at least $1 - 2\delta$:

$$\frac{1}{T_{\mathsf{sim}}} \sum_{t=1}^{T_{\mathsf{sim}}} (f_h(\widetilde{s}_h^t, \widetilde{a}_h^t) - \widetilde{f}_h(\widetilde{s}_h^t, \widetilde{a}_h^t))^2 \geq \mathbb{E}^{\mathsf{sim}, \pi_{\exp}^{\mathsf{sim}}}[(f_h(s_h, a_h) - \mathcal{T}^{\mathsf{sim}}\widehat{f}_{h+1}(s_h, a_h))^2] - 18V_{\max}^2 \sqrt{\frac{\log 2|\mathcal{F}_h|/\delta}{T_{\mathsf{sim}}}}.$$

Thus, if

$$\frac{1}{T_{\mathsf{sim}}} \sum_{t=1}^{T_{\mathsf{sim}}} (f_h(\widetilde{s}_h^t, \widetilde{a}_h^t) - \widetilde{f}_h(\widetilde{s}_h^t, \widetilde{a}_h^t))^2 \leq \gamma,$$

then

$$\mathbb{E}^{\mathsf{sim}, \pi_{\exp}^{\mathsf{sim}}}[(f_h(s_h, a_h) - \mathcal{T}^{\mathsf{sim}}\widehat{f}_{h+1}(s_h, a_h))^2] \leq \gamma + 18V_{\max}^2 \sqrt{\frac{\log 2|\mathcal{F}_h|/\delta}{T_{\mathsf{sim}}}}.$$

The result follows from a union bound. $\qquad \square$

## C.2 Data Collection with CoverTraj

**Lemma C.3.** *Consider running the* COVERTRAJ *algorithm of [65] for each* $h \in [H]$ *with parameters* $m \leftarrow \lceil \log_2 1/\beta \rceil$ *and* $\gamma_i \leftarrow 2^i \cdot \beta$ *for some* $\beta \in [0, 1]$*, and with* REGMIN *set to the policy optimization oracle of Oracle 4.2. Then this procedure collects*

$$T_{\mathsf{sim}} := H \cdot \sum_{i=1}^{m} \left\lceil \frac{24d}{2^i \cdot \beta^2} \log \frac{48d}{2^i \cdot \beta^2} \right\rceil$$

*episodes, calls the policy optimization oracle at most* $T_{\mathsf{sim}}$ *times, and produces covariates* $\mathbf{\Lambda}_{h,i}$ *and sets* $\mathcal{X}_{h,i}$ *such that, for each* $i \in [m]$*,*

$$\sup_{\pi} w_h^{\mathsf{sim}, \pi}(\mathcal{X}_{h,i}) \leq 2^{-i+1} \quad and \quad \phi^\top \mathbf{\Lambda}_{h,i}^{-1} \phi \leq 2^{2i} \cdot \beta^2, \forall \phi \in \mathcal{X}_{h,i},$$

*and* $\sup_{\pi} w_h^{\mathsf{sim}, \pi}(\mathcal{B}^d \setminus \cup_{i=1}^m \mathcal{X}_{h,i}) \leq \beta$*. Furthermore, we have*

$$\frac{12dH}{\beta^2} \leq T_{\mathsf{sim}} \leq \frac{48dH}{\beta^2} \log \frac{48d}{\beta^2}.$$

*Proof.* Instantiating REGMIN with the oracle of Oracle 4.2, we have that Definition 5.1 of [65] is met with $\mathcal{C}_1 = \mathcal{C}_2 = 0$. Therefore, we have that at each stage $i$ we collect exactly (using the precise form for $K_i$ given in the appendix of [65])

$$K_i = \lceil 2^i \cdot \frac{24d}{\gamma_i^2} \log \frac{48 \cdot 2^i d}{\gamma_i^2} \rceil$$

episodes. The result then follows by Theorem 3 of [65]. $\qquad \square$

**Lemma C.4.** *Consider running the procedure of Lemma C.3 to collect data. Then with probability at least $1 - 2\delta$, we have*

$$V_0^{\mathsf{sim},\star} - V_0^{\mathsf{sim},\pi^{\widehat{f}}} \le 2H \left( 2\log\frac{1}{\beta} + 3 \right) \cdot \left[ \beta\sqrt{512V_{\max}^2 A \log(4H|\mathcal{F}|/\delta)} + \sqrt{96AdH\log\frac{48d}{\beta^2} \cdot \gamma} \right.$$

$$\left. + \sqrt{2AV_{\max}^2\sqrt{96dH\log\frac{48d}{\beta^2}\log\frac{1}{\delta}} \cdot \beta} \right].$$

*Proof.* By Lemma A.4:

$$V_0^{\mathsf{sim},\star} - V_0^{\mathsf{sim},\pi^{\widehat{f}}} \le \max_{\pi\in\{\pi^{\widehat{f}},\pi^{\mathsf{sim},\star}\}} \sum_{h=0}^{H-1} 2\left| \mathbb{E}^{\mathsf{sim},\pi}[\widehat{f}_h(s_h,a_h) - \mathcal{T}^{\mathsf{sim}}\widehat{f}_{h+1}(s_h,a_h)] \right|$$

$$\le \max_{\pi\in\{\pi^{\widehat{f}},\pi^{\mathsf{sim},\star}\}} \sum_{h=0}^{H-1} 2\mathbb{E}^{\mathsf{sim},\pi}[|\widehat{f}_h(s_h,a_h) - \mathcal{T}^{\mathsf{sim}}\widehat{f}_{h+1}(s_h,a_h)|].$$

Denote $g(z_h) := |\widehat{f}_h(s_h,a_h) - \mathcal{T}^{\mathsf{sim}}\widehat{f}_{h+1}(s_h,a_h)|$ and $\mathbf{\Lambda}_{h-1} = \sum_{i=1}^m \mathbf{\Lambda}_{h,i} + I$, for $\mathbf{\Lambda}_{h,i}$ collected as in Lemma C.3, and note that

$$\mathbb{E}^{\mathsf{sim},\pi}[g(z_h)] = \mathbb{E}^{\mathsf{sim},\pi}\left[\int g(z)\mathrm{d}P_h^\pi(z \mid z_{h-1})\right]$$

$$= \mathbb{E}^{\mathsf{sim},\pi}\left[\int\int g(z)\pi(a \mid s)\mathrm{d}a\mathrm{d}\boldsymbol{\mu}_{h-1}^{\mathsf{s}}(s)^\top\boldsymbol{\phi}^{\mathsf{s}}(z_{h-1})\right]$$

$$= \mathbb{E}^{\mathsf{sim},\pi}\left[\int\int g(z)\pi(a \mid s)\mathrm{d}a\mathrm{d}\boldsymbol{\mu}_{h-1}^{\mathsf{s}}(s)^\top\mathbf{\Lambda}_{h-1}^{1/2}\mathbf{\Lambda}_{h-1}^{-1/2}\boldsymbol{\phi}^{\mathsf{s}}(z_{h-1})\right] \qquad \text{(C.1)}$$

$$\le \mathbb{E}^{\mathsf{sim},\pi}\left[\|\int\int g(z)\pi(a \mid s)\mathrm{d}a\mathrm{d}\boldsymbol{\mu}_{h-1}^{\mathsf{s}}(s)\|_{\mathbf{\Lambda}_{h-1}} \cdot \|\boldsymbol{\phi}^{\mathsf{s}}(z_{h-1})\|_{\mathbf{\Lambda}_{h-1}^{-1}}\right]$$

$$= \|\int\int g(z)\pi(a \mid s)\mathrm{d}a\mathrm{d}\boldsymbol{\mu}_{h-1}^{\mathsf{s}}(s)\|_{\mathbf{\Lambda}_{h-1}} \cdot \mathbb{E}^{\mathsf{sim},\pi}[\|\boldsymbol{\phi}^{\mathsf{s}}(z_{h-1})\|_{\mathbf{\Lambda}_{h-1}^{-1}}].$$

We bound each of these terms separately. First, we have

$$\mathbb{E}^{\mathsf{sim},\pi}[\|\boldsymbol{\phi}^{\mathsf{s}}(z_{h-1})\|_{\mathbf{\Lambda}_{h-1}^{-1}}] \le \sum_{i=1}^m \max_{\boldsymbol{\phi}\in\mathcal{X}_{h-1,i}}\|\boldsymbol{\phi}\|_{\mathbf{\Lambda}_{h-1}^{-1}} \cdot \sup_\pi \mathbb{E}^{\mathsf{sim},\pi}[\mathbb{I}\{\boldsymbol{\phi}^{\mathsf{s}}(z_{h-1}) \in \mathcal{X}_{h-1,i}\}]$$

$$+ \max_{\boldsymbol{\phi}\in\mathcal{B}^d\setminus\cup_{i=1}^m\mathcal{X}_{h-1,i}}\|\boldsymbol{\phi}\|_{\mathbf{\Lambda}_{h-1}^{-1}} \cdot \sup_\pi \mathbb{E}^{\mathsf{sim},\pi}[\mathbb{I}\{\boldsymbol{\phi}^{\mathsf{s}}(z_{h-1}) \in \mathcal{X}_{h-1,i}\}]$$

$$\overset{(a)}{\le} \sum_{i=1}^\iota \gamma_i \cdot 2^{-i+1} + \beta$$

$$\le (2m+1)\beta$$

where $(a)$ follows from Lemma C.3 and since $\|\boldsymbol{\phi}\|_{\mathbf{\Lambda}_{h-1}^{-1}} \le 1$ always.

We turn now to bounding the first term. Note that

$$\| \int \int g(z) \pi(a \mid s) \mathrm{d}a \mathrm{d}\boldsymbol{\mu}^{\mathfrak{s}}_{h-1}(s) \|_{\boldsymbol{\Lambda}_{h-1}}$$

$$= \sqrt{ \sum_{t=1}^{T_{\mathsf{sim}}} ( \int \int g(z) \pi(a \mid s) \mathrm{d}a \mathrm{d}\boldsymbol{\mu}^{\mathfrak{s}}_{h-1}(s)^{\top} \boldsymbol{\phi}^t_{h-1} )^2 }$$

$$= \sqrt{ \sum_{t=1}^{T_{\mathsf{sim}}} \mathbb{E}^{\pi}[g(z_h) \mid z^t_{h-1}]^2 }$$

$$\leq \sqrt{ \sum_{t=1}^{T_{\mathsf{sim}}} \mathbb{E}^{\pi}[g(z_h)^2 \mid z^t_{h-1}] }$$

$$\overset{(a)}{\leq} \sqrt{ A \cdot \sum_{t=1}^{T_{\mathsf{sim}}} \mathbb{E}^{\pi^{h-1,t}_{\exp}}[g(z_h)^2 \mid z^t_{h-1}] }$$

$$= \sqrt{ A \cdot \sum_{t=1}^{T_{\mathsf{sim}}} \mathbb{E}^{\pi^{h-1,t}_{\exp}}[(\widehat{f}_h(s_h, a_h) - \mathcal{T}^{\mathsf{sim}}\widehat{f}_{h+1}(s_h, a_h))^2 \mid z^t_{h-1}] }$$

$$\leq \sqrt{ 2A \cdot \sum_{t=1}^{T_{\mathsf{sim}}} \mathbb{E}^{\pi^{h-1,t}_{\exp}}[(\widetilde{f}_h(s_h, a_h) - \mathcal{T}^{\mathsf{sim}}\widehat{f}_{h+1}(s_h, a_h))^2 \mid z^t_{h-1}] + 2A \cdot \sum_{t=1}^{T_{\mathsf{sim}}} \mathbb{E}^{\pi^{h-1,t}_{\exp}}[(\widetilde{f}_h(s_h, a_h) - \widehat{f}_h(s_h, a_h))^2 \mid z^t_{h-1}] }$$

$$\overset{(b)}{\leq} \sqrt{ 512 V^2_{\max} A \log(4H|\mathcal{F}|/\delta) + 2A \cdot \sum_{t=1}^{T_{\mathsf{sim}}} \mathbb{E}^{\pi^{h-1,t}_{\exp}}[(\widetilde{f}_h(s_h, a_h) - \widehat{f}_h(s_h, a_h))^2 \mid z^t_{h-1}] }$$

where $(a)$ uses the fact that $\pi^{h-1,t}_{\exp}$ plays actions randomly at step $h$ and $(b)$ holds with probability at least $1 - \delta$ by Lemma C.6. By Azuma-Hoeffding, we have with probability $1 - \delta$:

$$\sum_{t=1}^{T_{\mathsf{sim}}} \mathbb{E}^{\pi^{h-1,t}_{\exp}}[(\widetilde{f}_h(s_h, a_h) - \widehat{f}_h(s_h, a_h))^2 \mid z^t_{h-1}] \leq \sum_{t=1}^{T_{\mathsf{sim}}} (\widetilde{f}_h(s^t_h, a^t_h) - \widehat{f}_h(s^t_h, a^t_h))^2 + \sqrt{2V^4_{\max} T_{\mathsf{sim}} \log 1/\delta}$$

$$\leq T_{\mathsf{sim}}\gamma + \sqrt{2V^4_{\max} T_{\mathsf{sim}} \log 1/\delta}$$

where the last inequality follows from the definition of $\widehat{f}_h$.

Altogether then we have shown that, with probability at least $1 - 2\delta$:

$$V^{\mathsf{sim},\star}_0 - V^{\mathsf{sim},\pi^{\widehat{f}}}_0 \leq 2H(2m+1)\beta \cdot \sqrt{512 V^2_{\max} A \log(4H|\mathcal{F}|/\delta) + 2AT_{\mathsf{sim}}\gamma + 2AV^2_{\max}\sqrt{2T_{\mathsf{sim}} \log 1/\delta}}.$$

Using that $T_{\mathsf{sim}} \leq \frac{48dH}{\beta^2} \log \frac{48d}{\beta^2}$ as given in Lemma C.3, we can bound this as

$$\leq 2H(2m+1) \left[ \beta \sqrt{512 V^2_{\max} A \log(4H|\mathcal{F}|/\delta)} + \sqrt{96AdH \log \frac{48d}{\beta^2} \cdot \gamma} \right.$$

$$\left. + \sqrt{2AV^2_{\max}\sqrt{96dH \log \frac{48d}{\beta^2} \log \frac{1}{\delta}} \cdot \beta} \right].$$

The result follows. □

**Lemma C.5.** *Assume that*

$$\mathbb{E}^{\mathsf{sim}, \pi^{\mathsf{sim}}_{\exp}}[(f_h(s_h, a_h) - \mathcal{T}^{\mathsf{sim}} f_{h+1}(s_h, a_h))^2] \leq \gamma.$$

*Then this implies that, with probability at least $1 - \delta$,*

$$\sup_{\pi} \left( \mathbb{E}^{\mathsf{sim},\pi}[f_h(s_h, a_h) - \mathcal{T}^{\mathsf{sim}} f_{h+1}(s_h, a_h)] \right)^2$$

$$\leq \left( 4 \log \frac{1}{\beta} + 6 \right) A \cdot \left[ 48d \log \frac{48d}{\beta^2} \cdot \gamma + V_{\max} \sqrt{96d \log \frac{48d}{\beta^2} \log \frac{1}{\delta}} \cdot \beta \right].$$

*Therefore,*

$$\{ f \in \mathcal{F} \ : \ \mathbb{E}^{\mathsf{sim},\pi_{\exp}^{\mathsf{sim}}}[(f_h(s_h, a_h) - \mathcal{T}^{\mathsf{sim}} f_{h+1}(s_h, a_h))^2] \leq \gamma \}$$

$$\subseteq \left\{ f \in \mathcal{F} \ : \ \sup_{\pi} \left( \mathbb{E}^{\mathsf{sim},\pi}[f_h(s_h, a_h) - \mathcal{T}^{\mathsf{sim}} f_{h+1}(s_h, a_h)] \right)^2 \right.$$

$$\left. \leq \left( 4 \log \frac{1}{\beta} + 6 \right) A \cdot \left[ 48dH \log \frac{48d}{\beta^2} \cdot \gamma + V_{\max}^2 \sqrt{96dH \log \frac{48d}{\beta^2} \log \frac{1}{\delta}} \cdot \beta \right] \right\}.$$

*Proof.* We follow a similar argument as the proof of Lemma C.4. Denoting $g(z_h) := f_h(s_h, a_h) - \mathcal{T}^{\mathsf{sim}} f_{h+1}(s_h, a_h)$, by the same calculation as (C.1) we have

$$\mathbb{E}^{\mathsf{sim},\pi}[g(z_h)] \leq \| \int \int g(z) \pi(a \mid s) \mathrm{d}a \mathrm{d}\boldsymbol{\mu}_{h-1}^{\mathsf{s}}(s) \|_{\boldsymbol{\Lambda}_{h-1}} \cdot \mathbb{E}^{\mathsf{sim},\pi}[\| \boldsymbol{\phi}^{\mathsf{s}}(z_{h-1}) \|_{\boldsymbol{\Lambda}_{h-1}^{-1}}]$$

and as in the proof of Lemma C.4, we can bound

$$\mathbb{E}^{\mathsf{sim},\pi}[\| \boldsymbol{\phi}^{\mathsf{s}}(z_{h-1}) \|_{\boldsymbol{\Lambda}_{h-1}^{-1}}] \leq (2m+1)\beta$$

and

$$\| \int \int g(z) \pi(a \mid s) \mathrm{d}a \mathrm{d}\boldsymbol{\mu}_{h-1}^{\mathsf{s}}(s) \|_{\boldsymbol{\Lambda}_{h-1}} \leq \sqrt{A \cdot \sum_{t=1}^{T_{\mathsf{sim}}} \mathbb{E}^{\pi_{\exp}^{h-1,t}}[(f_h(s_h, a_h) - \mathcal{T}^{\mathsf{sim}} f_{h+1}(s_h, a_h))^2 \mid z_{h-1}^t]}$$

By Azuma-Hoeffding, with probability at least $1 - \delta$ we can then bound

$$\sum_{t=1}^{T_{\mathsf{sim}}} \mathbb{E}^{\pi_{\exp}^{h-1,t}}[(f_h(s_h, a_h) - \mathcal{T}^{\mathsf{sim}} f_{h+1}(s_h, a_h))^2 \mid z_{h-1}^t] \leq T_{\mathsf{sim}} \cdot \mathbb{E}^{\pi_{\exp}^{\mathsf{sim}}}[(f_h(s_h, a_h) - \mathcal{T}^{\mathsf{sim}} f_{h+1}(s_h, a_h))^2]$$

$$+ \sqrt{2V_{\max}^4 T_{\mathsf{sim}} \log 1/\delta}$$

$$\leq T_{\mathsf{sim}} \gamma + \sqrt{2V_{\max}^4 T_{\mathsf{sim}} \log 1/\delta}$$

where the last inequality follows by assumption, and where $\pi_{\exp}^{\mathsf{sim}} = \mathrm{unif}(\{\pi_{\exp}^{h-1,t}\}_{t=1}^{T_{\mathsf{sim}}})$. Altogether then, for all $\pi$, we have

$$\mathbb{E}^{\mathsf{sim},\pi}[f_h(s_h, a_h) - \mathcal{T}^{\mathsf{sim}} f_{h+1}(s_h, a_h)] \leq (2m+1)\beta \cdot \sqrt{AT_{\mathsf{sim}}\gamma + AV_{\max}^2 \sqrt{2T_{\mathsf{sim}} \log 1/\delta}}.$$

Using that $T_{\mathsf{sim}} \leq \frac{48dH}{\beta^2} \log \frac{48d}{\beta^2}$ as given in Lemma C.3, we can bound this as

$$\leq (2m+1)\sqrt{48AdH \log \frac{48d}{\beta^2} \cdot \gamma} + (2m+1)\sqrt{AV_{\max}^2 \sqrt{96dH \log \frac{48d}{\beta^2} \log \frac{1}{\delta}} \cdot \beta}.$$

The result follows from some algebra. $\qquad \square$

**Lemma C.6.** *With probability at least $1 - \delta$, for each $h \in [H]$ simultaneously, we have*

$$\sum_{t=1}^{T_{\mathsf{sim}}} \mathbb{E}^{\mathsf{sim},\pi_{\exp}^{h-1,t}}[(\widetilde{f}_h(s_h, a_h) - \mathcal{T}^{\mathsf{sim}} \widehat{f}_{h+1}(s_h, a_h))^2 \mid s_{h-1}^t, a_{h-1}^t] \leq 256V_{\max}^2 \log(4H|\mathcal{F}|/\delta).$$

*Proof.* This follows from Lemma 3 of [57]. $\qquad \square$

**Algorithm 5** Learn Exploration Policies in $\mathcal{M}^{\mathsf{sim}}$ (LEARNEXPPOLICIES)

---

1: **input:** environment $\mathcal{M}$, confidence $\delta$, regularization $\zeta$, step $h$
2: $\mathbb{A}_{\mathcal{R}} \leftarrow$ policy optimization oracle of Oracle 4.2
3: **for** $j = 1, 2, 3, \ldots, \mathcal{O}(\log_2(\frac{d}{\zeta} \cdot \log \frac{1}{\delta} + \zeta^{-9} \cdot \log^{3/2} \frac{1}{\delta}))$ **do**
4:     $N_j \leftarrow \lceil 2^{j/3} \rceil - 1, K_j \leftarrow \lceil 2^{2j/3} \rceil, T_j \leftarrow (N_j + 1)K_j, \delta_j \leftarrow \frac{\delta}{4j^2}$
    // `DynamicOED algorithm from [64]`
5:     $\boldsymbol{\Sigma}_j, \Pi_j \leftarrow$ DYNAMICOED$(\Phi, N_j, K_j, \delta_j, \mathbb{A}_{\mathcal{R}})$ for $\Phi(\boldsymbol{\Lambda}_h) \leftarrow \mathrm{tr}((\boldsymbol{\Lambda}_h + \zeta \cdot I)^{-1})$
6:     **if** $\lambda_{\min}(\boldsymbol{\Sigma}_j) \geq 12544 d \log \frac{4 + 64T_j}{\delta}$ and $T_j \geq c \cdot \zeta^{-9} \cdot \log^{3/2} \frac{jT_j}{\delta}$ **then**
7:         **break**
8: **return** $\Pi_j$

---

## C.3 Learning Full-Rank Policies

We consider running the MINEIG algorithm (Algorithm 6) of [64] in sim. For a fixed $h$, we instantiate the setting of Appendix C of [64] with $\boldsymbol{\psi}(\boldsymbol{\tau}) = \boldsymbol{\phi}(s_h, a_h)\boldsymbol{\phi}(s_h, a_h)^\top$, $D = 1$, and $\mathbb{A}_{\mathcal{R}}$ the policy optimization oracle of Oracle 4.2 (and so $C_{\mathcal{R}} = 0$), and set $N = 1$ for MINEIG. We note that this algorithm is computationally efficient, given a policy optimization oracle.

**Lemma C.7.** *For $\mathcal{M} \leftarrow \mathcal{M}^{\mathsf{sim}}$, Algorithm 5 will call Oracle 4.2 at most $\widetilde{\mathcal{O}}(\frac{d}{\zeta} \cdot \log \frac{1}{\delta} + \zeta^{-9} \cdot \log^{3/2} \frac{1}{\delta})$ times, and with probability at least $1 - \delta$, under Assumption 3 and if $\zeta \leq \frac{\lambda^\star_{\min}}{4d}$, will return policies $\Pi$ such that*

$$\lambda_{\min}\left(\frac{1}{|\Pi|}\sum_{\pi \in \Pi} \boldsymbol{\Lambda}^{\mathsf{s}}_{\pi, h}\right) \geq \frac{\lambda^\star_{\min}}{8d} \tag{C.2}$$

*and each $\pi \in \Pi$ plays actions randomly for $h' > h$.*

*Proof.* We first argue that, if $\zeta \leq \frac{\lambda^\star_{\min}}{4d}$, then with probability at least $1 - \delta$, (C.2) holds. Let $\mathcal{E}$ denote the success event of each call to DYNAMICOED, and note that by our choice of $\delta_j$, we have $\mathbb{P}[\mathcal{E}] \geq 1 - \delta/2$. Let $j^\star$ denote the minimal value of $j$ such that

$$\frac{\lambda^\star_{\min}}{4d}T_j \geq 12544 d \log \frac{4 + 64T_j}{\delta} \quad \text{and} \quad T_j \geq c \cdot \zeta^{-9} \cdot \log^{3/2} \frac{jT_j}{\delta}. \tag{C.3}$$

By Lemma C.4 of [64] and if $\zeta \leq \frac{\lambda^\star_{\min}}{4d}$, we then have that, on $\mathcal{E}$, $\lambda_{\min}(\boldsymbol{\Sigma}_{j^\star}) \geq \frac{\lambda^\star_{\min}}{4d}T_{j^\star}$, which implies that the termination criteria of Algorithm 5 will be met. By Lemma C.5 of [64], it follows that with probability at least $1 - \delta/2$, we have $\lambda_{\min}(\frac{1}{|\Pi_{j^\star}|}\sum_{\pi \in \Pi_{j^\star}} \boldsymbol{\Lambda}^{\mathsf{s}}_{\pi, h}) \geq \frac{\lambda^\star_{\min}}{8d}$ (since $T_{j^\star} = |\Pi_{j^\star}|$), the desired conclusion.

Assume that Algorithm 5 terminates for some $j < j^\star$. This implies that $\frac{\lambda^\star_{\min}}{4d}T_j < 12544 d \log \frac{4 + 64T_j}{\delta}$. However, in this case, we then have that

$$\lambda_{\min}(\boldsymbol{\Sigma}_j) \geq 12544 d \log \frac{4 + 64T_j}{\delta} \geq \frac{\lambda^\star_{\min}}{4d}T_j.$$

From Lemma C.5 of [64], it then follows that with probability at least $1 - \delta/2$, we have $\lambda_{\min}(\frac{1}{|\Pi_j|}\sum_{\pi \in \Pi_j} \boldsymbol{\Lambda}^{\mathsf{s}}_{\pi, h}) \geq \frac{\lambda^\star_{\min}}{8d}$.

It follows that, assuming $T_j$ is large enough that (C.3) is met, and we are in the case when $\zeta \leq \frac{\lambda^\star_{\min}}{4d}$ holds, then Algorithm 5 will terminate and return a set of policies satisfying (C.2), with probability at least $1 - \delta$. Note that $T_j = \mathcal{O}(2^j)$. Given that Algorithm 5 does not terminate until $j = \mathcal{O}(\log_2(\frac{d}{\zeta} \cdot \log \frac{1}{\delta} + \zeta^{-9} \cdot \log^{3/2} \frac{1}{\delta}) \geq \mathcal{O}(\log_2(\frac{d^2}{\lambda^\star_{\min}} \cdot \log \frac{1}{\delta} + \zeta^{-9} \cdot \log^{3/2} \frac{1}{\delta}))$, we will have that $T_j$ will be large enough that (C.3) is met, if $\zeta \leq \frac{\lambda^\star_{\min}}{4d}$. The proof then follows since DYNAMICOED calls Oracle 4.2 at most $T_j$ times at round $j$, and the total sum of $T_j$ is bounded as $\widetilde{\mathcal{O}}(\frac{d}{\zeta} \cdot \log \frac{1}{\delta} + \zeta^{-9} \cdot \log^{3/2} \frac{1}{\delta})$ by the maximum of $j$, and since the actions chosen by $\pi \in \Pi$ for $h' > h$ are irrelevant for the operation of DYNAMICOED, so they can be set to random. $\qquad\square$

# D Lower Bound Proofs

## D.1 Proof of Propositions 1, 3 and 4

**Construction.** Consider the following variation of the combination lock. We let the action space $\mathcal{A} = \{1, 2\}$, and assume there are two states, $\mathcal{S} = \{s_1, s_2\}$, and horizon $H$. We start in state $s_1$. The sim dynamics are given as:

$$\forall h < H - 1 : \quad P_h^{\mathsf{sim}}(s_1 \mid s_1, a_1) = 1, \quad P_h^{\mathsf{sim}}(s_2 \mid, s_1, a_2) = 1$$
$$P_{H-1}^{\mathsf{sim}}(s_1 \mid s_1, a_1) = P_{H-1}^{\mathsf{sim}}(s_2 \mid s_1, a_1) = P_{H-1}^{\mathsf{sim}}(s_1 \mid s_1, a_2) = P_{H-1}^{\mathsf{sim}}(s_2 \mid s_1, a_2) = 1/2$$
$$\forall h \in [H] : \quad P_h^{\mathsf{sim}}(s_2 \mid s_2, a) = 1, a \in \{a_1, a_2\}.$$

We define two real instances, $\mathcal{M}_1 := \mathcal{M}^{\mathsf{real},1}$ and $\mathcal{M}_2 := \mathcal{M}^{\mathsf{real},2}$, where for both we have:

$$\forall h < H - 1 : \quad P_h^{\mathsf{real}}(s_1 \mid s_1, a_1) = 1, \quad P_h^{\mathsf{real}}(s_2 \mid, s_1, a_2) = 1$$
$$\forall h \in [H] : \quad P_h^{\mathsf{real}}(s_2 \mid s_2, a) = 1, a \in \{a_1, a_2\}$$

for $\mathcal{M}_1$:

$$P_{H-1}^{\mathsf{real}}(s_1 \mid s_1, a_1) = 1/2 + \epsilon_{\mathsf{sim}}, P_{H-1}^{\mathsf{real}}(s_2 \mid s_1, a_1) = 1/2 - \epsilon_{\mathsf{sim}},$$
$$P_{H-1}^{\mathsf{real}}(s_1 \mid s_1, a_2) = 1/2 - \epsilon_{\mathsf{sim}}, P_{H-1}^{\mathsf{real}}(s_2 \mid s_1, a_2) = 1/2 + \epsilon_{\mathsf{sim}},$$

and for $\mathcal{M}_2$:

$$P_{H-1}^{\mathsf{real}}(s_1 \mid s_1, a_1) = 1/2 - \epsilon_{\mathsf{sim}}, P_{H-1}^{\mathsf{real}}(s_2 \mid s_1, a_1) = 1/2 + \epsilon_{\mathsf{sim}},$$
$$P_{H-1}^{\mathsf{real}}(s_1 \mid s_1, a_2) = 1/2 + \epsilon_{\mathsf{sim}}, P_{H-1}^{\mathsf{real}}(s_2 \mid s_1, a_2) = 1/2 - \epsilon_{\mathsf{sim}}.$$

Note then that $\mathcal{M}_1, \mathcal{M}_2$, and sim only differ at step $H - 1$ in state $s_1$. Furthermore, it is easy to see that both $\mathcal{M}_1$ and $\mathcal{M}_2$ satisfy Assumption 1 with misspecification $\epsilon_{\mathsf{sim}}$. It is easy to see that Assumption 2 holds as well with $d = 4$ since this is a tabular MDP, and furthermore Assumption 3 also holds with $\lambda_{\min}^\star = 1/4$. We define the reward function as (note that this is deterministic, and the same for all instances):

$$\forall h \in [H] : \quad r_h(s_1, a_2) = 1/2 + \epsilon_{\mathsf{sim}}(1/2 - h/4H)$$
$$r_H(s_1, a) = 1, a \in \{a_1, a_2\},$$

and all other rewards are taken to be 0.

In sim, we see that the optimal policy always plays $a_2$. In both $\mathcal{M}_1$ and $\mathcal{M}_2$, the optimal policy plays $a_1$ for all $h < H - 1$, for $\mathcal{M}_1$ plays $a_1$ at $H - 1$, and for $\mathcal{M}_2$ plays $a_2$ at $H - 1$. Note that for both $\mathcal{M}_1$ and $\mathcal{M}_2$, we have $V_0^\star = 1/2 + \epsilon_{\mathsf{sim}}$.

The most natural choice of $\mathcal{F}$ would be the set of all tabular $Q$-value functions, however, this set is infinite, and would require a covering argument to incorporate. For simplicity, consider $\mathcal{F}_H$ the set of functions mapping to $\{0, 1\}$, and $\mathcal{F}_h$ the set of functions mapping to a finite set containing $\{0, 1/2 - \epsilon_{\mathsf{sim}}, 1/2 + \epsilon_{\mathsf{sim}}\} \cup \{1/2 + \epsilon_{\mathsf{sim}}(1/2 - h'/4H)\}_{h'=0}^H$. Note that such a set satisfies Assumption 4 and we can construct it such that $\log |\mathcal{F}| \le \mathcal{O}(H)$.

**Lower Bound for Direct Policy Transfer (Proposition 3).** We consider direct sim2real transfer with randomized exploration. In particular, as noted, the optimal policy in sim always plays $a_2$, so we consider the $\zeta$-greedy policy that at every state plays $a_2$ with probability $1 - \zeta$, and plays $\mathsf{unif}(\{a_1, a_2\})$ with probability $\zeta$. Denote this policy as $\widetilde{\pi}$. We then wish to lower bound:

$$\inf_{\widehat{\pi}} \sup_{i \in \{1,2\}} \mathbb{E}^{\mathcal{M}_i, \widetilde{\pi}}[V_0^{\mathcal{M}_i, \star} - V_0^{\mathcal{M}_i, \widehat{\pi}}]$$

after running our procedure for $T$ episodes. Note that on $\mathcal{M}_1$, regardless of the actions $\widehat{\pi}$ chooses in other states, we have

$$V_0^{\mathcal{M}_1, \star} - V_0^{\mathcal{M}_1, \widehat{\pi}} \ge \frac{\epsilon_{\mathsf{sim}}}{2}(1 - \widehat{\pi}_{H-1}(a_1 \mid s_1)),$$

since the only way $\widehat{\pi}$ can achieve a reward of $1/2 + \epsilon_{\mathrm{sim}}$ is by playing $a_1$ in $s_1$ at step $H - 1$, and all other sequences of actions obtain a reward of at most $1/2 + \epsilon_{\mathrm{sim}}/2$. Similarly for $\mathcal{M}_2$ we have

$$V_0^{\mathcal{M}_2,\star} - V_0^{\mathcal{M}_2,\widehat{\pi}} \geq \frac{\epsilon_{\mathrm{sim}}}{2}(1 - \widehat{\pi}_{H-1}(a_2 \mid s_1)).$$

Using this, and replacing the max over $i \in \{1, 2\}$ with the average of them, we obtain

$$\inf_{\widehat{\pi}} \sup_{i \in \{1,2\}} \mathbb{E}^{\mathcal{M}_i,\widetilde{\pi}}[V_0^{\mathcal{M}_i,\star} - V_0^{\mathcal{M}_i,\widehat{\pi}}] \geq \inf_{\widehat{\pi}} \frac{1}{2}\mathbb{E}^{\mathcal{M}_1,\widetilde{\pi}}[\frac{\epsilon_{\mathrm{sim}}}{2}(1 - \widehat{\pi}_{H-1}(a_1 \mid s_1))] + \frac{1}{2}\mathbb{E}^{\mathcal{M}_2,\widetilde{\pi}}[\frac{\epsilon_{\mathrm{sim}}}{2}(1 - \widehat{\pi}_{H-1}(a_2 \mid s_1))]$$

$$= \frac{\epsilon_{\mathrm{sim}}}{2}\left[1 - \frac{1}{2} \cdot \sup_{\widehat{\pi}}\left(\mathbb{E}^{\mathcal{M}_1,\widetilde{\pi}}[\widehat{\pi}_{H-1}(a_1 \mid s_1)] + \mathbb{E}^{\mathcal{M}_2,\widetilde{\pi}}[\widehat{\pi}_{H-1}(a_2 \mid s_1)]\right)\right].$$

Since $\widehat{\pi}_{H-1}(a_1 \mid s_1) = 1 - \widehat{\pi}_{H-1}(a_2 \mid s_1)$, we have

$$\mathbb{E}^{\mathcal{M}_1,\widetilde{\pi}}[\widehat{\pi}_{H-1}(a_1 \mid s_1)] + \mathbb{E}^{\mathcal{M}_2,\widetilde{\pi}}[\widehat{\pi}_{H-1}(a_2 \mid s_1)] = 1 + \mathbb{E}^{\mathcal{M}_1,\widetilde{\pi}}[\widehat{\pi}_{H-1}(a_1 \mid s_1)] - \mathbb{E}^{\mathcal{M}_2,\widetilde{\pi}}[\widehat{\pi}_{H-1}(a_1 \mid s_1)]$$

$$\leq 1 + \mathrm{TV}(\mathbb{P}^{\mathcal{M}_1,\widetilde{\pi}}, \mathbb{P}^{\mathcal{M}_2,\widetilde{\pi}})$$

$$\leq 1 + \sqrt{\frac{1}{2}\mathrm{KL}(\mathbb{P}^{\mathcal{M}_1,\widetilde{\pi}} \parallel \mathbb{P}^{\mathcal{M}_2,\widetilde{\pi}})}$$

where TV denotes the total-variation distance, KL the KL-divergence, and the last inequality follows from Pinsker's inequality. We therefore have

$$\inf_{\widehat{\pi}} \sup_{i \in \{1,2\}} \mathbb{E}^{\mathcal{M}_i,\widetilde{\pi}}[V_0^{\mathcal{M}_i,\star} - V_0^{\mathcal{M}_i,\widehat{\pi}}] \geq \frac{\epsilon_{\mathrm{sim}}}{4}\left(1 - \sqrt{\frac{1}{2}\mathrm{KL}(\mathbb{P}^{\mathcal{M}_1,\widetilde{\pi}} \parallel \mathbb{P}^{\mathcal{M}_2,\widetilde{\pi}})}\right).$$

Now note that, since $\mathcal{M}_1$ and $\mathcal{M}_2$ only differ at state $s_1$ and step $H - 1$, we have

$$\mathrm{KL}(\mathbb{P}^{\mathcal{M}_1,\widetilde{\pi}} \parallel \mathbb{P}^{\mathcal{M}_2,\widetilde{\pi}}) = \mathbb{E}^{\mathcal{M}_1,\widetilde{\pi}}[T_{H-1}(s_1, a_1)]\mathrm{KL}(P_{H-1}^{\mathcal{M}_1}(\cdot \mid s_1, a_1) \parallel P_{H-1}^{\mathcal{M}_2}(\cdot \mid s_1, a_1))$$

$$+ \mathbb{E}^{\mathcal{M}_1,\widetilde{\pi}}[T_{H-1}(s_1, a_2)]\mathrm{KL}(P_{H-1}^{\mathcal{M}_1}(\cdot \mid s_1, a_2) \parallel P_{H-1}^{\mathcal{M}_2}(\cdot \mid s_1, a_2)),$$

where $T_{H-1}(s_1, a_i)$ denotes the total number of visits to $(s_1, a_i)$ at step $H - 1$ after $T$ episodes (see e.g. [59]). We have

$$\mathrm{KL}(P_{H-1}^{\mathcal{M}_1}(\cdot \mid s_1, a_1) \parallel P_{H-1}^{\mathcal{M}_2}(\cdot \mid s_1, a_1)) = \mathrm{KL}(P_{H-1}^{\mathcal{M}_1}(\cdot \mid s_1, a_2) \parallel P_{H-1}^{\mathcal{M}_2}(\cdot \mid s_1, a_2))$$

$$= \frac{1}{4}\log\frac{1/4}{3/4} + \frac{3}{4}\log\frac{3/4}{1/4} \leq \frac{3}{5}$$

where the last inequality holds as long as $\epsilon_{\mathrm{sim}} \leq 1/6$. Note that the only way for a policy to reach $s_1$ at step $H - 1$ is to play action $a_1$ $H - 1$ consecutive times. Since $\widetilde{\pi}$ only plays $a_1$ at any given step with probability $\zeta/2$, it follows that the probability that $\widetilde{\pi}$ reaches $s_1$ at step $H - 1$ on any given episode is only $(\zeta/2)^{H-1}$. Thus,

$$\mathrm{KL}(\mathbb{P}^{\mathcal{M}_1,\widetilde{\pi}} \parallel \mathbb{P}^{\mathcal{M}_2,\widetilde{\pi}}) \leq \frac{3}{5}\left(\mathbb{E}^{\mathcal{M}_1,\widetilde{\pi}}[T_{H-1}(s_1, a_1)] + \mathbb{E}^{\mathcal{M}_1,\widetilde{\pi}}[T_{H-1}(s_1, a_2)]\right)$$

$$= \frac{3}{5}\mathbb{E}^{\mathcal{M}_1,\widetilde{\pi}}[T_{H-1}(s_1)]$$

$$= \frac{3}{5}\left(\frac{\zeta}{2}\right)^{H-1} \cdot T.$$

We thus have:

$$\inf_{\widehat{\pi}} \sup_{i \in \{1,2\}} \mathbb{E}^{\mathcal{M}_i,\widetilde{\pi}}[V_0^{\mathcal{M}_i,\star} - V_0^{\mathcal{M}_i,\widehat{\pi}}] \geq \frac{\epsilon_{\mathrm{sim}}}{4}\left(1 - \sqrt{\frac{3}{10}\left(\frac{\zeta}{2}\right)^{H-1} \cdot T}\right)$$

and we therefore have $\inf_{\widehat{\pi}} \sup_{i \in \{1,2\}} \mathbb{E}^{\mathcal{M}_i,\widetilde{\pi}}[V_0^{\mathcal{M}_i,\star} - V_0^{\mathcal{M}_i,\widehat{\pi}}] \geq \epsilon_{\mathrm{sim}}/8$ unless

$$T \geq \frac{5}{6} \cdot \left(\frac{2}{\zeta}\right)^{H-1}.$$

**Lower Bound for $\zeta$-Greedy Without sim (Proposition 1).** In order to quantify the performance of a $\zeta$-greedy algorithm, we must specify how it chooses $\widehat{f}$ when it has not yet observed any samples from a given $(s, a, h)$. Following the lead of Theorem 2 of [11], to avoid an overly optimistic or pessimistic initialization, we assume that the replay buffer is initialized with a single sample from each $(s, a, h)$. Note that the conclusion would hold with other initializations, however, e.g. initializing $\widehat{f}_h(s, a) = 0$ or randomly if we have no observations from $(s, a, h)$.

Assume that the observation from $(s_1, a_1, H - 1)$ transitions to $s_2$, which occurs with probability at least $1/4$. In this case, we then have that, for each $h$, $\widehat{f}_h^0(s_1, a_2) \geq \widehat{f}_h^0(s_1, a_1)$. Thus, following the $\zeta$-greedy policy, we have that $\pi_h^0(a_1 \mid s_1) \leq 1/2$. Denote this event on $\mathcal{E}_0$. Furthermore, the only way we will have $\widehat{f}_h^0(s_1, a_2) < \widehat{f}_h^0(s_1, a_1)$ is if we visit $(s_1, a_1, H - 1)$ again and observe a transition to $s_1$. For this to occur, however, we must play action $a_1$ $H - 1$ times consecutively which, in this case, will occur with probability at most $\max\{1/2, \zeta/2\}^{H-1} \leq 1/2^{H-1}$.

Following the argument in the direct policy transfer case, we have

$$\inf_{\widehat{\pi}} \sup_{i \in \{1,2\}} \mathbb{E}^{\mathcal{M}_i, \widetilde{\pi}}[V_0^{\mathcal{M}_i, \star} - V_0^{\mathcal{M}_i, \widehat{\pi}}] \geq \inf_{\widehat{\pi}} \sup_{i \in \{1,2\}} \frac{1}{4} \mathbb{E}^{\mathcal{M}_i, \widetilde{\pi}}[V_0^{\mathcal{M}_i, \star} - V_0^{\mathcal{M}_i, \widehat{\pi}} \mid \mathcal{E}_0]$$

$$\geq \frac{\epsilon_{\mathsf{sim}}}{16} \left( 1 - \sqrt{\frac{3}{10} \mathbb{E}^{\mathcal{M}_1}[T_{H-1}(s_1) \mid \mathcal{E}_0]} \right)$$

where $\mathbb{E}^{\mathcal{M}_1}[T_{H-1}(s_1) \mid \mathcal{E}_0]$ is the expected number of visitations to $(s_1, H - 1)$ after $T$ episodes of running the $\zeta$-greedy policy. We can rewrite

$$\mathbb{E}^{\mathcal{M}_1}[T_{H-1}(s_1) \mid \mathcal{E}_0] = \sum_{t=1}^{T} \mathbb{E}^{\mathcal{M}_1}[\mathbb{I}\{s_{H-1} = s_1\} \mid \mathcal{E}_0].$$

Let $\mathcal{E}$ be the event that we have reached $(s_1, H - 1)$ in the first $T$ rounds. Then,

$$\mathbb{E}^{\mathcal{M}_1}[\mathbb{I}\{s_{H-1} = s_1\} \mid \mathcal{E}_0] = \mathbb{E}^{\mathcal{M}_1}[\mathbb{I}\{s_{H-1} = s_1\} \mid \mathcal{E}, \mathcal{E}_0]\mathbb{P}^{\mathcal{M}_1}[\mathcal{E} \mid \mathcal{E}_0] + \mathbb{E}^{\mathcal{M}_1}[\mathbb{I}\{s_{H-1} = s_1\} \mid \mathcal{E}^c, \mathcal{E}_0]\mathbb{P}^{\mathcal{M}_1}[\mathcal{E}^c \mid \mathcal{E}_0]$$

$$\leq \mathbb{P}^{\mathcal{M}_1}[\mathcal{E} \mid \mathcal{E}_0] + \mathbb{E}^{\mathcal{M}_1}[\mathbb{I}\{s_{H-1} = s_1\} \mid \mathcal{E}^c, \mathcal{E}_0].$$

By what we have just argued, we have $\mathbb{P}^{\mathcal{M}_1}[\mathcal{E} \mid \mathcal{E}_0] \leq T \cdot \frac{1}{2^{H-1}}$, and $\mathbb{E}^{\mathcal{M}_1}[\mathbb{I}\{s_{H-1} = s_1\} \mid \mathcal{E}^c, \mathcal{E}_0] \leq \frac{1}{2^{H-1}}$. Thus, $\mathbb{E}^{\mathcal{M}_1}[T_{H-1}(s_1) \mid \mathcal{E}_0] \leq \frac{2T^2}{2^{H-1}}$. It follows that,

$$\inf_{\widehat{\pi}} \sup_{i \in \{1,2\}} \mathbb{E}^{\mathcal{M}_i, \widetilde{\pi}}[V_0^{\mathcal{M}_i, \star} - V_0^{\mathcal{M}_i, \widehat{\pi}}] \geq \frac{\epsilon_{\mathsf{sim}}}{16} \left( 1 - \sqrt{\frac{3}{10} \frac{2T^2}{2^{H-1}}} \right)$$

and we therefore have $\inf_{\widehat{\pi}} \sup_{i \in \{1,2\}} \mathbb{E}^{\mathcal{M}_i, \widetilde{\pi}}[V_0^{\mathcal{M}_i, \star} - V_0^{\mathcal{M}_i, \widehat{\pi}}] \geq \epsilon_{\mathsf{sim}}/32$ unless

$$T \geq \sqrt{\frac{5}{8} \cdot 2^{H-1}}.$$

**Upper Bound for Exploration Policy Transfer (Proposition 4).** To obtain an upper bound for Algorithm 1, we can apply Theorem 1, so long as

$$\epsilon_{\mathsf{sim}} \leq \frac{\lambda_{\min}^{\star}}{64 d H A^3}.$$

Note that in our setting we have $d = 4$, $A = 2$, $\lambda_{\min}^{\star} = 1/4$, so this condition reduces to $\epsilon_{\mathsf{sim}} \leq \frac{1}{8192 H}$. Taking $\mathcal{F}$ to simply be the set of $Q$-functions defined above (so $V_{\max} = H$), Theorem 1 then gives that with probability at least $1 - \delta$, Algorithm 1 learns an $\epsilon$-optimal policy as long as $T \geq c \cdot \frac{H^{17}}{\epsilon^8} \cdot \log \frac{H}{\delta}$.

### D.2  Proof of Proposition 5

We define three MDPs: $\mathcal{M}^{\mathsf{sim}}$, and two possible real MDPs, $\mathcal{M}_1 := \mathcal{M}^{\mathsf{real},1}$ and $\mathcal{M}_2 := \mathcal{M}^{\mathsf{real},2}$. In all cases we have states $\mathcal{S} = \{s_1, s_2\}$, actions $\mathcal{A} = \{a_1, a_2, a_3, a_4\}$, and $H = 2$, and set the starting state to $s_1$. We define

$$P_1^{\mathsf{sim}}(s_1 \mid s_1, a_1) = 1, \quad P_1^{\mathsf{sim}}(s_1 \mid s_1, a) = P_1^{\mathsf{sim}}(s_2 \mid s_1, a) = 1/2, a \in \{a_2, a_3, a_4\}.$$

For both $\mathcal{M}_1$ and $\mathcal{M}_2$, we have:

$$P_1^{\mathsf{real}}(s_1 \mid s_1, a_1) = 1, P_1^{\mathsf{real}}(s_1 \mid s_1, a_4) = P_1^{\mathsf{real}}(s_2 \mid s_1, a_4) = 1/2$$

for $\mathcal{M}_1$, we have

$$P_1^{\mathsf{real}}(s_2 \mid s_1, a_2) = 1 + \epsilon_{\mathsf{sim}}, P_1^{\mathsf{real}}(s_1 \mid s_1, a_2) = 1 - \epsilon_{\mathsf{sim}}, P_1^{\mathsf{real}}(s_1 \mid s_1, a_3) = P_1^{\mathsf{real}}(s_2 \mid s_1, a_3) = 1/2$$

and for $\mathcal{M}_2$,

$$P_1^{\mathsf{real}}(s_2 \mid s_1, a_3) = 1 + \epsilon_{\mathsf{sim}}, P_1^{\mathsf{real}}(s_1 \mid s_1, a_3) = 1 - \epsilon_{\mathsf{sim}}, P_1^{\mathsf{real}}(s_1 \mid s_1, a_2) = P_1^{\mathsf{real}}(s_2 \mid s_1, a_2) = 1/2.$$

We take the reward to be 0 everywhere, except $r_2(s_2, a) = 1$ for all $a$.

Note that each of these can be represented as a linear MDP in $d = 2$ dimensions, so Assumption 2 holds. In particular, for $\mathcal{M}^{\mathsf{sim}}$ we can take:

$$\boldsymbol{\phi}^{\mathsf{s}}(s, a_1) = e_1, \boldsymbol{\phi}^{\mathsf{s}}(s, a) = e_2, a \in \{a_2, a_3, a_4\}, s \in \mathcal{S},$$
$$\boldsymbol{\mu}_1^{\mathsf{s}}(s_1) = [1, 1/2], \boldsymbol{\mu}_1^{\mathsf{s}}(s_2) = [0, 1/2].$$

For $\mathcal{M}_1$ we can instead take:

$$\boldsymbol{\phi}^{\mathsf{r}}(s, a_1) = e_1, \boldsymbol{\phi}^{\mathsf{r}}(s, a) = [1/2, 1/2], a \in \{a_3, a_4\}, s \in \mathcal{S},$$
$$\boldsymbol{\phi}^{\mathsf{r}}(s, a_2) = [1/2 - \epsilon_{\mathsf{sim}}, 1/2 + \epsilon_{\mathsf{sim}}], s \in \mathcal{S},$$
$$\boldsymbol{\mu}_1^{\mathsf{r}}(s_1) = [1, 0], \boldsymbol{\mu}_1^{\mathsf{r}}(s_2) = [0, 1].$$

$\mathcal{M}_2$ follows similarly with the role of $a_2$ and $a_3$ flipped.

It is easy to see that Assumption 1 is met on this instance for both choices of $\mathcal{M}^{\mathsf{real}}$. On $\mathcal{M}^{\mathsf{sim}}$, the policy $\pi_{\mathrm{exp}}$ which in every states plays action $a_1$ with probability 1/2 and action $a_4$ with probability 1/2 satisfies $\lambda_{\min}(\mathbb{E}^{\mathsf{sim}, \pi_{\mathrm{exp}}}[\boldsymbol{\phi}^{\mathsf{s}}(s_h, a_h)\boldsymbol{\phi}^{\mathsf{s}}(s_h, a_h)^\top]) \geq 1/2$ (which shows that Assumption 3 holds).

Note, however, that $\pi_{\mathrm{exp}}$ does not play action $a_2$ or $a_3$. As $\mathcal{M}_1$ and $\mathcal{M}_2$ differ only on $a_2$ and $a_3$, playing $\pi_{\mathrm{exp}}$ will not allow for $\mathcal{M}_1$ and $\mathcal{M}_2$ to be distinguished. As $a_2$ is the optimal action on $\mathcal{M}_1$ and $a_3$ the optimal action on $\mathcal{M}_2$, it follows that playing $\pi_{\mathrm{exp}}$ will not allow for the identification of the optimal policy on $\mathcal{M}_1$ and $\mathcal{M}_2$. This can be formalized identically to Appendix D.1, yielding the stated result.

# E    Experimental Details

## E.1    Didactic Tabular Example

Consider the following variation of the combination lock. We let the action space $\mathcal{A} = \{1, 2\}$, and assume there are two states, $\mathcal{S} = \{s_1, s_2\}$, and horizon $H$. We start in state $s_1$. The sim dynamics are given as:

$$\forall h < H - 1 : \quad P_h^{\mathsf{sim}}(s_1 \mid s_1, a_1) = 1, \quad P_h^{\mathsf{sim}}(s_2 \mid, s_1, a_2) = 1$$
$$P_{H-1}^{\mathsf{sim}}(s_1 \mid s_1, a_1) = 1/4, P_{H-1}^{\mathsf{sim}}(s_2 \mid s_1, a_1) = 3/4, P_{H-1}^{\mathsf{sim}}(s_2 \mid s_1, a_2) = 1$$
$$\forall h \in [H] : \quad P_h^{\mathsf{sim}}(s_2 \mid s_2, a) = 1, a \in \{a_1, a_2\},$$

and the real dynamics are given as:

$$\forall h < H - 1 : \quad P_h^{\mathsf{real}}(s_1 \mid s_1, a_1) = 1, \quad P_h^{\mathsf{real}}(s_2 \mid, s_1, a_2) = 1$$
$$P_{H-1}^{\mathsf{real}}(s_1 \mid s_1, a_1) = 3/4, P_{H-1}^{\mathsf{real}}(s_2 \mid s_1, a_1) = 1/4, P_{H-1}^{\mathsf{real}}(s_2 \mid s_1, a_2) = 1$$
$$\forall h \in [H] : \quad P_h^{\mathsf{real}}(s_2 \mid s_2, a) = 1, a \in \{a_1, a_2\}.$$

Note that these only differ on $(s_1, a_1)$ at $h = H - 1$, and we have $\epsilon_{\mathsf{sim}} = 1/2$. We define the reward function as (note that this is deterministic, and the same for both sim and real):

$$\forall h \in [H] : \quad r_h(s_1, a_2) = 1/8 - h/8H$$
$$r_H(s_1, a) = 1/5, a \in \{a_1, a_2\},$$

and all other rewards are taken to be 0.

The intuition for this example is as follows. In both sim and real, the only way the agent can get reward is to either end up in state $s_1$ at step $H$, or to take action $a_2$ in state $s_1$ at any point. In sim, the probability of ending up in state $s_1$ at step $H$, even if the optimal sequence of actions to do this is taken, is only $1/4$, due to the final transition, and thus the average reward obtained by the policy which aims to end up in $s_1$ is only 1/4. In contrast, if we take action $a_2$ in $s_1$, we will always collect reward of at least 3/8 (and the earlier we take action $a_2$ the more reward we collect, up to 1/2). Thus, in sim the optimal thing to do in $s_1$ is always to play $a_2$. However, if we play $a_2$ even once, we will transition out of $s_1$ and never return, so there is no chance we will reach $s_1$ at step $H$.

In real, the transitions at the final step are flipped, so that now the probability of finishing in $s_1$, if we take the optimal sequences of actions to do this, is 3/4, and the expected reward for this is then also 3/4. Since the reward for taking $a_2$ in $s_1$ does not change, and is bounded as 1/2, then in real the optimal policy is to seek to end up in $s_1$ at the final step.

The challenge with ending up in $s_1$ at the end is that it requires playing action $a_1$ at every step. In this sense it is then a classic combination lock instance, and randomized exploration will fail, requiring $\Omega(2^H)$ episodes to reach the final state (since the probability of randomly taking $a_1$ at every state decreases exponentially with the horizon). Similarly, if we transfer the optimal policy from sim to real, it will never take action $a_1$, so will never reach $s_1$ at the end, and if we transfer the optimal policy from sim with some random exploration, it will fail for the same reason random exploration from scratch fails.

However, note that we can transfer a policy from sim that is able to reach $s_1$ at the second-to-last step with probability 1, i.e. the policy that takes action $a_1$ at every step. Thus, if in sim we aim to learn exploration policies that can traverse the MDP, and we transfer these exploration policies, they will transfer, and will allow us to easily reach $s_1$ at the final step, and quickly determine that it is indeed the optimal thing in real.

We provide additional experimental results on this instance in Appendix E.1.

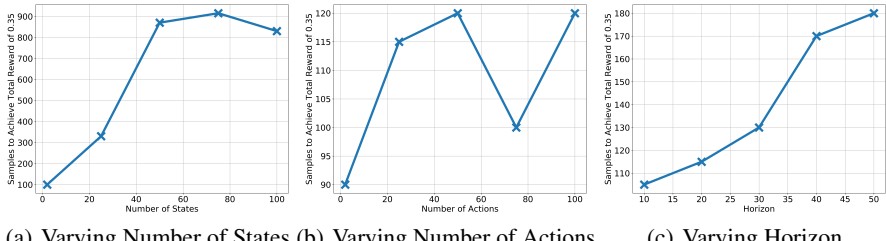

(a) Varying Number of States (b) Varying Number of Actions (c) Varying Horizon

Figure 5: Performance of Exploration Policy Transfer on instance from Section 5.2, varying number of states, actions, and horizon. We plot the number of samples required to achieve a reward of 0.35, which is approximately solving the task. All results are averaged across 20 trials. When increasing the number of states, we add additional 0-reward states (i.e. states given in yellow in Figure 2), and when adding additional actions we add additional low-reward actions (i.e. actions that have the same behavior as action $a_2$ in Figure 2). We observe that increasing the number of states and horizon increases the number of samples needed, while increasing the number of actions does not substantially. We emphasize, however, that this is for a *particular* example, and this scaling may not be the same for all examples—Theorem 1, however, gives an upper bound on *all* examples.

## E.2 Practical Algorithm Details

The core of our work is to decouple the optimal policy training from exploration strategies in reinforcement learning fine-tuning. Specifically, we propose a framework that uses a set of diverse exploration policies to collect samples from the environment. These exploration policies are fixed while we run off policy RL updates on the collected samples to extract an optimal policy. Our theoretical derivation suggests that this decoupling can improve sample efficiency and overall learning performance.

**Algorithm 6** sim2real transfer using OS for exploration and SAC for optimization

1: **Input:** Simulator $\mathcal{M}^{\text{sim}}$, real environment $\mathcal{M}^{\text{real}}$, simulator training budget $N$, exploration reward balancing $\alpha$, reward threshold $\epsilon$, exploration set size $n$.
2: **Pre-train Exploration Policies in $\mathcal{M}^{\text{sim}}$:**
3: Initialize $\Pi_{exp} = \{\pi_\theta(\cdot|z)|z \in \{1 \ldots n\}\}$
4: Initialize discriminator $D_\phi$
5: **for** $i = 1$ to $N$ **do**                                      ▷ Learn diverse exploration policies
6:     Sample latent $z \sim \text{unif}(1, n)$ and initial state $s_0$.
7:     **for** $t = 1$ to max_steps_per_episode **do**
8:         Sample action $a_t \sim \pi_\theta(a_t|s_t, z)$.
9:         Step environment: $s_{t+1} \sim p(s_{t+1}|s_t, a_t)$.
10:         Compute discriminator score $d_t = D(s_{t+1}, z)$
11:         Compute exploration reward $r_e(s_{t+1}, z) = \log \frac{\exp(d_t)}{\sum_{z'} \exp(d(s_{t+1}, z'))}$.
12:         **if** $R_\pi \geq \epsilon$ **then**
13:             Compute reward $r_t = r(s_t, a_t) + \alpha \cdot r_e(s_{t+1}, z)$.
14:         **else**
15:             Compute reward $r_t = r(s_t, a_t)$
16:         Let $\mathcal{D} \leftarrow \mathcal{D} \cup \{(s_t, a_t, r_t, s_{t+1}, z)\}$.
17:         Update $\pi_\theta$ to maximize $J_\pi$ with SAC.
18:         Update $\phi$ to maximize $J_u$, $\phi \leftarrow \phi + \eta \nabla_\phi \mathbb{E}_{s,z \sim \mathcal{D}} [\log D_\phi(s, z)]$
19:     Compute $R_\pi = \sum_t r_t$
20: **Explore in $\mathcal{M}^{\text{real}}$ and Estimate Optimal Policy :**
21: Initialize SAC agent (either from scratch or to weights of optimal sim policy).
22: **while** not converged **do**
23:     Sample $z \sim \text{unif}(1, n)$, play $\pi_\theta(\cdot \mid z)$ in $\mathcal{M}^{\text{real}}$, add data to replay buffer of SAC.
24:     Roll out SAC policy for one step, perform standard SAC update.

Our framework is complementary to (a) RL works on diversity or exploration that generate diverse policies and (b) off policy RL algorithms that optimize for policies. One can plug in (a) to extract a set of exploration policy from a simulator and use them for data collect in the real world but use (b) to optimize for the final policy. The design choice to use simulator to extract a set of exploration policies where each policy is not necessarily optimizing for the task at hand marks our distinction from previous works in (a) and (b).

We provide a practical instantiation of our framework using an approach inspired by One Solution is Not All You Need (OS) [30] to extract exploration policies and Soft Actor Critic (SAC) [30] to optimize for the optimal policy. We details the instantiation in Algorithm 6. OS trains a set of policy to optimize not only the task reward but also a discriminator reward where the discriminator encourages each policy to achieve different state. Unlike OS which carefully balances the task and exploration rewards to ensure all policies have a chance at solving the desired task, we emphasize only on having diverse policies. With a known sim2real gap, we posit that some sub-optimal policies that are not solving the task in the simulator is actually helpful for exploration in the real world, which allows us to simplify the balance between task and exploration. We uses standard off-shelf SAC update to optimize for the policy.

### E.3   TychoEnv sim2sim **Experiment Details**

For the TychoEnv experiment we run a variant of Algorithm 6. We set $n = 20$, and set the reward to $r_t^i = (1 - \alpha_i)r + \alpha_i r_e$ where we vary $\alpha_i$ from 0 to 0.5. While we use a sparse reward in $\mathcal{M}^{\text{real}}$, to speed up training in $\mathcal{M}^{\text{sim}}$ we use a dense reward that penalizes the agent for its distance to the target. We train in $\mathcal{M}^{\text{sim}}$ for 7M steps to obtain exploration policies. Rather than simply transferring the converged version of the exploration policies trained in $\mathcal{M}^{\text{sim}}$, we found it most effective to save the weights of the policies throughout training, and transfer all of these policies. As the majority of these policies do not collect any reward in $\mathcal{M}^{\text{sim}}$, we run an initial filtering stage where we identify several policies from this set that find reward (this can be seen in Figure 4 with the initial region of 0 reward). We then run SAC in $\mathcal{M}^{\text{real}}$, initialized from scratch, feeding in the data collected by these refined exploration policies into the replay buffer. We found it most effective to only inject data from the

| Hyperarameter | Value |
|---|---|
| reward balance $\alpha$ (OS) | $\{\frac{1}{38}i - \frac{1}{38} \ : \ i = 1, 2, \ldots, 20\}$ |
| learning rate | 0.0003 |
| Q update magnitude $\tau$ | 0.005 |
| discount $\gamma$ | 0.99 |
| batch size | 2048 |
| steps per episode | 45 |
| replay buffer size | $5 \times 10^6$ |
| training steps $N$ (in $\mathcal{M}^{\text{real}}$) | $7 \times 10^7$ |

Table 1: Hyperparameters used in Tycho training and finetuning

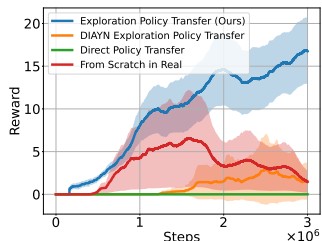

Figure 6: Additional results on Tycho, including baselines training from scratch in $\mathcal{M}^{\text{real}}$, and training exploration policies in $\mathcal{M}^{\text{sim}}$ with reward as stated above but with $\alpha_i = 1$ (which is equivalent to simply training exploration policies with DIAYN [14]). As can be seen, while training from scratch in $\mathcal{M}^{\text{real}}$ is able to learn, it learns at a much slower rate than exploration policy transfer, and achieves a much lower final value. Furthermore, training the exploration policies to maximize a mix of the task and diversity reward yields a substantial gain over simply training them to be diverse.

exploration policies in the replay buffer on episodes where they observe reward. We run vanilla SAC with UTD = 3 and target entropy of -3. We rely on the implementation of SAC from stable-baselines3 [51].

For direct policy transfer, we train a policy to convergence in $\mathcal{M}^{\text{sim}}$ that solves the task (using SAC), and then transfer this single policy, otherwise following the same procedure as above.

In $\mathcal{M}^{\text{real}}$, our reward is chosen to have a value of 50 if the end effector makes contact with the ball, and otherwise 0. If the robot successfully makes contact with the ball the episode terminates. To generate a realistic transfer environment, we change the control frequency (doubling it in $\mathcal{M}^{\text{real}}$) and the action bounds.

For both methods, we run the $\mathcal{M}^{\text{sim}}$ training procedure 4 times, and then with each of these run it in $\mathcal{M}^{\text{real}}$ twice. Error bars in our plot denote one standard error.

All experiments were run on two Nvidia V100 GPUs, and 32 Intel(R) Xeon(R) CPU E5-2620 v4 @ 2.10GHz CPUs. Additional hyperparameters in given in Table 1.

We provide results on several additional baselines for the Tycho setup in Figure 6.

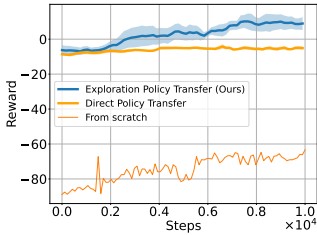

Figure 7: Results on Franka sim2real experiment, comparing to training from scratch in real.

| Hyperarameter | Value |
|---|---|
| reward balance $\alpha$ (OS) | 0.5 |
| reward threshold $\epsilon$ (OS) | -16 |
| learning rate | 0.0003 |
| Q update magnitude $\tau$ | 0.005 |
| discount $\gamma$ | 0.99 |
| batch size | 256 |
| steps per episode | 45 |
| replay buffer size | $1 \times 10^6$ |
| training steps $N$ | $2 \times 10^7$ |

Table 2: Hyperparameters used in Franka training and finetuning

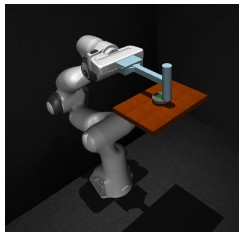

Figure 8: Franka Hammering Task Setup

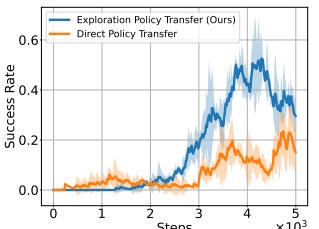

Figure 9: Results on sim2sim Transfer in Franka Simulator

### E.4 sim2sim **Transfer on Franka Emika Panda Robot Arm**

We next turn to the Franka Emika Panda robot arm [17], for which we use a realistic custom simulator built using the MuJoCo simulation engine [61]. We consider a hammering task, where the Franka arm holds a hammer, and the goal is to hammer a nail into the board (see Figure 8). Success is obtained when the nail is fully inserted. We simulate sim2real transfer by setting $\mathcal{M}^{\text{real}}$ to be a version of the simulator with nail location and stiffness significantly beyond the range seen during training in $\mathcal{M}^{\text{sim}}$.

We compare exploration policy transfer with direct sim2real policy transfer. Unlike the Tycho experiment, where we trained policies from scratch in $\mathcal{M}^{\text{real}}$ and simply used the policies trained in $\mathcal{M}^{\text{sim}}$ to explore, here we initialize the task policy in $\mathcal{M}^{\text{real}}$ to $\pi^{\text{sim},\star}$, which we then finetune on the data collected in $\mathcal{M}^{\text{real}}$ by running SAC. For direct sim2real transfer, we collect data in $\mathcal{M}^{\text{real}}$ by simply rolling out $\pi^{\text{sim},\star}$ and feeding this data to the replay buffer of SAC. For exploration policy transfer, we train an ensemble of $n = 10$ exploration policies in $\mathcal{M}^{\text{sim}}$ and run these policies in $\mathcal{M}^{\text{real}}$, again feeding this data to the replay buffer of SAC to finetune $\pi^{\text{sim},\star}$. During training in $\mathcal{M}^{\text{sim}}$, we utilize domain randomization for both methods, randomizing nail stiffness, location, radius, mass, board size, and damping.

The results of this experiment are shown in Figure 9. We see that, while direct policy transfer is able to learn, it learns at a significantly slower rate than our exploration policy transfer approach, and achieves a much smaller final success rate.

### E.5 **Franka** sim2real **Experiment Details**

We use Algorithm 6 to train a policy on the Franka robot with $n = 15$.

The reward of the pushing task is given by:

$$r(s_t, a_t) = -\|\mathbf{p}_{\text{ee}} - \mathbf{p}_{\text{obj}}\|^2 - \|\mathbf{p}_{\text{obj}} - \mathbf{p}_{\text{goal}}\|^2 + \mathbb{I}_{\mathbf{p}_{\text{obj}} - \mathbf{p}_{\text{goal}} \leq 0.025} - \mathbb{I}_{\mathbf{p}_{\text{obj}}\text{offtable}} \quad \text{(E.1)}$$

where $\mathbf{p}_{\text{goal}}$ is the desired position of the puck by the edge of the surface.

The network architecture of the actor and critic networks are identical, consisting of a 2-layer MLP, each of size 256 and ReLU activations.

We use stable-baselines3 [51] for our SAC implementation, using all of their default hyperparameters. The implemention of OS is built on top of this SAC implementation. Values of hyperparameters

are shown in Table 2. Gaussian noise with mean 0 and standard deviation 0.005 meters is added in simulation to the position of the puck. Hyperparameters are identical between exploration policy transfer and direct transfer methods.

For finetuning in real, we start off by sampling exclusively from the buffer used during simulation. Then, as finetuning proceeds, we gradually start taking more samples from the real buffer, with the proportion of samples taken from sim equal to $1 - s/3000$, where $s$ is the current number of steps. After 3000 steps, all samples are taken from the real buffer.

Experiments were run using a standard Nvidia RTX 4090 GPU. Training in simulation takes about 3 hours, while finetuning was ran for about 90 minutes.

In Figure 7, we provide results on this setup running the additional baseline of training a policy from scratch in real. As can be seen, this is significantly worse than either transfer method.

