# OpenReview forum: "Overcoming the Sim-to-Real Gap: Leveraging Simulation to Learn to Explore for Real-World RL"
_NeurIPS.cc/2024/Conference — NeurIPS 2024 poster_

### Official Review · Reviewer_3Mfq · 2024-06-19

**Soundness:** 3
**Presentation:** 3
**Contribution:** 3
**Rating:** 6
**Confidence:** 2

**Summary:**

The authors contribute a mathematical proof showing that transferring a "exploration strategy" in sim2real scenarios as opposed to transfer the learned policy result in an exponential improvement on samples needed to learn the real task.

**Strengths:**

- While not necessarily a fundamentally new idea (the transfer learning and sim2real areas have been long avoiding the naive transfer of policies and biasing exploration has always been one of the chosen strategies to perform transfer), the theoretical results presented confirm and fundament this idea that has been informally explored by the literature.

- The authors present a practical, easy to understand, exploration method and show that it overperforms greatly a simple policy transfer method in a somewhat complex sim2real robotic evaluation.

- The topic explored is not only adherent to neurips but also very timely in supporting more targeted efforts to apply RL to novel practical domains.

**Weaknesses:**

- I am not sure why a new basic "sim2real" formalization was used to in this paper when it already exists. The problem explored in this paper is completely equivalent to Multi-Fidelity MDPs (described in {1}), in fact, MF-MDPs even have something not incorporated in the description provided that is the cost of sampling in each fidelity. The MF-MDP paper was not even cited so I assume this happened because the authors have not read the paper. I suggest the paper is rewritten to describe the problem as an MF-MDP for standardization (there shouldn't be any impact to the conclusions or contribution fro the paper), but even if not rewritten at the very least this paper has to be cited and incorporated to the related works:

{1} - Silva, F. L., et al. Toward multi-fidelity reinforcement learning for symbolic optimization. ALA workshop @ AAMAS, 2023.

- Another thing that made me be in the fence regarding my score to this paper is the lack of baselines for the experimental evaluation. There has been many papers exploring a similar idea in the past, so I feel like they should have been incorporated to the experimental evaluation (Cuttler {2} comes to mind immediately, but you also have other newer options)..

{2} Cutler, Mark, Thomas J. Walsh, and Jonathan P. How. "Reinforcement learning with multi-fidelity simulators." 2014 IEEE International Conference on Robotics and Automation (ICRA). IEEE, 2014.

- A limitation both in the theoretical and practical side of the paper is that it is assumed that the transfer will happen "one shot", neglecting the fact that either the simulator could be modified to get closer to the real world {3} and the learning process in the simulator repeated, or the approach could already account that the simulator and real world will be different and adapt for that (e.g. {2} and {3}). The present paradigm presented by the authors to "learn from the simulation once and never get back to it" is at least an incomplete picture of all that could be done.

{3} - Hanna, Josiah, and Peter Stone. "Grounded action transformation for robot learning in simulation." Proceedings of the AAAI Conference on Artificial Intelligence. Vol. 31. No. 1. 2017.

**Questions:**

- Why is it assumed that the transfer HAS to be made one-shot (transferred completely in the beginning of the learning process in the "real environment")? Wouldn't it be better to have the exploration transfer strategy work as a "loop" where the exploration could be revised according to how the real world differs to the simulator?

- Why weren't MF-MDPs used to describe the problem?

- it is stated that the conditions for us being able to find a policy that solve the problem in the simulation is that ϵ ≥ 2H^2ϵ_sim. What is the meaning of ϵ here? Finding conditions for a direct policy transfer to work would also be an important contribution, specially if there is some way to test for those conditions with some samples from both the real world and simulator.

**Limitations:**

Basic research, no potential negative societal impact to explore.

---

> ### Author Rebuttal · Authors · 2024-08-06
>
> We thank the reviewer for their feedback, and will work to incorporate all suggested improvements.
>
> > I am not sure why a new basic "sim2real" formalization was used to in this paper when it already exists. The problem explored in this paper is completely equivalent to Multi-Fidelity MDPs (described in {1})...
>
> We apologize for missing this reference. We will include it in the final version of the paper, and make clear what the connections are between our setting and MF-MDPs. In particular, our setting is a special case of MF-MDPs with two environments. However, as we wish to emphasize the role of each of these environments—a “sim” and “real” environment—and since there are only two such environments, we believe it is somewhat more clear to maintain our current terminology.
>
> > Another thing that made me be in the fence regarding my score to this paper is the lack of baselines for the experimental evaluation…
>
> Please see our comment to all reviewers on what we believe are the key contributions of this work. Briefly, our primary contribution is theoretical: we provide the first result showing that simulators can provably help real-world RL in non-trivial settings. Our experimental results further validate this by showing that the algorithmic insights we derive are applicable in practice. We agree, however, that comparisons to additional baselines would be useful in fully validating the empirical effectiveness of our approach. We have aggregated all the reviewers' suggestions and added two additional baselines: using DIAYN [Eysenbach, 2018] to train exploration policies and transferring as our theory suggests, and training from scratch in real. We include these results in the rebuttal pdf.
>
> Eysenbach, Benjamin, et al. "Diversity is all you need: Learning skills without a reward function." arXiv preprint arXiv:1802.06070 (2018).
>
> > A limitation both in the theoretical and practical side of the paper is that it is assumed that the transfer will happen "one shot"...
>
> We acknowledge the potential for iterative adaptation between the simulator and the real world as suggested by the reviewer. We agree that in general, simulations can be improved (i.e., through system identification). However, we are primarily interested in settings with an irreducible gap between the simulator and real world. For example, for our Franka sim2real experiment, we have chosen physical parameters to match the real world dynamics as effectively as possible. Nevertheless, a sim2real gap persists, likely coming from imperfect friction cone modeling, imperfect contact modeling, unmodeled dynamics, latency, etc. Many real-world robotics work dealing with contacts and manipulation shared our observations [OpenAI et al, 2018; Höfer et al., 2021; Zhang et al., 2023] and confirmed that, for many problems there can be an irreducible sim2 real gap.
>
> In settings such as ours with irreducible sim2real gap, while it may be possible to use a simulator in a more iterative fashion, we see our results as proof of concept, illustrating that, even in the limited setting where you can only do 0-shot transfer, extracting exploration policies from a simulator yield a provable (exponential) gain over simply extracting the optimal policy from the simulator. Thus, while there may be other ways to utilize a simulator, this would not change the conclusion of the paper: simulators yield a provable gain in real-world RL and extracting exploration policies yield a provable gain than extracting the optimal policy with domain randomization (something not previously known). Understanding the most effective way to utilize a simulator (which may involve using a more iterative fashion) is an interesting direction for future work.
>
> Andrychowicz, OpenAI: Marcin, et al. "Learning dexterous in-hand manipulation." The International Journal of Robotics Research 39.1 (2020).
>
> Höfer, Sebastian, et al. "Sim2real in robotics and automation: Applications and challenges." IEEE transactions on automation science and engineering 18.2 (2021): 398-400.
>
> Zhang, Yunchu, et al. "Cherry-Picking with Reinforcement Learning: Robust Dynamic Grasping in Unstable Conditions." arXiv preprint arXiv:2303.05508 (2023).
>
> > It is stated that the conditions for us being able to find a policy that solve the problem in the simulation is that $\epsilon \ge 2 H^2 \epsilon_{sim}$...
>
> Here, $\epsilon$ denotes the desired optimality tolerance we wish to learn a policy up to—our goal is to find an $\epsilon$-optimal policy—while $\epsilon_{sim}$ denotes the mismatch between sim and real.
>
> Our results do provide necessary and sufficient conditions for direct policy transfer to succeed: as Proposition 1 shows, as long as $\epsilon \ge 2 H^2 \epsilon_{sim}$, direct policy transfer succeeds in finding an $\epsilon$-optimal policy, and as Proposition 2 shows, direct policy transfer cannot in general find a policy that is better than $\epsilon_{sim}/32$-optimal.
>
> It is unclear in practical settings how to measure when this condition is met—obtaining such a method is an interesting direction for future work.

---

> > ### Comment · Reviewer_3Mfq · 2024-08-10
> >
> > I am not sure why DIAYN was considered as the best baseline to add to the paper especially looking at its performance being way worse than just training from scratch.
> >
> > Still, given the main contribution of the paper is the theoretical framework I will increase a bit my already-positive grade.

---

### Official Review · Reviewer_ceeA · 2024-07-11

**Soundness:** 2
**Presentation:** 3
**Contribution:** 2
**Rating:** 5
**Confidence:** 4

**Summary:**

The authors propose a method where, instead of directly transferring a trained policy from a simulator to the real world, exploratory policies are learned in the simulator and transferred. This approach aims to enable efficient exploration in the real world, particularly in low-rank MDP settings.

**Strengths:**

1. The idea of transferring exploratory policies instead of directly transferring policies trained in simulator is interesting and can potentially help sim-to-real transfer.

**Weaknesses:**

1. The approach relies on several assumptions (e.g., low-rank MDPs, specific access to simulators and oracles) that may not hold in all real-world scenarios.

2. While the method shows promise in specific settings, its generality to a wider range of RL tasks and environments remains to be demonstrated.

3. Limited comparison with other state-of-the-art methods, especially those addressing the sim-to-real gap through domain randomization or adaptation.

**Questions:**

1. How robust is the method to violations of the low-rank MDP assumption?

2. Can the approach be extended to more complex MDP settings?

3. How does the method scale with the complexity of the task and the size of the state/action spaces?

**Limitations:**

As in weaknesses.

---

> ### Author Rebuttal · Authors · 2024-08-06
>
> We thank the reviewer for their feedback, and address questions and weaknesses below.
>
> ## Weaknesses
> 1. > The approach relies on several assumptions (e.g., low-rank MDPs, specific access to simulators and oracles) that may not hold in all real-world scenarios.
>
> Please see our comment to all reviewers for further justification of the low-rank MDP assumption. Briefly, we consider the low-rank MDP assumption to draw theoretical conclusions, but our proposed algorithm extends beyond the setting of low-rank MDPs, and our empirical results (on a real world task that does not strictly follow the setup of a low-rank MDP) show it is effective in general settings. Furthermore, we emphasize that the low-rank MDP setting is canonical in the theory community, and we believe that results in this setting are interesting in their own right.
>
> We note that our simulator access is quite weak—we only require black-box access to a simulator, which is the weakest type of access one could consider—and believe it is reasonable that we assume access to a simulator since we are studying sim2real transfer. Furthermore, we believe the oracles we consider are very reasonable—a regression oracle is quite standard and can be implemented even with neural networks, and many RL approaches exist that can successfully learn to solve a task in simulation (for example, standard approaches such as SAC and PPO are in general able to effectively solve tasks in simulation where samples are cheap).
>
> 2. > While the method shows promise in specific settings, its generality to a wider range of RL tasks and environments remains to be demonstrated.
>
>
> Please see our comment to all reviewers on what we believe are the key contributions of this work. Briefly, our primary contribution is theoretical: we provide the first result showing that simulators can provably help real-world RL in non-trivial settings. Our experimental results further validate this by showing that the algorithmic insights we derive are applicable in practice, which we demonstrate both through challenging sim2sim settings, as well as real-world robotic settings. We agree that further empirical work is necessary to fully validate the effectiveness of our proposal in general problem settings. Given the theoretical nature of the paper, however, we believe this is beyond the current scope, but is an interesting direction for future research.
>
> 3. > Limited comparison with other state-of-the-art methods, especially those addressing the sim-to-real gap through domain randomization or adaptation.
>
> Though it was not clearly stated in the paper, the baseline in our presented results on the real robot utilized domain randomization—on the Franka experiments, we randomized the friction, the size of the puck and the observation noise. Our results, therefore, show our proposed method is more sample efficient than doing only domain randomization. We will state this more clearly in the final version. We have also added an additional baseline, DIAYN [Eysenbach et al., 2018], to our sim2sim experiment—please see the rebuttal pdf, Figure 1.
>
> Regarding adaptation, we are primarily interested in settings where there is an irreducible gap between sim and real, given a fixed simulator. Indeed, for our Franka sim2real experiment, adaptation is unlikely to help us, as the parameters used are already chosen to match the real world dynamics as effectively as possible. The source of the sim2real gap is likely due to unmodeled dynamics, perhaps latency, and thus adaptation would require redesigning and improving the fidelity of our simulation.
>
> Eysenbach, Benjamin, et al. "Diversity is all you need: Learning skills without a reward function." arXiv preprint arXiv:1802.06070 (2018).
>
> ## Questions
>
> 1. > How robust is the method to violations of the low-rank MDP assumption?
>
> While we did not explicitly consider this in our theory, small violations of the low-rank MDP assumption (up to tolerance $O(\epsilon)$) would not affect our result. Furthermore, our empirical results illustrate that in practical settings where the low-rank assumption may or may not hold, our algorithmic insights are still effective. We remark as well that existing work shows that algorithms relying on the low-rank MDP assumption can be successfully applied to many standard RL benchmarks [Zhang et al., 2022].
>
> Zhang, Tianjun, et al. "Making linear mdps practical via contrastive representation learning." ICML, 2022.
>
> 2. > Can the approach be extended to more complex MDP settings?
>
> Our experiments illustrate that our algorithmic insights are still effective in more complex MDP settings. Extending our theory to more general settings (for example, bilinear classes) is an interesting direction for future work—at present it is unclear whether efficient sim2real transfer is possible in such settings.
>
> 3. > How does the method scale with the complexity of the task and the size of the state/action spaces?
>
> Theorem 1 shows that the sample complexity scales independently of the size of the action space, and only quadratically in the feature dimension (and does not directly scale with the number of states at all). We have also included additional experiments in the rebuttal pdf illustrating how our approach scales varying the number of states, actions, and horizon on the didactic example of Section 5.2.

---

> > ### Author Response · Authors · 2024-08-12
> >
> > We thank the reviewer for the time and effort spent reviewing our paper. As the discussion phase is about to end, we would like to make sure our responses have sufficiently addressed your concerns. We look forward to your feedback.

---

> > > ### Comment · Reviewer_ceeA · 2024-08-13
> > >
> > > Thanks authors for addressing the concerns, I raised my score to 5.

---

### Official Review · Reviewer_emUx · 2024-07-12

**Soundness:** 3
**Presentation:** 3
**Contribution:** 3
**Rating:** 7
**Confidence:** 2

**Summary:**

The paper shows that for transferring with a large sim-to-real gap, exploration policies have improved capabilities compared to optimally pre-trained policies, which tend to overfit the simulation up to a point where exploration is not sufficient to adapt to the changed circumstances.

**Strengths:**

Overall, the paper is easy to follow and theoretically sound. It includes theoretical deliberations and empirical results with a convincing experiment design. I expect the considered perspective on sim-to-real shift to positively impact research in this area.

**Weaknesses:**

Overall, providing further examples for the theoretical elaborations could help comprehensibility (e.g., Proposition 3). Further minor issues include:

- Definition 3.2: PAC is not defined, what is $\delta$ used for?
- In Proposition 1: Is the success of the transfer dependent on the horizon of M?
- In Proposition 2: What does $\Omega$ refer to?
- What is the difference between $\epsilon$ and $\epsilon_{sim}$?
- Why is the transition probability $P$ defined to be dependent on the horizon H?

Also, I am missing a thorough definition of the Low-Rank MDP, which seems central to the theoretical elaboration.

Even though the experimental setup is convincing, I am missing comparisons directly training $\pi$ in the real task as a baseline. Also, to better illustrate and link the aforementioned theoretical elaboration, providing the distance between the considered benchmark tasks would have been interesting. Furthermore, more detailed comparisons (both experimental and theoretical) to advanced pre-training approaches like APT (Liu et al., 2021) or DIAYN (Eysenbach et al., 2018) would further enhance empirical credibility.

**Questions:**

What are the implications of using a low-rank MDP?

If the simulation is considered to be free, wouldn't learning $\Pi$ optimal policies improve the transfer (similar to using a pool of exploration policies)?

Regarding the transfer described in Alg. 1, are all exploration policies updated, or only the best-performing?

If we were to consider even larger shifts, is there a point at which exploration policies no longer provide benefits, and learning from scratch is the most sufficient approach?

When considering the transfer from simulation to the real world, pre-training is often motivated by the need to ensure a certain level of safety for execution in the real world. How does this concern affect your approach?

**Limitations:**

Limitations have been addressed by the authors but could have been extended.

---

> ### Author Rebuttal · Authors · 2024-08-06
>
> We thank the reviewer for their helpful feedback.
>
> > Providing further examples for the theoretical elaborations…
>
> The example used to prove Prop. 1 and 2 is given in Sec. 5.2 and is a variant of the classic combination lock instance. In “real” the correct action must be taken for $H$ steps, and therefore, if we are exploring randomly, it will take exponentially long in $H$ to find this sequence of actions. In “sim”, the final transition is altered such that the optimal sequence in real is no longer optimal, and if we transfer the optimal sim policy to real, it does not reach the high reward region. However, if we transfer exploratory policies from sim, they are still able to traverse the MDP, and find the high reward, therefore quickly learning the optimal policy. Please see Figure 1 for a visual illustration of this construction.
>
> > Definition 3.2: PAC is not defined… What is the difference between $\epsilon$ and $\epsilon_{sim}$?
>
> PAC refers to “probably approximately correct”. $\delta$ denotes the confidence with which you hope to find an $\epsilon$-optimal policy. $\epsilon_{sim}$ is the amount of mismatch between sim and real.
>
> > In Proposition 1: Is the success of the transfer…
>
> The effectiveness of the policy learned in the simulator when deployed in real does depend on the horizon. This is illustrated in Prop. 1, which states that the optimal policy in sim can only be shown to be $2 H^2 \epsilon_{sim}$ optimal in real—as $H$ increases, the effectiveness of the policy learned in sim could decrease.
>
> > In Proposition 2: What does $\Omega$ refer to?
>
> $T \ge \Omega(2^H)$ means that $T$ must be on order $2^H$ (up to constants and lower-order terms).
>
> > Why is the transition probability $P$...
>
> This is common in the theory literature and simply lets us also handle cases where the transitions are time-dependent.
>
> > I am missing a thorough definition of the Low-Rank MDP
>
> We provide a standard definition for low-rank MDP in Def. 3.1. A low-rank MDP assumes that (1) there exists a featurization of the state-action pairs and (2) the transition functions can be approximated by a linear product based on the featurization. Note that the featurization does not need to be known a-priori.
>
> > I am missing comparisons directly training $\pi$ in the real task as a baseline.
>
> We have run an additional baseline training from scratch in “real” for both our sim2sim task and sim2real task. Please see the rebuttal pdf Figures 1 and 4. for these results. In both cases, we found that this performed significantly worse than our approach or direct policy transfer.
>
> > Providing the distance…
>
> For the didactic example (Section 5.2), the distance between the real and sim is $\epsilon_{sim} = 1/2$. For the other tasks we consider, there is unfortunately no straightforward way to measure the distance between tasks in terms of the closeness of their transitions, but highlight that our method does not require knowledge of $\epsilon_{sim}$.
>
> > More detailed comparisons…
>
> We have added a comparison to DIAYN for our sim2sim experiment (see Figure 1 in the rebuttal pdf), where we use DIAYN to train a set of exploration to transfer. While this performs worse than our approach—which instantiates our meta-algorithm, Algorithm 2, with a diversity-based approach similar to DIAYN, but also incorporating the task reward (see Algorithm 6)—we found it to still be somewhat effective. We remark that this use of DIAYN also fits directly within our meta-algorithm as an alternative method for generating exploration policies.
>
> DIAYN, APT, and all other practical approaches we are aware of lack theoretical guarantees, so a comparison between such approaches and our theoretical results is not possible. We are not aware of any other existing theoretical results that address the same problem we consider.
>
> > What are the implications of using a low-rank MDP?
>
> Please see our comment to all reviewers for discussion of the low-rank MDP assumption. Theoretically, this assumption is necessary to show that high-coverage exploration policies in the simulator will also yield effective coverage in real. Extending our work to more general settings is an interesting direction for future work. Empirically, our experimental results show that insights derived from low-rank MDPs are also relevant in more general practical settings.
>
> > If the simulation is considered to be free…
>
> Theoretically we do not believe this would improve the result—the exploration policies are already yielding sufficient data coverage, and transferring a family of near-optimal sim policies would not yield improved coverage. In practice this may yield improved performance, however, and indeed, this is similar to the method we utilize to generate the exploratory policies for our practical experiments (see Algorithm 6).
>
> > Regarding the transfer described in Alg. 1…
>
> In Algorithm 1, the exploration policies are not updated at all, they are simply played as is. The data collected from them is then used to train a single policy that solves the goal task in real.
>
> > If we were to consider even larger shifts…
>
> Yes, equation (4.1) in Theorem 1 gives a sufficient condition $\epsilon_{sim}$ must satisfy in order for exploration policy transfer to provably succeed. If this condition is not met, we are unable to guarantee exploration policy transfer succeeds.
>
> > When considering the transfer from simulation to the real world…
>
> While we did not explicitly consider it in this work, safety constraints could be incorporated by restricting the exploration policies so as not to simply explore, but to explore in a safe manner (e.g., incorporating safety-based approaches such as [Berkenkamp et al., 2017], or requiring that the exploration policies do not induce unsafe behavior in the simulator). Further study formalizing this is an interesting direction for future work.
>
> Berkenkamp, Felix, et al. "Safe model-based reinforcement learning with stability guarantees." NeurIPS, 2017.

---

> > ### Comment · Reviewer_emUx · 2024-08-12
> >
> > Thank you for your extensive response and clarifications and for providing additional experimental results as suggested. My concerns are fully addressed, and I hope some of the detailed elaborations are included in the final paper. I will retain my original, already positive score.

---

### Author Rebuttal · Authors · 2024-08-06

We thank each of the reviewers for their feedback and suggested improvements. We will work to incorporate all suggestions. We highlight several points regarding our contributions and the low-rank MDP setting, which we believe are relevant to all reviews.


### Key Contribution of Paper

We want to reiterate what we believe to be the core contributions of this paper. Our primary contribution is our theoretical results showing that simulators can yield a provable (exponential) gain in real-world RL, as compared to training only in the real world, and that this is true even in non-trivial settings where the mismatch between sim and real is large enough that direct policy transfer fails. To the best of our knowledge, this is the first theoretical justification for sim2real transfer in non-trivial settings.

Our experimental results further illustrate that the algorithmic principles our theoretical results motivate are applicable in real-world settings of interest, and yield a significant improvement over standard methods for sim2real transfer. However, we see this primarily as a proof-of-concept complementing our theory: additional empirical evidence would certainly be useful in providing a rigorous justification that this is a practical approach in a wide range of settings and, furthermore, more work could be done optimizing the instantiation of our meta-algorithm (Algorithm 2).

We have, however, added several additional empirical benchmarks to further support our empirical validation, as well as a new evaluation task (block hammering). The results for these experiments can be found in the rebuttal pdf.


### Low-Rank MDPs

Several reviewers commented on the limitation of requiring the low-rank MDP assumption. We make several comments on this. First, while the low-rank MDP assumption is required for the theory, our experimental results show that the algorithmic insights obtained from studying the low-rank MDP setting yield effective algorithmic approaches in more general settings. Thus, while the theory may be limited to the low-rank setting, the principles derived from studying this setting are much more general. Second, existing work has shown that the low-rank assumption often holds true in practice, and has applied existing algorithms for low-rank MDPs to a variety of standard RL benchmarks such as the Deepmind Control Suite [Zhang et al., 2022]. Finally, the low-rank MDP setting is a canonical setting in the RL theory community, and a significant amount of work has been devoted to understanding it (see, for example, references [1, 2, 36, 38, 53] in the paper). Thus, we believe that results on low-rank MDPs are of interest in their own right.

Zhang, Tianjun, et al. "Making linear mdps practical via contrastive representation learning." International Conference on Machine Learning. PMLR, 2022.

---

### Decision · Program_Chairs · 2024-09-25

**Decision:**

Accept (poster)

**Comment:**

A simple yet clear idea (transfer exploration policies, not exploitation policies) in an important and timely setting (sim2real), backed up with both theory and real-world robotics experiments. I concur with the reviewers' consensus that this paper should be accepted, even if there is room for further improvements (various ones pointed out in the reviews).